# State and Parameter Estimation of Two Land Surface Models Using the Ensemble Kalman Filter and the Particle Filter

Hongjuan Zhang[1,2], Harrie-Jan Hendricks Franssen[1,2], Xujun Han[1,2], Jasper A. Vrugt[3,4] and Harry Vereecken[1,2]

[1]Forschungszentrum Jülich, Agrosphere (IBG 3), Jülich, Germany
[2]Centre for High-Performance Scientific Computing in Terrestrial Systems: HPSC TerrSys, Forschungszentrum Jülich, Jülich, Germany
[3]Department of Civil and Environmental Engineering, University of California Irvine, Irvine, USA
[4]Department of Earth Systems Science, University of California Irvine, Irvine, USA

*Correspondence to*: Hongjuan Zhang (ho.zhang@fz-juelich.de)

**Abstract.** Land surface models (LSMs) use a large cohort of parameters and state variables to simulate the water and energy balance at the soil-atmosphere interface. Many of these model parameters cannot be measured directly in the field, and require calibration against measured fluxes of carbon dioxide, sensible and/or latent heat, and/or observations of the thermal and/or moisture state of the soil. Here, we evaluate the usefulness and applicability of four different data assimilation methods for joint parameter and state estimation of the Variable Infiltration Capacity Model (VIC-3L) and the Community Land Model (CLM) using a 5-month calibration (assimilation) period (March – July, 2012) of areal-averaged SPADE soil moisture measurements at 5, 20 and 50 cm depth in the Rollesbroich experimental test site in the Eifel mountain range in western Germany. We used the EnKF with state augmentation or dual estimation, respectively, and the residual resampling PF with a simple, statistically deficient, or more sophisticated, MCMC-based parameter resampling method. The performance of the "calibrated" LSM models was investigated using SPADE water content measurements of a 5-month evaluation period (August – December, 2012). As expected, all DA methods enhance the ability of the VIC and CLM models to describe spatiotemporal patterns of moisture storage within the vadose zone of the Rollesbroich site, particularly if the maximum baseflow velocity (VIC) or fractions of sand, clay, and organic matter of each layer (CLM) are estimated jointly with the model states of each soil layer. The differences between models soil moisture simulations of VIC-3L and CLM are much larger than the discrepancies among the four data assimilation methods. The EnKF with state augmentation or dual estimation yields the best performance of VIC-3L and CLM during the calibration and evaluation period, yet results are in close agreement with the PF using MCMC resampling. Overall, CLM demonstrated the best performance for the Rollesbroich site. The large systematic underestimation of water storage at 50 cm depth by VIC-3L during the first few months of the evaluation period questions, in part, the validity of its fixed water table depth at the bottom of the modelled soil domain.

## 1. Introduction and Scope

Land surface models (LSMs) are used widely to simulate and predict the exchanges of momentum, energy, and mass between the terrestrial biosphere and overlying atmosphere at local, regional, and global scales. These models also play a key role in assessing impacts of environmental changes (climate, land-use, and land-cover) on energy, water, and biogeochemical fluxes (e.g. $CO_2$, $CH_4$, $N_2O$) at the soil-atmosphere interface, and simplify

analysis of cause-effect relationships among the myriad of processes that govern land-atmosphere interactions and feedbacks, and emulate spatiotemporal variations in climate through greenhouse gas exchanges, carbon-nitrogen feedbacks, soil moisture-precipitation, and soil moisture-temperature coupling. LSMs use relatively simple mathematical equations to conceptualize and aggregate the complex, spatially distributed, and interrelated (bio)physical, chemical, and ecological processes that govern the exchange of mass, energy, and momentum between the land-surface and the atmosphere. This approach simplifies considerably the topology of the land-surface system, and reduces to much lower dimensions its state and parameter space. The consequence of this process aggregation and simplification is, however that the LSM parameters often do not represent directly measurable entities, and instead must be estimated via calibration by fitting the model against measured data records of soil moisture, soil temperature, and/or $CO_2$, water vapour, and/or energy fluxes across a range of biomes and timescales. These measurements are of crucial importance to quantify properly LSM parameter and predictive uncertainty, and to identify poorly represented or missing processes (Williams et al., 2009; Bonan, 2008).

Many of the parameters of a LSM are model dependent and therefore difficult to transfer between different land-surface schemes. Nevertheless, all LSMs use soil hydraulic, vegetation, and thermal parameters to describe heat transport, water flow, and root water uptake (canopy transpiration) in the variably saturated soil domain, and share a reflection coefficient (aka surface albedo) to calculate the reflected shortwave radiation. Two main approaches exist to determine the hydraulic and thermal properties of the considered soil domain. Some LSMs such as the Community Land Model (CLM) use basic soil data (soil texture and organic matter fraction) to estimate hydraulic and thermal parameters via pedotransfer functions (Oleson et al., 2013; Han et al., 2014; Vereecken et al., 2016). Other land-surface schemes such as the Variable Infiltration Capacity Model (VIC) (Liang et al., 1994; Gao et al., 2010) expect users to specify values for the hydraulic and thermal parameters. Pedotransfer functions are particularly useful in large-scale application of CLM as they simplify tremendously soil hydraulic characterization. Nevertheless, soil hydraulic parameter values derived from pedotransfer functions are subject to considerable uncertainty, and might therefore not accurately describe soil water movement and storage, particularly at larger spatial scales. What is more, (measurement) errors of the atmospheric forcing (e.g. wind speed, temperature, radiation, vapour pressure deficit, and precipitation) and errors in the auxiliary model input (e.g. topographic properties, vegetation characteristics) further enhance LSM prediction uncertainty.

In the past decades, many different search and optimization methods have been developed for automatic calibration of dynamic system models. Of these, Bayesian methods have found widespread application and use in Earth systems modelling due to their innate ability to treat, at least in principle, model input (forcing), output (forecast), parameter and structural errors. The Bayesian approach relaxes the assumption of a single optimum parameter value in favour of a posterior parameter and forecast distribution which summarizes the coordinated impact of different uncertainties on the modelling results. Yet, general-purpose methods such as DREAM (Vrugt et al., 2008, 2009; Vrugt, 2016) require a relatively large number of LSM evaluations to estimate parameter and forecast uncertainty. This can pose significant computational challenges for CPU-intensive and parameter-rich LSMs, and complicates treatment of input data uncertainty via latent variables (e.g. Vrugt et al., 2008).

Data assimilation offers an attractive alternative as general framework to account for LSM parameter, input, output, and other sources of uncertainty to take advantage of all available ground-based, airborne or spaceborne

observations to improve the compliance between numerical models and corresponding data. This approach enables joint estimation of model state variables and parameters and simplifies treatment of forcing data errors (Liu and Gupta, 2007). Many different studies published in the hydrologic literature have demonstrated the benefits of parameter estimation in the context of data assimilation for soil moisture characterization (e.g., Montzka et al., 2011; Lee, et al., 2014), rainfall-runoff (e.g., Moradkhani et al., 2005a; Vrugt et al., 2005) and land surface modelling (e.g., Pauwels et al., 2009).

Data assimilation methods merge uncertain observations with predictions (output) of imperfect models to optimally estimate the state of a dynamical system. The prototype of this method, the Kalman filter (KF) was developed in the 1960s by Rudy Kalman for optimal control of linear dynamical systems (Kalman, 1960). The KF is a maximum likelihood estimator of the dynamic state of the system if the model error and measurement error distributions are (multivariate) normal. For nonlinear dynamical models this Gaussian assumption is not generally valid, and the KF will not give a maximum likelihood state estimate. The ensemble Kalman Filter, or EnKF, is a stochastic generalization of the KF to nonlinear system models, in which the evolution of the model error covariance matrix is derived from a finite set of state realizations (Evensen, 1994). The use of this Monte Carlo ensemble not only makes possible state estimation for complex system models but also enables the explicit treatment of different sources of modelling error. Two decades on from its inception, the EnKF has received operational status in real-time weather, tsunami, and flood prediction systems (amongst others) due its proven ability to enhance a model's forecast skill and characterize accurately prediction uncertainty.

State estimation via the EnKF advances significantly the capabilities of hydrologic and land-surface models to predict spatiotemporal dynamics of water movement and storage in soils, lakes, and reservoirs, and fluxes of mass, energy, and momentum between the soil and the atmosphere. The predictive skill of these models is, however determined in large part by their parameterization. This has led hydrologists and hydrometeorologists to develop data assimilation approaches that permit the simultaneous inference of model state variables and parameter values. The power and usefulness of such joint state and parameter estimation methods have been investigated by many different authors in the water resources literature. Most of these publications use synthetic (or twin) experiments with assimilation of artificially generated data. Examples include studies with simulated measurements of the groundwater table depth or hydraulic head (Franssen and Kinzelbach, 2008; Bailey and Bau, 2012; Kurtz et al., 2014; Shi et al., 2014; Song et al., 2014; Tang et al., 2015), discharge/streamflow (Bailey and Bau, 2012; Moradkhani et al., 2012; Vrugt et al., 2013; Rasmussen et al., 2015), groundwater temperature (Kurtz et al., 2014), soil moisture (Wu and Margulis, 2011; Plaza et al., 2012; Erdal et al., 2014; Shi et al., 2014; Song et al., 2014; Pasetto et al., 2015), brightness temperature from passive remote sensing (Montzka et al., 2013; Han et al., 2014), and contaminant concentration (Gharamti et al., 2013). These studies use a variety of different methods for joint parameter and state estimation, among which the EnKF (Franssen and Kinzelbach, 2008; Wu et al., 2011; Gharamti et al., 2013; Erdal et al., 2014; Kurtz et al., 2014; Shi et al., 2014; Pasetto et al., 2015), the iterative EnKF (Song et al., 2014), the extended KF (Pauwels et al., 2009), the local ensemble transform KF (Han et al., 2014), the ensemble transform KF (Rasmussen et al., 2015), and the normal score EnKF (Tang et al., 2015).

The overarching conclusion from the body of synthetic experiments is that the joint estimation of parameters and state variables via data assimilation enhances significantly the predictive capabilities of hydrologic and land-surface models. This finding is corroborated by results for real-world assimilation studies documented in a

rapidly growing list of publications and involving model structural inadequacies, measurement errors of the atmospheric forcing variables and calibration (assimilation) data, inadequate characterization of the lower boundary condition (aquifer), and uncertainty of other, auxiliary, model inputs. This includes assimilation of measurements of the electrical conductivity (Wu and Margulis, 2013), hydraulic head in wells (Kurtz et al., 2014; Shi et al., 2015), groundwater temperature (Kurtz et al., 2014), streamflow and discharge (Moradkhani et al., 2012; Shi et al., 2015), active remote sensing (Pauwels et al., 2009), passive brightness temperature (Qin et al., 2009), soil moisture from lysimeters (Lue et al., 2011; Wu and Margulis, 2013; Erdal et al., 2014; Shi et al., 2015), land surface temperature (Bateni and Entekhabi, 2012) and sensible and latent heat fluxes (Shi et al., 2015) using methods such as the PF (Qin et al., 2009), PMCMC (Moradkhani et al., 2012), EnKF (Bateni and Entekhabi, 2012; Wu and Margulis, 2013; Erdal et al., 2014; Kurtz et al., 2014; Shi et al., 2015) and the extended KF (Pauwels et al., 2009; Lue et al., 2011). Despite this growing body of applications, relatively few studies (e.g. Lue et al., 2011; Shi et al., 2015) have focused on an accurate characterization of soil moisture dynamics simulated by LSMs. This is particularly surprising, as root zone moisture storage modulates spatiotemporal variations in climate and weather, and governs the production and health status of crops, and the organization of natural ecosystems and biodiversity (Vereecken et al., 2008).

In this paper, we evaluate the usefulness and applicability of four different data assimilation methods for joint parameter and state estimation of VIC-3L and CLM using a 5-month calibration (assimilation) period of soil moisture measurements at 5, 20 and 50 cm depth in the Rollesbroich experimental test site in the Eifel mountain range in western Germany. This grassland site is part of the TERENO network of observatories and extensively monitored since 2011 to catalogue long-term ecological, social and economic impact of global change at regional level. We used the EnKF with state augmentation (Chen and Zhang, 2006) or dual estimation (Moradkhani et al., 2005a), respectively, and the residual resampling PF (Douc et al., 2005) with a simple, statistically deficient (Moradkhani et al., 2005b), or more sophisticated, MCMC-based (Vrugt et al., 2013) parameter resampling method. The "calibrated" LSM models were tested using SPADE water content measurements from a 5-month evaluation period. To the best of our knowledge, this is only the second study after Chen et al. (2015) that compares sequential data assimilation methods for joint parameter and state estimation of a LSM. Related work by Dechant and Moradkhani (2012) and Dumedah and Coulibaly (2013) consider application to the rainfall-runoff transformation of a watershed.

The three main objectives of our study may be summarized as follows, (1) to evaluate the usefulness and applicability of joint parameter and state estimation for soil moisture characterization with LSMs, (2) to compare the performance of four commonly used parameter and state estimation methods in their ability to predict soil moisture dynamics at different depths in the Rollesbroich experimental test site, and (3) to compare, contrast and juxtapose the soil moisture simulations and predictions of CLM and VIC.

The remainder of this paper is organized as follows. Section 2 discusses briefly VIC-3L and CLM which are used as our LSMs to characterize soil moisture dynamics of the Rollesbroich experimental site in Germany. In this section, we contrast the numerical approaches, boundary conditions, and spatial discretization (soil layers), that are used by VIC-3L and CLM to describe water flow and storage in the modelled soil domain, and are particularly concerned with selection of their calibration parameters. Section 3 then reviews the basic concepts and theory of the four different data assimilation algorithms used herein. This is followed in section 4 with a detailed discussion of the Rollesbroich experimental site, and the numerical implementation and setup of each

data assimilation method. Section 5 introduces the results of the different parameter and state estimation methods and two LSMs, and section 6 discusses the main findings of our numerical experiments and assimilation studies. Finally, this paper concludes in section 7 with a summary of our main findings.

## 2. Land Surface Models and Calibration Parameters

LSMs simulate terrestrial biosphere fluxes of matter and energy via numerical solution of the water, energy, and carbon balance of the land-surface. This includes hydrologic processes such as soil evaporation, infiltration, surface runoff, canopy interception and transpiration, aquifer discharge, groundwater recharge, and precipitation (Schaake et al., 1996) and energy fluxes such as latent and sensible heat from soil, snow, surface water and vegetated surfaces (Bertoldi, 2004). Their respective equations contain parameters whose values depend on global or regional distributions of vegetation and soil properties (Milly and Shmakin, 2002).

The Rollesbroich site investigated herein covers an area of about 270,000 $m^2$ with grassland vegetation that is dominated by perennial ryegrass (Loliumperenne) and smooth meadow grass (Poapratensis). The limited size of our site and its rather uniform vegetation and topography, justify treatment of our land surface domain as a single grid cell in LSM with apparent parameters that characterize the mass and energy exchange between the soil and atmosphere. This assumption of homogeneity is computationally convenient and also simplifies somewhat our subsequent mathematical notation. We conveniently write the LSM as a (nonlinear) regression function, $\mathcal{M}(\cdot)$, which returns a $m \times n$ matrix $\mathbf{Y}$ with the simulated (predicted) values of $m$ different variables (e.g. soil moisture content, latent and sensible heat fluxes) at discrete times, $t \in \{1, \dots, n\}$, as follows

$$\mathbf{Y} \leftarrow \mathcal{M}(\boldsymbol{\alpha}, \tilde{\mathbf{x}}_0, \widetilde{\mathbf{B}}, \widetilde{\mathbf{U}}), \tag{1}$$

where $\boldsymbol{\alpha} = \{\alpha_1, \dots, \alpha_d\}$ is the $d$-vector of model parameters, $\tilde{\mathbf{x}}_0$ signifies the $k \times 1$-vector with measured (inferred) values of the state variables of the land surface model for the domain at the start of simulation, $\widetilde{\mathbf{B}}$ denotes the $l \times n$ control matrix with temporal measurements of $l$ forcing variables (e.g. air temperature, precipitation, vapour pressure deficit, wind speed, short and long-wave radiation), $\widetilde{\mathbf{U}}$ represents a vector with auxiliary constants, variables, or properties (e.g. plant functional type, land cover, soil texture, and other variables/constants) deemed necessary to simulate the water and energy balance of the land-surface domain of interest, and $\mathbf{Y} = [\mathbf{y}_{1:n}^1, \dots, \mathbf{y}_{1:n}^m]^T$, where $T$ denotes transpose. Without loss of generality, we restrict the model parameters to a closed space, $\mathbf{A}$, equivalent to some $d$-dimensional hypercube, $\boldsymbol{\alpha} \in \mathbf{A} \in \mathbb{R}^d$, called the feasible parameter space.

The assumption of homogeneity simplifies considerably the model definition in equation (1). Yet, this lumped topology might not characterize adequately real-world soil-land-surface systems that exhibit considerable spatial variations in soils, vegetation, and land properties. Such systems might necessitate distributed application of equation (1) via spatial discretization of the considered land-surface domain into different grid cells. This discretization should honour spatial variations in vegetation and soil properties, and could account for small-scale (within-grid-cell) variability. Nevertheless, in our present application of LSM we the treat the Rollesbroich site as a single grid cell with grassland vegetation and homogeneous, but layered, soil (details to follow).

We now discuss briefly two different land surface schemes, VIC-3L and CLM which are used to describe temporal variations in soil water storage at different depths in the Rollesbroich experimental site in Germany.

## 2.1. The Variable Infiltration Capacity Model (VIC)

The VIC model is a macro-scale semi-distributed hydrological model which solves for the water and energy balance of each grid cell using explicit consideration of within-grid-cell vegetation variations. Accordingly, each grid cell is divided into land cover tiles (Liang et al., 1994; Liang et al., 1996; Cherkauer and Lettenmaier, 1999), and assumes constant values of the soil properties (e.g., soil texture, hydraulic conductivity, thermal conductivity). The total evapotranspiration, sensible heat flux, effective land surface temperature and runoff are then obtained for each grid cell by summing over all the land cover tiles (vegetation types and bare soil) weighted by their respective fractional coverage (Gao et al., 2010). The VIC model can either be executed in a water balance mode or a water-and-energy balance mode. In this paper, we assume the latter and use a 70 cm deep soil composed of a 10 cm surface layer followed by a middle and bottom layer of 20 and 40 cm, respectively. The relatively thin surface layer is used to capture rapid fluctuations in soil moisture due to rainfall and bare soil evaporation, and the deepest and thickest layer summarizes seasonal water content dynamics and base flow. We use herein, VIC-3L, and force the model with atmospheric boundary conditions (e.g. precipitation, wind speed, air temperature, longwave and shortwave radiation, and relative humidity) for the Rollesbroich experimental site in Germany. In the absence of detailed information about the hydraulic properties of the considered soil domain, we treat each layer's saturated hydraulic conductivity, $\log_{10}k_s$ [$\log_{10}$(m/s)] and exponent of Brooks-Corey's drainage equation, $\beta$ [-], as calibration parameters. What is more, we also include the infiltration shape parameter, $b$ [-], and the maximum baseflow velocity, $D_m$ [mm/day] as calibration parameters. Thus, this involves estimation of $d = 8$ parameters in VIC-3L for the Rollesbroich site. Appendix A summarizes the soil module of VIC-3L, including a brief description of the main processes and model parameters.

## 2.2. The Community Land Model (CLM)

CLM is the land model for the Community Earth System Model (CESM) (Oleson et al., 2013), and is made up of multiple different building blocks, or modules, which resolve processes related to land biogeophysics, the hydrological cycle, biogeochemistry, and dynamic vegetation composition, structure, and phenology. The model recognizes explicitly surface heterogeneity by dividing each individual grid cell into multiple subgrid levels. For example, a grid cell can be made up of different land cover types, each with their own respective patches of plant functional types (PFTs) and associated stem area index and canopy height. The first subgrid level is defined by land units (vegetated, lake, urban, glacier, and crop), each composed of a number of different columns (second subgrid level) for which separate energy and water calculations are made. Vegetated land units, as well as lakes and glaciers, use one column. Urban land uses five separate columns, and for crop land there is a distinction between irrigated and unirrigated columns with one single crop occupying each column. The third subgrid level is composed of PFTs and includes bare soil. The vegetated column has 16 possible PFTs besides bare soil. For the crop column, several crop types are available. Processes such as canopy evaporation and transpiration are calculated for each individual PFT, whereas soil and snow processes are calculated at the column level using areal-weighted values of the properties of the PFTs of individual patches. Note, that a similar aggregation approach is used by VIC-3L.

In our application of CLM to the Rollesbroich experimental site in Germany, we calculate soil temperature for 15 different soil layers, and simulate hydrological states and fluxes for the top 10 soil layers only. Appendix B

presents a brief description of the soil module of CLM, and discusses the main parameters used. The CLM is forced with atmospheric conditions (e.g. precipitation, vapour pressure deficit, wind speed, incoming short wave and long wave radiation) using values for the model parameters and initial states, and land surface data and other physical constants and/or variables as auxiliary input. The soil hydraulic (e.g. saturated hydraulic conductivity) and thermal parameters of CLM are derived from built-in pedotransfer functions (see Equations (B1) - (B4) of Appendix B) using as inputs the auxiliary vector/matrix $\tilde{\mathbf{U}}$ with sand, clay and organic matter fractions of each individual soil layer. We treat these auxiliary soil variables as unknown parameters in the present application of CLM. Thus, this involves $d = 30$ parameters in CLM for the Rollesbroich site.

### 2.3. Main Differences of VIC-3L and CLM

Before we proceed, we first summarize the main differences of VIC-3L and CLM in their calculations of the water and energy balance of the land-surface. In the first place, VIC-3L treats the vadose zone as a multi-layer bucket with variable infiltration capacity, whereas CLM uses a more physics-based description of soil water movement, storage, and associated hydrological fluxes (e.g. root water uptake) by numerical solution of a modified form of Richards' equation (Zeng and Decker, 2009). A bucket model is computationally convenient, but sacrifices important detail regarding the vertical distribution of soil water storage. The latter is a prerequisite to characterize accurately processes such as infiltration, redistribution, root-water uptake, drainage, and groundwater recharge. We refer the interested reader to Romano et al. (2011) for a detailed comparison of bucket-type and Richards' based vadose zone flow models.

Second, VIC-3L treats the saturated and variably-saturated soil domain as two separate, lumped, control volumes which are decoupled from the underlying groundwater reservoir. In other words, a fixed lower boundary condition is imposed. CLM, on the contrary, simulates interactions between the modelled soil domain and an unconfined aquifer. The resulting water table variations of the aquifer affect soil water movement in the unsaturated zone via a variable recharge flux. In our application of CLM, this recharge flux emanates at the bottom of the tenth soil layer. The calculation of this recharge flux may be best explained via the use of a virtual soil layer, say layer 11, whose depth extends from the bottom of layer 10 to the groundwater table. If we assume hydrostatic conditions in layer 11, then we can calculate the recharge flux from layer 10 using Equation (B9) in Appendix B. This recharge flux then changes the depth of the water table according to Equation (B11). This Equation also takes into consideration drainage from the water table due to topographic gradients. If the groundwater table is within the upper 10 soil layers, a drainage flux emanates from the upper most saturated layer according Equation (B10).

Third, VIC-3L expects the user to specify values for the soil hydraulic (e.g. saturated hydraulic conductivity), thermal, and baseflow parameters of the first, second, and third layer of each grid-cell, respectively, whereas CLM derives their counterparts (e.g. hydraulic conductivity at saturation, matric head at saturation, Clapp-Hornberger exponent $B$, and soil thermal conductivity) for each of the fifteen soil layers using built-in pedotransfer functions.

Finally, VIC-3L allows the user to determine freely the number and thickness of the soil layers in the bucket model (default is three layers), whereas CLM assumes a fixed thickness of each soil layer.

### 2.4. Selection of Calibration Parameters

LSMs contain a large number of parameters whose values can be adjusted by fitting model output to observed data. Yet, only a few of those parameters will affect noticeably model performance. Various authors have investigated the parameter sensitivity of VIC-3L via Monte Carlo simulation, Generalized Likelihood Uncertainty Estimation (GLUE), or model calibration methods (Demaria et al., 2007; Xie et al., 2007; Troy et al., 2008). These studies demonstrated a strong dependency of parameter sensitivity on climatic conditions. Table 1 lists the VIC-3L and CLM parameters that have been selected for calibration via data assimilation, and reports their units, feasible ranges, perturbation, and spatial configuration. To honour prior information (e.g. soil textural data) we do not draw the model parameters from their feasible ranges, but rather sample their initial values around some best-guess VIC-3L and CLM parameterization using the normal and uniform distributions listed under the header "Perturbation". This makes up the prior parameter distribution and is further explained in section 4.2.

Appendix A (VIC-3L) and B (CLM) summarize the main variables, processes, and equations which are used by both models to describe the storage and vertical and/or horizontal movement of water in the variably-saturated soil domain of the Rollesbroich site. These two appendices help to better understand the role of the different calibration parameters of Table 1, and will be most beneficial to readers which are rather unfamiliar with both models. Note that CLM estimates the hydraulic and thermal parameters of each soil layer from built-in pedotransfer functions (Oleson et al., 2013; Han et al., 2014) using as input the sand, clay, and organic matter fraction of each soil layer.

## 3. Data Assimilation Methods

Data assimilation methods merge uncertain observations with predictions (output) of imperfect models to optimally estimate the state and/or parameters of a dynamical system. This includes the use of four-dimensional variational data assimilation (4D-Var), EnKF, PF, and related assimilation schemes. These methods have been applied successfully to a large number of different fields for model-data fusion in the atmospheric, oceanic, biogeochemical and hydrological sciences. We now briefly discuss the theory of four different data assimilation methods which are used herein with VIC-3L and CLM to characterize spatiotemporal soil moisture dynamics at our experimental site.

### 3.1. EnKF

The EnKF was proposed by Evensen (1994) as generalization of the Kalman filter to nonlinear system models with many state variables. This method uses a Monte Carlo approach to generate an ensemble of different model trajectories from which the time evolution of the probability density of the model states, and related error covariances are estimated. The EnKF uses a state-space implementation of the dynamic system model of equation (1) and implements the following steps (Burgers et al., 1998):

$$\mathbf{x}_t^{i-} = \mathcal{M}(\boldsymbol{\alpha}, \mathbf{x}_{t-1}^i, \tilde{\mathbf{b}}_{t-1}^i, \mathbf{I}) + \mathbf{w}_t, \tag{2}$$

where $\mathbf{x}_t^{i-}$ is the $k \times 1$ vector of predicted values of the state variables of the $i$th ensemble member, $i = \{1,\dots,N\}$, $\tilde{\mathbf{b}}_{t-1}^i$ signifies the corresponding vector (or matrix) of measured values of the forcing variables, $\mathbf{w}_t$ denotes a $k \times 1$ process noise vector that accounts for structural imperfections of the LSM, and $t$ denotes time. In our specific implementation, the state vector is made up of areal-averaged soil moisture content values at three

different measurement depths. What is more, the observed precipitation forcing was perturbed with a member-dependent vector of measurement errors. From the ensemble of $N$ state vectors, we can calculate the $k \times k$ background error covariance matrix, $\mathbf{C}$, using:

$$\mathbf{C} = \frac{1}{N-1} \sum_{i=1}^{N} (\mathbf{x}_t^{i-} - \bar{\mathbf{x}}_t)(\mathbf{x}_t^{i-} - \bar{\mathbf{x}}_t), \tag{3}$$

where $\bar{\mathbf{x}}_t$ denotes the $k \times 1$ vector with ensemble mean values of the states at time $t$. The $m \times 1$ vector of measured soil moisture data at time $t$ can be written for each individual ensemble member as follows:

$$\hat{\mathbf{y}}_t^i = \tilde{\mathbf{y}}_t + \mathbf{v}_t^i, \tag{4}$$

where $\mathbf{v}_t^i$ signifies a $m \times 1$ vector of measurement errors drawn randomly from a $m$-variate normal distribution, $\mathcal{N}_m(0, \mathbf{R})$ with zero-mean and $m \times m$ observation error covariance matrix $\mathbf{R}$. We assume the soil moisture measurement errors at each depth to have a fixed and common variance $\sigma^2$, and to be uncorrelated in space and time. Thus we can write $\mathbf{R} = \sigma^2 \mathbf{I}_m$, where $\mathbf{I}_m$ signifies the $m \times m$ identity matrix with zeros everywhere except on the main diagonal which stores values of $\sigma^2$.

We can now update the predicted state values of each ensemble member as follows

$$\mathbf{x}_t^i = \mathbf{x}_t^{i-} + \mathbf{K}(\hat{\mathbf{y}}_t^i - \mathbf{H}\mathbf{x}_t^{i-}), \tag{5}$$

where $\mathbf{x}_t^i$ denotes the $k \times 1$ vector with updated estimates of the state variables (also called analysis state), $\mathbf{K}$ is a $k \times m$ matrix called the Kalman gain, and the $m \times k$ matrix $\mathbf{H}$ signifies the measurement operator which maps the model output to the measurement space. It is linear for EnKF. In our present application, we observe directly the soil moisture content of respective measurement depth, and thus the matrix $\mathbf{H}$ is made up of values of zero and unity. The Kalman gain is computed as follows:

$$\mathbf{K} = \mathbf{C}\mathbf{H}^T(\mathbf{H}\mathbf{C}\mathbf{H}^T + \mathbf{R})^{-1}, \tag{6}$$

where the symbol $T$ denotes transpose. The updated values of the states now enter equation (2) and are used to predict the soil moisture values at the next observation time, $t = t + 1$, and so forth.

In some cases it might be appropriate to estimate the model parameters along with the state variables. This requires a slight modification to the state-space formulation of equation (2) as the $d$-vector of parameter values, $\boldsymbol{\alpha}$, must now vary among the $N$ ensemble members to facilitate parameter estimation from the measured data. Three different approaches have been published in the literature for joint estimation of model states and parameters in the EnKF. This includes, state augmentation, dual and outer estimation. The first two approaches assume the LSM parameters to be time-variant, and infer their values sequentially along with the model states. The third approach assumes the parameters to be time-invariant, and estimates their posterior distribution in a loop outside the EnKF by maximizing the marginal likelihood of the $N$ state trajectories (Vrugt et al., 2005, 2013). We will consider herein only the first two approaches, that is, state augmentation and dual estimation, as these two methods are most CPU-efficient.

### 3.1.1. State augmentation

In state augmentation, the $k \times 1$ vector of state variables, $\mathbf{x}_t$, the model error covariance matrix $\mathbf{C}$, the measurement operator $\mathbf{H}$, and the Kalman gain $\mathbf{K}$ consist of two separate blocks (Franssen and Kinzelbach, 2008):

$$\mathbf{x}^{*i} = \begin{bmatrix} \mathbf{x}^i \\ \boldsymbol{\alpha}^i \end{bmatrix} \tag{7}$$

$$\mathbf{C}^* = \begin{bmatrix} \mathbf{C_{xx}} & \mathbf{C}_{\alpha x}^T \\ \mathbf{C}_{\alpha x} & \mathbf{C}_{\alpha\alpha} \end{bmatrix} \tag{8}$$

$$\mathbf{H}^* = [\mathbf{H_x}, 0], \tag{9}$$

where the subscripts $\mathbf{x}$ and $\boldsymbol{\alpha}$ refer to the model states and parameters respectively. The state vector, $\mathbf{x}^*$, now consists of $k + d$ elements, the model error covariance matrix $\mathbf{C}^*$ is made up of four smaller matrices, $\mathbf{C_{xx}}$, $\mathbf{C}_{\alpha x}^T$, $\mathbf{C}_{\alpha x}$, and $\mathbf{C}_{\alpha\alpha}$, and the measurement operator $\mathbf{H}^*$ includes $\mathbf{H_x}$ and additional values of zero. The Kalman gain matrix $\mathbf{K}$ is now given by:

$$\mathbf{K} = \mathbf{C}^*\mathbf{H}^{*T}(\mathbf{H}^*\mathbf{C}^*\mathbf{H}^{*T} + \mathbf{R})^{-1} = \begin{bmatrix} \mathbf{C_{xx}} & \mathbf{C}_{\alpha x}^T \\ \mathbf{C}_{\alpha x} & \mathbf{C}_{\alpha\alpha} \end{bmatrix} \begin{bmatrix} \mathbf{H}_x^T \\ 0 \end{bmatrix} \left( [\mathbf{H_x}, 0] \begin{bmatrix} \mathbf{C_{xx}} & \mathbf{C}_{\alpha x}^T \\ \mathbf{C}_{\alpha x} & \mathbf{C}_{\alpha\alpha} \end{bmatrix} \begin{bmatrix} \mathbf{H}_x^T \\ 0 \end{bmatrix} + \mathbf{R} \right)^{-1}$$

$$= \begin{bmatrix} \mathbf{C_{xx}}\mathbf{H}_x^T(\mathbf{H_x}\mathbf{C_{xx}}\mathbf{H}_x^T + \mathbf{R})^{-1} \\ \mathbf{C}_{\alpha x}\mathbf{H}_x^T(\mathbf{H_x}\mathbf{C_{xx}}\mathbf{H}_x^T + \mathbf{R})^{-1} \end{bmatrix}$$

$$= \begin{bmatrix} \mathbf{K_x} \\ \mathbf{K}_\alpha \end{bmatrix}. \tag{10}$$

This results in the following equation for the updated states and parameter values:

$$\begin{bmatrix} \mathbf{x}_t^i \\ \boldsymbol{\alpha}_t^i \end{bmatrix} = \begin{bmatrix} \mathbf{x}_t^{i-} \\ \boldsymbol{\alpha}_t^{i-} \end{bmatrix} + \begin{bmatrix} \mathbf{K_x}(\hat{\mathbf{y}}_t^i - \mathbf{H_x}\mathbf{x}_t^{i-}) \\ \mathbf{K}_\alpha(\hat{\mathbf{y}}_t^i - \mathbf{H_x}\mathbf{x}_t^{i-}) \end{bmatrix}. \tag{11}$$

### 3.1.2. Dual estimation

In the dual estimation approach, the state variables and model parameters are stored in two separate vectors and updated using their own individual steps (Moradkhani et al., 2005a). The parameter values of each ensemble member are first updated according to:

$$\boldsymbol{\alpha}_t^i = \boldsymbol{\alpha}_t^{i-} + \mathbf{K}_\alpha(\hat{\mathbf{y}}_t^i - \mathbf{H_x}\mathbf{x}_t^{i-}). \tag{12}$$

Then, the updated parameter values are used with equation (2) to predict, for the second time, the state variables at time $t$, after which their values are updated via equation (5). This approach necessitates running the LSM twice for the time period between two successive measurements, thereby doubling the required CPU-time of each ensemble member for this dual estimation method compared to the state augmentation approach.

The EnKF suffers from filter inbreeding, that is, the ensemble spread degrades after several data assimilation steps. In extreme cases, the covariance matrix $\mathbf{C}$, of the state ensemble is so small that the measurements receive a negligible weight via equation (6) and do not affect much the state trajectories of the individual ensemble members. This reflects a situation similar to model calibration in which state variable errors are ignored and all uncertainty in the input-output representation of the model is attributed to the parameters. Filter inbreeding is aggravated by the use of a relatively low number of ensemble members (small $N$) which results in spurious correlations among state variables and/or parameters, and underestimation of the spread of the ensemble. Other reasons for an insufficient ensemble spread are model structural errors, and the use of an underdispersed prior parameter distribution or too small variance of the measurement errors of the forcing variables. Ensemble inflation methods are an effective way to ameliorate filter inbreeding (Anderson, 2007; Whitaker and Hamill, 2012). We apply the inflation algorithm of Whitaker and Hamill (2012) to the $d$ parameter values of each ensemble member as follows:

$$\alpha_{j,t}^i = \bar{\alpha}_{j,t} + \frac{V_j}{W_j}\left(\alpha_{j,t}^i - \bar{\alpha}_{j,t}\right), \tag{13}$$

where $\bar{\alpha}_{j,t}$ signifies the analysis mean (after update) of the $j$th parameter at time $t$, the scalars $V_j$ and $W_j$ denote

the prior (before update) and analysis standard deviation of the $j$th parameter (derived from ensemble), and $j = \{1, \dots, d\}$. This method promotes a parameter spread that is in agreement with the width of the prior parameter distribution, and is particularly important to avoid a strong underestimation of ensemble variance and associated filter inbreeding in applications with relatively small ensemble sizes. As the spread is kept artificially constant, it cannot be assessed properly how data assimilation affects reduction of prediction uncertainty. In

addition, it is important that the initial ensemble spread is adequate. This is a drawback of the applied inflation.

### 3.2 Residual Resampling Particle Filter (RRPF) and Parameter Estimation

The PF was first suggested in the research area of object recognition, robotics and target tracking (Gordon et al., 1993) and was introduced to hydrology by Moradkhani et al. (2005a). The PF differs from the EnKF in that it describes the evolving probability density function (PDF) of the LSM state variables by a set of $N$ random

samples, also called particles. Each particle carries a non-zero weight which determines its underlying probability, and these weights are updated as soon as a new datum (observation) becomes available. Before we proceed with a brief theoretical description of the PF we must first explicate our notation. We denote with symbol $\mathbf{X}_{1:t}$ the collection of simulated values of the LSM state variables between the first observation at $t = 1$ and the present datum, $t$, hence $\mathbf{X}_{1:t} = [\mathbf{x}_1, \dots, \mathbf{x}_t]$ is a $k \times t$ matrix with the LSM states at each measurement

time stored as a column vector. The corresponding observations are stored in the $m \times t$ matrix, $\widetilde{\mathbf{Y}}_{1:t} = [\tilde{\mathbf{y}}_1, \dots, \tilde{\mathbf{y}}_t]$. Finally, we use braces, $\{\cdot\}$, to denote our Monte Carlo ensemble of $N$ particle trajectories, $\{\mathbf{X}_{1:t}^{1:N}\}$, and thus $\{\mathbf{x}_t^{1:N}\}$ is a $k \times N$ matrix with sampled values of the LSM state variables at time $t$. The subsequent description of the PF follows closely the description of Vrugt et al. (2013). Interested readers are referred to this publication for further details.

If we assume the parameters to be known, then we can write the evolving posterior distribution, $p_{\alpha}(\mathbf{X}_{1:t}|\widetilde{\mathbf{Y}}_{1:t})$, for the state-space formulation of equation (2) as follows:

$$p_{\alpha}\left(\mathbf{X}_{1:t}\big|\widetilde{\mathbf{Y}}_{1:t}\right) = \overbrace{p_{\alpha}\left(\mathbf{X}_{1:t-1}\big|\widetilde{\mathbf{Y}}_{1:t-1}\right)}^{\text{prior}} \underbrace{\frac{\overbrace{\mathcal{M}_{\alpha}(\mathbf{x}_t|\mathbf{x}_{t-1})}^{\text{model}}\overbrace{L_{\alpha}(\tilde{\mathbf{y}}_t|\mathbf{x}_t)}^{\text{likelihood function}}}{p_{\alpha}(\tilde{\mathbf{y}}_t|\widetilde{\mathbf{Y}}_{1:t-1})}}_{\text{normalization constant}}, \tag{14}$$

where $p_{\alpha}(\mathbf{X}_{1:t-1}|\widetilde{\mathbf{Y}}_{1:t-1})$ denotes the prior state distribution, $\mathcal{M}_{\alpha}(\mathbf{x}_t|\mathbf{x}_{t-1})$ signifies the transition probability density of the state variables (= equation (2)), $L_{\alpha}(\tilde{\mathbf{y}}_t|\mathbf{x}_t)$ is the likelihood function, and $p_{\alpha}(\tilde{\mathbf{y}}_t|\widetilde{\mathbf{Y}}_{1:t-1})$ represents a

normalization constant which ensures that the posterior state distribution integrates to unity. Equation (14) follows directly from Bayes' law (see Appendix A of Vrugt et al. (2013)), and does not use at once the data up to time $t$ to estimate $p_{\alpha}(\mathbf{X}_{1:t}|\widetilde{\mathbf{Y}}_{1:t})$ but rather estimates the evolving system state recursively over time using some mathematical model and new incoming measurements. If we integrate out the state trajectory $\mathbf{X}_{1:t-1}$ from equation (14) then we can derive an expression for the marginal PDF of the state variables, $p_{\alpha}(\mathbf{x}_t|\widetilde{\mathbf{Y}}_{1:t})$, at time

$t$:

$$p_{\alpha}\left(\mathbf{x}_t\big|\widetilde{\mathbf{Y}}_{1:t}\right) = \frac{L_{\alpha}(\tilde{\mathbf{y}}_t|\mathbf{x}_t)p_{\alpha}(\mathbf{x}_t|\widetilde{\mathbf{Y}}_{1:t-1})}{p_{\alpha}(\tilde{\mathbf{y}}_t|\widetilde{\mathbf{Y}}_{1:t-1})}, \tag{15}$$

which is also referred to as the **update step** of the optimal filter (conditional independence of measurements). The state **prediction step** is equivalent to the Chapman-Kolmogorov equation:

$$p_\alpha(\mathbf{x}_t|\widetilde{\mathbf{Y}}_{1:t-1}) = \int_\Omega \mathcal{M}_\alpha(\mathbf{x}_t|\mathbf{x}_{t-1}) \, p_\alpha(\mathbf{x}_{t-1}|\widetilde{\mathbf{Y}}_{1:t-1}) \mathrm{d}\mathbf{x}_{t-1}, \tag{16}$$

where $\Omega$ signifies the feasible state space.

We conveniently assume herein, a Gaussian likelihood function:

$$L_\alpha(\tilde{\mathbf{y}}_t|\mathbf{x}_t) = \frac{1}{(2\pi)^{m/2}|\mathbf{R}|^{1/2}} \exp\left(-\frac{1}{2}(\tilde{\mathbf{y}}_t - \mathbf{H_x}\mathbf{x}_t)^T \mathbf{R}^{-1}(\tilde{\mathbf{y}}_t - \mathbf{H_x}\mathbf{x}_t)\right), \tag{17}$$

where $\mathbf{R}$ is the $m \times m$ measurement error covariance matrix, $|\cdot|$ signifies the determinant operator, and $m$ denotes the length of the observation vector, $\tilde{\mathbf{y}}_t$, at time $t$.

The PF makes use of the following identity of equation (14) to approximate the evolving state PDF:

$$p_\alpha(\mathbf{X}_{1:t}|\widetilde{\mathbf{Y}}_{1:t}) \propto p_\alpha(\mathbf{X}_{1:t-1}|\widetilde{\mathbf{Y}}_{1:t-1})\mathcal{M}_\alpha(\mathbf{x}_t|\mathbf{x}_{t-1})L_\alpha(\tilde{\mathbf{y}}_t|\mathbf{x}_t). \tag{18}$$

This recursion implies that we can use reuse the particles (samples) at $t-1$ that define the prior distribution, $p_\alpha(\mathbf{X}_{1:t-1}|\widetilde{\mathbf{Y}}_{1:t-1})$, to approximate the posterior state PDF, $p_\alpha(\mathbf{X}_{1:t}|\widetilde{\mathbf{Y}}_{1:t})$, at the next observation time. Yet, such recycling poses a problem, that is, we cannot sample directly from $p_\alpha(\mathbf{X}_{1:t}|\widetilde{\mathbf{Y}}_{1:t})$ as we do not know its

multivariate distribution. We therefore resort to an easy-to-sample-from importance density, $q_\alpha(\cdot\,|\mathbf{x}_{t-1}, \tilde{\mathbf{y}}_t)$, and draw $\{\mathbf{x}_t^{1:N}\}$ taking into consideration the current observation, $\tilde{\mathbf{y}}_t$, and previous state samples, $\{\mathbf{x}_{t-1}^{1:N}\}$. We then calculate the unnormalized importance weight of the $i$th particle, $W_t^i$, as follows

$$W_t^i \propto \overline{W}_{t-1}^i w_t(\{\mathbf{X}_{1:t}^i\}), \tag{19}$$

where $w_t(\mathbf{X}_{1:t}^i)$ signifies the incremental importance weight:

$$w_t(\{\mathbf{X}_{1:t}^i\}) = \frac{\mathcal{M}_\alpha(\{\mathbf{x}_t^i\}|\{\mathbf{x}_{t-1}^i\})L_\alpha(\tilde{\mathbf{y}}_t|\{\mathbf{x}_t^i\})}{q_\alpha(\{\mathbf{x}_t^i\}|\{\mathbf{x}_{t-1}^i\}, \tilde{\mathbf{y}}_t)}, \tag{20}$$

and $\overline{W}_t^i = W_t^i / \sum_{i=1}^N W_t^i$ denote the normalized importance weights, which vary between 0 and 1.

Before we can implement the PF in practice, we need to specify the importance density, $q_\alpha(\cdot\,|\{\mathbf{x}_{t-1}^{1:N}\}, \tilde{\mathbf{y}}_t)$, for $t = \{2, \dots, n\}$. We follow Gordon et al. (1993) and set $q_\alpha(\mathbf{x}_t|\mathbf{x}_{t-1}, \tilde{\mathbf{y}}_t) = \mathcal{M}_\alpha(\mathbf{x}_t|\mathbf{x}_{t-1})$ which results in the following equation for the incremental particle weights:

$$w_t(\{\mathbf{X}_{1:t}^i\}) = \frac{\mathcal{M}_\alpha(\{\mathbf{x}_t^i\}|\{\mathbf{x}_{t-1}^i\})L_\alpha(\tilde{\mathbf{y}}_t|\{\mathbf{x}_t^i\})}{\mathcal{M}_\alpha(\{\mathbf{x}_t^i\}|\{\mathbf{x}_{t-1}^i\})} = L_\alpha(\tilde{\mathbf{y}}_t|\{\mathbf{x}_t^i\}). \tag{21}$$

This approach gives satisfactory results if the transition density or model operator, $\mathcal{M}_\alpha(\mathbf{x}_t|\mathbf{x}_{t-1})$, adequately describes the observed system dynamics, and/or the observations, $\widetilde{\mathbf{Y}}_{1:t}$, are not too informative. Otherwise, the repeated application of equation (19) causes particle impoverishment in which the sampled particle trajectories drift away from the actual posterior state distribution, and receive a negligible importance weight. This ensemble

degeneracy (e.g. Carpenter et al., 1999) deteriorates PF performance and results in a poor computational efficiency of the filter as much of the CPU-time is devoted to carrying forward particle trajectories whose contribution to $p_\alpha(\mathbf{X}_{1:t}|\widetilde{\mathbf{Y}}_{1:t})$ for $t > 1$ is virtually zero.

To combat particle degeneracy we monitor the effective sample size (ESS) after assimilation of each new observation:

$$\text{ESS} = 1/\sum_{i=1}^{N}\left(\overline{W}_t^i\right)^2. \tag{22}$$

If the ESS is smaller than some default threshold, say $N/2$, then the particle ensemble is said to be degenerating. Several methods have been developed in the statistical literature to rejuvenate the particle ensemble. Gordon et al. (1993) introduced Sequential Importance Resampling (SIR), where $N$ particles are drawn from the ensemble using selection probabilities equal to their normalized importance weights. This step replaces samples with low importance weights with exact copies of the most promising particles, and produces a resampled set of $N$ particles with equal weights of $1/N$. In our application of the PF we implement Residual Resampling (RR) developed by Liu and Chen (1998). This method has an important advantage over SIR in that it produces a resampled set of particles with more diverse weights (Weerts and Serafy, 2006). First, we compute a selection probability, $p_{\{x_t^i\}}$, of each individual particle as follows:

$$p_{\{x_t^i\}} = \frac{N\overline{W}_t^i - \lfloor N\overline{W}_t^i \rfloor}{N - M}, \tag{23}$$

where the $\lfloor \cdot \rfloor$ operator rounds down to the nearest integer, and $M = \sum_{j=1}^{N}\lfloor N\overline{W}_t^j \rfloor$. Then, the $M$ particles with largest normalized importance weights are retained, and the remaining $N - M$ spots are filled by drawing from the $M$ retained particles using their selection probabilities from equation (23). The resulting filter is referred to as RRPF.

In the present application of the RRPF, we not only estimate the LSM states but also jointly infer the values of the model parameters. We use state augmentation and add the model parameters to the vector of LSM state variables. Yet, this approach requires definition of an importance density for the parameters to avoid parameter impoverishment after several successive assimilation steps. This has been demonstrated numerically by Plaza et al. (2012) using a series of data assimilation experiments. In principle, we could corrupt the posterior parameter distribution using the ensemble inflation method of Whitaker and Hamill (2012) detailed in equation (13). This approach was used by Qin et al. (2009) to avoid degeneracy of the parameter values. Instead, we use the approach described by Plaza et al. (2012) and perturb the parameter values of the resampled particles using draws from a zero-mean $d$-variate Gaussian distribution with diagonal covariance matrix. This $d \times d$ matrix has zero entries everywhere (uncorrelated dimensions) except on the main diagonal which stores values of $s^2\text{Var}\left[\{\alpha_{0,j}^{1:N}\}\right]$, where $s$ is a scaling factor, $\text{Var}\left[\{\alpha_{0,j}^{1:N}\}\right]$ signifies the prior variance of the $j$th parameter (at $t = 0$), and $j = \{1, \ldots, d\}$. This is an adaptation of the method introduced by Moradkhani et al. (2005b) and uses the prior variance of the parameters rather than their variance at the previous measurement time, $t - 1$. Yet, in the absence of a formal guidelines on the choice of $s$, this perturbation approach suffers from a lack of adequate statistical underpinning [Vrugt et a., 2013; Yan et al., 2015]. In our present application, we set $s = 0.1$, and evaluate the RRPF performance for VIC-3L model using other values for this scaling factor as well.

### 3.3. Particle Markov Chain Monte Carlo (PMCMC) Simulation

The RR procedure produces a sample with more evenly distributed weights, but many of the particles are exact copies of one another. To enhance sample diversity, we therefore evaluate another resampling step using Markov

chain Monte Carlo (MCMC) simulation. We follow herein the MCMC resampling method of Vrugt et al. (2013) and create candidate particles after RR using a discrete proposal distribution with state and parameter jumps equal to a multiple of the difference of two or more pairs of resampled particles. Each candidate particle is then re-evaluated between $t-1$ and $t$ by the LSM model, and the Metropolis acceptance probability is used to determine whether to replace the "old" particle or not. This combined PF and MCMC methodology is also referred to as PMCMC. Interested readers are referred to Vrugt et al. (2013) for a detailed description of this method.

### 3.4. Important Differences of EnKF and PF

Before we proceed with application of the EnKF-AUG, EnKF-DUAL, RRPF and PMCMC data assimilation methods, we reminisce about the key differences of the EnKF and PF. These differences are often overlooked and misunderstood but of crucial importance to help understand the two filters, and analyse and interpret our findings (see Vrugt et al., 2013). Most critically, the EnKF uses the measured values of the state variables (via measurement operator, if appropriate) to correct (update) the forecasted states of each ensemble member. The state PDF at each time is approximated by a weighted average of the distributions of the measured and forecast states. The PF on the other hand does not use a state analysis step, but rather assigns a likelihood to each particle. This likelihood is a dimensionless scalar which measures in a probabilistic sense the distance between the measured and forecasted state variables. The state PDF at each time is then constructed via the likelihoods (normalized importance weights) of the particles. Resampling is required to rejuvenate the ensemble, but this step is rather inefficient compared to the state analysis step of the EnKF as the measured states are only used indirectly in the PF via calculation of the likelihood. What is more, a single resampling step in RRPF or PMCMC does not guarantee a good approximation of the actual state PDF, as the particles' forecasted states may be systematically biased. Consequently, the PF may need a very large ensemble and/or many resampling steps to characterize properly the state PDF. On the contrary, the state analysis step of the EnKF resurrects rapidly a biased ensemble by migrating the members' forecasted states in closer vicinity of their measured values. This crucial difference between the EnKF and PF is the result of their dichotomous design, as is also evident from our mathematical notation. The EnKF estimates separately at each time the state PDF via equation (5), whereas the PF is designed to estimate the posterior distribution of the entire state trajectory via the recursion of equation (18). This latter task is much more difficult in practice, and requires use of the laws of probability to ensure that each particles' state trajectory constitutes a plausible realization from the transition density, $\mathcal{M}(\{\mathbf{x}_t^i\}|\{\mathbf{x}_{t-1}^i\})$, juxtaposed by the distribution of the model errors. This latter requirement of plausibility renders impossible the use of an analysis step in the PF (such as EnKF), as the resulting state updates may violate the statistics of the transition density and model error distribution and jeopardize the realism of each particle's state trajectory. Therefore, the PF requires a proper resampling method that takes into explicit account the statistical properties of the state transition density and model error distribution to replace bad particles and ensure an exact characterization of the evolving state PDF.

## 4. Case study

### 4.1. The Rollesbroich Experimental Site

We apply the four data assimilation approach to characterize soil moisture dynamics of the 27 ha Rollesbroich experimental test site (50°37'27"N, 6°18'17"E) in Germany. This site is located in the Eifel hills and ranges in elevation between 474 and 518 m with mean slope of 1.63°. The watershed constitutes a sub-basin of the TERENO Rur experimental catchment (Bogena et al., 2010; Qu et al., 2014) and consists of grassland with a soil texture that is predominantly silty loam. The mean annual air temperature and precipitation are 7.7 °C and 1033 mm, respectively. An eddy covariance tower (50°37'19"N, 6°18'15"E, elevation 514.7 m) and a soil moisture and soil temperature sensor network (with measurements at 5, 20 and 50 cm depth) have been installed (amongst others) at the Rollesbroich site. Water content data are measured at 41 different locations (see Figure 1) using SPADE soil moisture probes (sceme.de GmbH i.G., Horn-Bad Meinberg, Germany) (Hübner et al., 2009) installed at 5 cm, 20 cm and 50 cm depth along a vertical profile. The SPADE probe is a ring oscillator and the frequency of the oscillator is a function of the dielectric permittivity of the surrounding medium, which depends strongly on local soil water content because of the high relative permittivity of water ($\approx$ 80) as compared to mineral soil solids ($\approx$ 2-9), and air ($\approx$ 1). The SPADE probe was calibrated following the procedure outlined in (Qu et al., 2014). The soil moisture measurements are subject to several sources of error. This includes an inadequate contact of the sensors with the surrounding soil, and structural imperfections of the equations which relate the sensor response to the dielectric permittivity, and this permittivity to soil moisture.

The atmospheric LSM forcing data in this study were measured at the eddy covariance tower and include hourly measurements of air temperature, air pressure, relative humidity, wind speed, and incoming shortwave and longwave radiation. Precipitation was measured by a tipping bucket located in close proximity of the eddy covariance station. Soil texture was determined using 273 soil samples, taken from three different depths, ranging between 5 and 11 cm, 11 and 35 cm, and 35 to 65 cm. The sample locations coincided exactly with the location of the SoilNet sensors. The soil textural composition, organic carbon content, and bulk density were determined for each sample using standard laboratory experiments. These values were averaged to obtain mean values for the listed depths. Soil hydraulic parameters were then estimated for each of these three measurement depths from pedotransfer functions using as input data the basic soil measurements.

In this work, we conveniently assume the soil-land-surface domain of the Rollesbroich site to be homogeneous and characterized by areal average values of soil moisture content at 5, 20 and 50 cm depth. In other words, we consider only vertical variations in soil water storage. Common LSM data assimilation experiments published in the literature usually involve application to much larger spatial scales, especially when remote sensing data are used. Hence, it is important to evaluate the LSM performance for a site where heterogeneities are neglected. Qu et al. (2014) investigated the geostatistical properties of the soils of the Rollesbroich test site. This work demonstrated a rather small spatial variability of the soil texture. This does not suggest, however that we can ignore spatial variations in the measured soil moisture values. Indeed, the standard deviations of soil moisture vary between 0.04 and 0.07 $cm^3/cm^3$ depending on the actual soil layer. This spatial heterogeneity of the soil moisture data documents variability in the soil hydraulic properties, and complicates the application and upscaling of LSMs.

## 4.2. Numerical Experiments

A total of $N = 100$ ensemble members (particles) were used in all our data assimilation experiments. The period from January 1, 2011 to February 29, 2012 was used to spin-up VIC-3L and CLM using measured hourly forcing data. The subsequent period between March 1, 2012 and July 31, 2012 served as our "calibration period" during which the daily soil moisture observations at the three measurement depths were used to update the LSM state variables and possibly also its parameter values. The following 5-months from August 1, 2012 to December 31, 2012 were used as an independent evaluation period. During this last period, we did not update the states and set the parameters to their "optimized" values derived from the calibration period. Soil moisture assimilation was initiated in March 2012 as the SPADE water content sensors were deemed unreliable (at least in February) in the preceding winter season due to soil freezing. We terminated our numerical experiments at the end of December 2012, as a large number of sensors seemed to be malfunctioning in subsequent readings which could impact too much the mean soil moisture values.

Soil moisture contents measured at 5 cm, 20 cm and 50 cm depth were assimilated jointly. The three (default) soil layers in VIC-3L (0-10 cm, 10-30 cm, and 30-70 cm) were synchronized to match the three measurement depths. Soil parameters were defined separately for all individual layers, measured or not. In CLM, we used ten (default) soil layers with increasing thickness downwards (see Table 2). The 5, 20 and 50 cm measurement depths correspond to the third, fifth and the sixth layer in CLM. Spatial relationships (covariance matrices) between the soil parameters of the measured layers and their values of the unmeasured layers were used in the EnKF to update the parameterization of layers 1, 2, 4, 7, 8, 9 and 10. A slightly different approach was followed in RRPF and PMCMC, in which the soil parameters of the unmeasured moisture layers in CLM were updated to their weighted-average values of the resampled particles using the vector of normalized importance weights.

The measurement errors of the soil moisture observations are assumed to be zero-mean Gaussian with standard deviation, $\sigma = 0.02$ m³/m³. This results in $\mathbf{R} = 4 \cdot 10^{-4}\mathbf{I}_m$ in equations (4) and (17), respectively. We admit that 0.02m³/m³ is clearly larger than the uncertainty of the mean soil moisture content averaged over the 41 values. A larger observation error alleviates potential problems with filter inbreeding. Also, we account crudely for errors in LSM model formulation via parameter uncertainty and the use of a stochastic description of the precipitation record of the Rollesbroich site (discussed next). In other words, the $k \times 1$ process noise vector, $\mathbf{w}_t$, in equation (2) consists of zeros. However, we agree that it can be expected that we have other model structural errors, for example in relation to the representation of photosynthesis.

The hyetograph of each ensemble member is derived by multiplying the measured hourly precipitation rates of the tipping bucket with multipliers drawn from a unit-mean normal distribution with standard deviation of 0.10. This is equivalent to a heteroscedastic error of 10% of the observed precipitation (Hodgkinson et al., 2004). Forcing variables which govern evapotranspiration (incoming shortwave and longwave radiation, air temperature, relative humidity, and wind speed) were not corrupted.

The initial values of the VIC-3L and CLM parameters are sampled at random using a simple two-step procedure. This approach honours soil textural data and is consistent with related results published in the literature. First, we draw $N$ times from each marginal distribution listed in Table 1 under the column "perturbation". These distributions originate from Han et al. (2014) for CLM, and Demaria et al. (2007) and Troy et al. (2008) in case of VIC-3L. This results in a $N \times d$ matrix of perturbations for VIC-3L and CLM, respectively. We then create

the initial $N \times d$ parameter ensemble of VIC-3L and CLM by adding each perturbation matrix to a deterministic vector of "best-guess" parameter values for each model. This initial parameter ensemble is the same for all the assimilation methods. For CLM, this best-guess vector is simply equivalent to the areal-averaged sand, clay, and

organic matter fraction of each of the ten soil layers, respectively. In case of VIC-3L, we guess that $\beta = 15$ (all layers), $b = 0.2$, and $D_m = 13$ (mm/d), and derive the value of $\log_{10}k_s$ ($\log_{10}$(m/s)) of all three soil layers from the measured mean areal sand fraction at each of those depths. The best-guess parameter values of VIC-3L and CLM and their respective marginal distributions are jointly also referred to hereafter as *prior parameter distribution*. We want to compare EnKF and PF starting from the same prior distribution in order to make a more

meaningful comparison. EnKF assumes a Gaussian distribution, but the PF not. We believe that assuming an initial uniform distribution is a neutral assumption good for comparing EnKF and PF.

One may debate our best-guess parameter values of VIC-3L and CLM and their respective marginal distributions. Nevertheless, the prior parameter distribution used herein introduces more than sufficient dispersion in the best-guess parameter values to rapidly overcome a possibly deficient initial model

parameterization. Note, that the prior uncertainty of the two texture parameters (sand and clay fraction) in CLM is much larger than their spread derived from the texture measurements of each soil layer. This inflation of the prior distribution is done purposely to account indirectly for the epistemic uncertainty of the pedotransfer functions that are used to predict the soil hydraulic parameters. Indeed, the prior parameter uncertainty of the sand and clay fraction should be large enough to guarantee a sufficient soil moisture spread of the ensemble,

which is of crucial importance for an adequate performance of the different data assimilation methods.

Figure 2 shows the measured records of daily precipitation and daily air temperature for the 10 month measurement period used herein. The measurement period is rather wet with several intensive precipitation events during the summer. For example, notice the event on the 27th of July in 2012 in which 31 mm of precipitation fell in just one hour. Our experience suggests that such extreme rainfall events corrupt the

parameter estimates, in large part due to an inadequate description and/or characterization of surface runoff. What is more, the correlation between the hydraulic parameters of the different layers of our soil domain and the moisture state deteriorates rapidly close to saturation. Therefore, on days with rainfall in excess of 20 mm we resort to state estimation only, and proceed with this the next two consecutive days to give VIC-3 and CLM sufficient opportunity to remove, via deficient surface transport or state updating, the excess water. On the third

day after each 20 mm+ precipitation event, we resume joint LSM state and parameter estimation.

To evaluate joint state-parameter estimation algorithms for the two LSMs and the four different data assimilation algorithms, we carried out the following three numerical experiments for VIC-3L and CLM (see also Table 3):
(1) Open loop simulation. We evaluate the LSMs from March 1, 2012 to December 31, 2012 with time-invariant parameters via Monte Carlo simulation using a large number of draws from the prior parameter distribution

summarized in Table 1 and section 4.2.
(2) State updating with EnKF. The soil moisture state variables were updated during the five-month calibration period using the SPADE moisture content measurements. In theory, soil moisture assimilation should improve our estimates of the initial states of the evaluation period. We posit that this enhanced state-value characterization should improve the accuracy of the LSM simulated (predicted) soil moisture values during the

first few days/weeks of the evaluation period, after which the model performance deteriorates rapidly over time in the absence of recursive state adjustments.

(3) Joint state-parameter estimation using RRPF, PMCMC, and EnKF with state augmentation and dual estimation. The soil moisture state variables and model parameters are estimated during the five-month calibration period using the SPADE soil moisture measurements. The parameter values and state variables at the end of the calibration data period are used for the evaluation period.

### 4.3. Summary Statistics

We used the Nash-Sutcliffe model efficiency (NSE) and the Root Mean Square Error (RMSE) to evaluate the quality-of-fit of the VIC-3L and CLM predicted (simulated) soil moisture values during the calibration (assimilation) and evaluation period. These two metrics are computed separately for the 5, 20, and 50 cm measurement depths as follows:

$$\text{NSE}_i = 1 - \frac{\sum_{t=1}^{n}(\tilde{y}_{i,t} - \bar{y}_{i,t})^2}{\sum_{t=1}^{n}(\tilde{y}_{i,t} - \frac{1}{n}\sum_{t=1}^{n}\tilde{y}_{i,t})^2} \quad ; \quad \text{RMSE}_i = \sqrt{\frac{1}{n}\sum_{t=1}^{n}(\tilde{y}_{i,t} - \bar{y}_{i,t})^2}, \tag{24}$$

where $\tilde{y}_{i,t}$ and $\bar{y}_{i,t}$ denote the measured and ensemble mean predicted soil moisture contents at time $t$, the subscript $i$ constitutes an index for measurement depth, $i = \{1,\dots,3\}$, and $t = \{1,\dots,n\}$. The $3 \times 1$ vector of ensemble mean predicted moisture contents, $\bar{y}_t$, is simply equivalent to the mean of the VIC-3L or CLM forecasted state variables at these respective measurement depths. Larger values of the NSE and smaller values of the RMSE are preferred as they indicate a better LSM performance. In the absence of reliable information about the soil hydraulic properties of the different layers, the soil moisture observations were the only data available to evaluate the results of VIC-3L and CLM and each data assimilation method.

### 5. Results

In this section we present the results of our numerical experiments. We first discuss our findings for VIC-3L followed by the results of CLM. Section 6 proceeds with a discussion of the main findings.

### 5.1. VIC-3L

Figure 3 displays the observed (blue dots) and VIC-3L predicted soil moisture values (solid lines) at (A) 5, (B) 20, and (C) 50 cm depths using PMCMC (black), RRPF (red), EnKF-AUG (green), and EnKF-DUAL (cyan). As the Rollesbroich test site experiences a yearly average precipitation of more than about 1000 mm it is not surprise that the upper soil layer at 5 cm is rather wet with volumetric soil moisture contents that vary dynamically between 0.3 and 0.5 cm$^3$/cm$^3$ in response to atmospheric forcing. This is especially true during the summer months (week $12 - 22$) and explained by a rapid succession of rainfall and drying events. The larger porosity values of the surface layer explain the relatively high soil moisture contents of the 5 cm measurement depth. The storage time series of the deeper soil layers at 20 and 50 cm depth exhibit a rather negligible temporal variation with soil moisture values that range between 0.3-0.4 cm$^3$/cm$^3$ and show a damped and lagged response to rainfall. Note that the soil water storage of the deepest layer increases steadily during the year. This implies a drainage flux from the top soil to the aquifer (and drainage channels).

The different data assimilation methods demonstrate a rather similar performance with VIC-3L predicted moisture contents that track reasonably well the three different layers. Note, however that RRPF does not reproduce well the measured data at 50cm depth in the period from March (week 1) to June (week 17). This might be caused by filter inbreeding of the states, and will be discussed later (see also Fig. 9b). Nevertheless,

RRPF recovers the observed soil moisture data in week 18. Although difficult to see, the EnKF produces the best results at 50 cm depth (state augmentation and dual estimation).

Table 4 summarizes the NSE and RMSE values of PMCMC, RRPF, EnKF-DUAL and EnKF-AUG for the calibration (assimilation) period. We also list the performance of VIC-3L without data assimilation (OpenLoop) using the mean soil moisture time series of many different realizations of the prior parameter distribution, and include RMSE and NSE values of the EnKF for state estimation only (noParamUpdate) using VIC-3L parameterizations drawn randomly from its prior parameter distribution. The open loop deviates most from the

measured values with RMSE values of 0.036, 0.037 and 0.129 cm$^3$/cm$^3$ for the 5, 20, and 50 cm measurement depths. The different data assimilation methods improve significantly the quality of fit of VIC-3L compared to the open loop run. EnKF-AUG and EnKF-DUAL exhibit an almost identical performance with similar NSE and RMSE values. The particle filters, RRPF and PMCMC demonstrate comparable results for the 5 and 20 cm depth, but exhibit somewhat inferior performance compared to EnKF-AUG and EnKF-DUAL for the 50 cm

layer. The Table confirms our previous finding that the PF exhibits difficulties to track the soil moisture data of the deepest measurement layer. Indeed, the RMSE value of 0.088 of the PF for this layer is much larger than its counterparts of 0.021, 0.014 and 0.016 derived from PMCMC, EnKF-AUG and EnKF-DUAL, respectively. Perhaps surprisingly, but the best performance of VIC-3L is obtained for state estimation only (noParamUpdate) using model parameterizations drawn randomly from the prior parameter distribution. We posit that the nonlinear

relationship between states and parameters may introduce inconsistencies in PMCMC, RRPF, EnKF-AUG and EnKF-DUAL which jointly estimate VIC-3L states and parameters. Overall, the EnKF gives somewhat better results than the PF, particularly for the deepest measurement layer, and PMCMC exhibits a better performance than RRPF.

Figure 4 presents traceplots of the VIC-3L parameters during the 5-month calibration period using the PMCMC

(black), PF (red), EnKF-AUG (green), and EnKF-DUAL (cyan) data assimilation methods. We display the ensemble mean saturated hydraulic conductivity ($\log_{10} k_s$ in m/s) at (A) 5 cm, (B) 20 cm, and (C) 50 cm depth, (D) b, $\beta$ at (E) 5 cm, (F) 20 cm, and (G) 50 cm depth, and (H) the maximum baseflow velocity, $D_m$ in mm/day. In general, the different data assimilation methods result in somewhat similar trajectories of the ensemble mean parameter values during the calibration period. In particular, the parameter traceplots of EnKF-AUG and EnKF-

DUAL appear almost identical, with the exception of parameter b and $\beta$ at 50 cm depth. Note that the parameters of the surface layer exhibit most dynamics in response to atmospheric forcing. This is largely due to the trajectories of PMCMC which exhibit significant temporal dynamics. This is not surprising, and a consequence of the MCMC resampling step that is used to rejuvenate the parameter samples (e.g. Vrugt et al., 2013). In the first place, the DREAM-type proposal distribution that is used to create candidate particles allows for relatively

large moves in the parameter space. Second, only a small LSM trajectory between two successive soil moisture observations is used to determine the acceptance probability of each candidate particle. With such a short (re)-simulation period, insensitive parameters are allowed to transition to very different values, as they do not affect the model output between the two observations, and thus likelihood of a candidate particle. The use of a larger historical simulation period (going back further in time) would better constrain the VIC-3L parameters, but also

increase significantly the computational burden of resampling. Nonetheless, the ensemble mean VIC-3L parameter values of the different data assimilation methods are remarkably similar at the end of the calibration period, after assimilating the soil moisture observations of week 22. The exception to this is parameter $b$ whose

trajectories differ most with values at the end of the calibration period that range between values of 0.11 for RRPF and 0.25 for EnKF-DUAL. Finally, parameter $D_m$ converges systematically to values of 1 - 2 mm/day but at a different rate for the data assimilation methods. The EnKF-AUG, EnKF-DUAL and PMCMC methods need just a few soil moisture observations to determine the value of $D_m$, whereas RRPF converges at a much slower pace. This might explain the rather inferior performance of RRPF for the 50 cm measurement depth during a substantial part of the assimilation period.

To provide a better understanding of the ensemble spread of the VIC-3L parameters, please consider Figure 5 which presents traceplots of the sampled $\log_{10}k_s$ (left column) and $\beta$ (right column) values at the 20 cm measurement depth for the $N = 100$ members. Results are presented in order of (A-B) PMCMC (gray), (C-D) RRPF (red), (E-F) EnKF-AUG (green) and (G-H) EnKF-DUAL (cyan) and the ensemble mean is indicated with the solid black line. The ensemble members cover a relatively large part of the prior distribution of both parameters, with the exception of RRPF which seems to underestimate the actual uncertainty of $\log_{10}k_s$ and $\beta$. This is an artefact of equation (13) which discourages large parameter adjustments with small $s$. Nevertheless, note that the ensemble mean of the parameters is rather unaffected by assimilation of the soil moisture data, except for the small increase of $\log_{10}k_s$ and $\beta$ late April due to increased precipitation in the following months (see also Fig. 2).

Figure 6 displays VIC-3L simulated soil moisture time series for the independent 5-month evaluation period at (A) 5, (B) 20, and (C) 50 cm depths using initial states and parameter values derived from PMCMC (black), PF (red), EnKF-AUG (green), and EnKF-DUAL (cyan). The observed soil moisture values are separately indicated with the solid blue dots. The water content simulations of VIC-3L are hardly distinguishable, except for the deepest soil layer at 50 cm depth. Apparently, it does not matter which data assimilation method is used to estimate the VIC-3L parameter values and initial states of the evaluation period. VIC-3L tracks very well the soil moisture data at 20 cm depth, but does not do a particularly good job in describing water content dynamics at 5 and 50 cm depth. In particular, the model systematically underestimates the observed storage of the bottom soil layer between weeks 25-36. This might be a consequence of the use of a fixed lower boundary condition (no connection with underlying aquifer) and/or the relatively simple baseflow parameterization. Although not further shown herein, a separate VIC-3L run using state estimation only (noParamUpdate) produces similar results after a few days to an open loop simulation.

We summarize in Table 5 the NSE and RMSE values of PMCMC, RRPF, EnKF-DUAL and EnKF-AUG during the 5-month evaluation period. We also list the performance of VIC-3L without data assimilation (OpenLoop) using the mean soil moisture time series of many different realizations of the prior parameter distribution, and include RMSE and NSE values of the EnKF for state estimation only (noParamUpdate) using VIC-3L parameterizations drawn randomly from its prior parameter distribution. In general, the RMSE values of the evaluation period are much higher than their counterparts of the assimilation period, and noParamUpdate produces RMSE values similar to that of an open loop simulation. VIC-3L parameter estimation is productive, as it substantially reduces the RMSE values of 20 and 50 cm measurement depths compared to a model run with state estimation only (noParamUpdate) and parameters drawn randomly from their prior distribution. More specifically, the PMCMC, RRPF, EnKF-AUG and EnKF-DUAL show a RMSE improvement of about 54% and 42% for the second and third measurement depth compared to OpenLoop and noParamUpdate. The NSE values

of VIC-3L for the 50 cm depth are negative for all six methods, conclusively demonstrating an inferior performance of the model for this soil layer.

We now investigate in more detail the effect of MCMC resampling with the PF as Fig. 4 has demonstrated that PMCMC produces rather dynamic trajectories of the sampled parameter values. Nevertheless, the parameters converge to stable values at the end of the assimilation period. This suggests that the choice of the length of the calibration period is crucially important in determining the performance of PMCMC during the evaluation period. To investigate this in more detail we use June 11, June 30, July 20, and July 31, 2012 as end dates of the PMCMC calibration period and verify VIC-3L performance for the same 5-month evaluation period. The different end dates are conveniently referred to as PMCMC_0611, PMCMC_0630, PMCMC_0720 and PMCMC_0731 in Figure 7. The simulated soil moisture trajectories of PMCMC_0630, PMCMC_0720 and PMCMC_0731 are in excellent agreement, but deviate from PMCMC_0611. Thus, a 4-month calibration period would have led to the same results of PMCMC.

The effect of initial uncertainties on the performance of EnKF with the ensemble inflation method is also tested with the VIC-3L model. Table 6 compares the RMSE values of EnKF-AUG and EnKF-DUAL for the calibration and evaluation period using heteroscedastic precipitation data errors equivalent to 10% (default) and 20% of their measured hourly rates plotted in Figure 2. We list separate RMSE values for each soil moisture measurement depth. In short, the results are equivalent for both EnKF implementations.

Next, we evaluate the effect of the choice of the scaling factor $s$ in RRPF on VIC-3L output. This scalar plays a crucial role in the resampling of the parameters in the PF. If $s$ is taken too large, the resampling step will introduce parameter drift and corrupt the approximation of $p(\mathbf{X}_{1:t}|\widetilde{\mathbf{Y}}_{1:t})$ and $p(\mathbf{x}_t|\widetilde{\mathbf{Y}}_{1:t})$. On the contrary, if $s$ is too small, then the resampled parameters exhibit insufficient dispersion, and underestimate the actual parameter uncertainty. In the absence of theoretical convergence proofs and clear guidelines on the selection of $s$, the RRPF cannot estimate exactly the posterior state and parameter PDF (Vrugt et al., 2013; Yan et al., 2015). Previous applications of RRPF have suggested a value of $s = 0.01$ (DeChant and Moradkhani, 2012; Plaza et al., 2012), but thus far we have used $s = 0.1$ to avoid sample impoverishment. Table 7 lists RMSE values of VIC-3L for the 5, 20, and 50 cm measurement depth for the calibration and evaluation period using RRPF with $s = 0.01$, $s = 0.1$, and $s = 0.5$, respectively. These three runs are coined RRPF_0.01, RRPF_0.1 and RRPF_0.5, respectively. These results demonstrate that a value of $s = 0.5$ significantly enhances the performance of RRPF during the calibration period. The RMSE values are reduced from 0.025, 0.012, and 0.113 to 0.015, 0.007, and 0.037 for the 5, 20 and 50 cm measurement depths. RRPF_0.5 also shows substantial improvements over RRPF_0.01 during the evaluation period. This improvement is most apparent for the 20 and 50 cm soil depths with RMSE values that have decreased from 0.025 and 0.119 to 0.020 and 0.071, respectively. These results are on par with our default setting of $s = 0.1$ in RRPF. These findings provide evidence for our claim that the scaling factor $s$ plays a crucial role in RRPF. What is more, it provides support for our conclusion in Fig. 5 that RRPF underestimates the actual uncertainty of $\log_{10}k_s$ and $\beta$. Larger values of $s$ will increase the parameter spread, which in turn will enhance the uncertainty among the particles' forecasted states. This makes it easier for RRPF to track the observed soil moisture data during the calibration period.

Figure 8 displays traceplots of the sampled $N = 100$ trajectories of the saturated hydraulic conductivity ($\log_{10}k_s$ in m/s) at 50 cm depth (left column) and parameter $\beta$ (right column) of VIC-3L during the 5-month assimilation

period using (A-B) RRPF_0.01, (C-D) RRPF_0.1, and (E-F) RRPF_0.5. As expected, larger values of $s$ increase the spread of the sampled values of the VIC-3L parameters as evidenced by an increasingly larger particle coverage of the prior parameter distribution. This larger spread of the particles' parameter values also enhances the ability of RRPF to track properly the joint parameter and state PDF. Yet, larger values of $s$ have two important drawbacks. Not only can it obstruct parameter convergence (as evidenced in Fig. 8e), but also many of the resampled parameter values might be deemed nonbehavioral, enhancing considerably the chances of particle degeneration. To demonstrate this more explicitly, Figure 9 shows traceplots of the VIC-3L predicted soil moisture contents of the $N = 100$ particles at 50 cm depth using (A) RRPF_0.01, (B) RRPF_0.1, and (C) RRPF_0.5. The RRPF is excessively optimistic for $s = 0.01$ with a negligible uncertainty in the predicted soil moisture values between weeks 2-14. Note that in weeks 2-4 the ensemble has collapsed to a deterministic simulation (appears as single line). A similar result is observed for RRPF_0.1 but with enhanced uncertainty in soil moisture values for the second part of the calibration period. In PF_0.1 particle degeneration from March to June explains its bad performance from March to June in Fig. 3. The use of $s = 0.5$ enhances considerably the spread of the VIC-3L soil moisture predictions. Yet, the ensemble spread has become quite large from week 15 onwards. For these reasons, we are satisfied with our value of $s = 0.1$ in RRPF, although this decision is subjective and would require much testing via trial-and-error. This has stimulated Vrugt et al. (2013) to introduce a parameter resampling method which is properly rooted in statistical theory and uses laws of probability to rejuvenate the ensemble.

### 5.2. CLM

Figure 10 shows the observed (blue dots) and ensemble mean predicted soil moisture values by CLM (solid lines) at (A) 5, (B) 20, and (C) 50 cm depths during the assimilation period using PMCMC (black), PF (red), EnKF-AUG (green), and EnKF-DUAL (cyan). The most important results are as follows. First, the ensemble mean soil moisture time series of CLM exhibit a larger spread than VIC-3L depicted previously in Fig. 3. Second, the EnKF-AUG and EnKF-DUAL exhibit a superior performance with ensemble mean CLM simulations that track closely the observed soil moisture observations at each depth. Third, the moisture time series (and data) demonstrate most dynamics at the 5 cm depth in response to the variable atmospheric boundary conditions. Fourth, the worst performance is observed for RRPF, as evidenced by systematic deviations of this filter 's soil moisture predictions with the observed data between weeks 3-6 and 18-21 for the 5 cm depth, weeks 1-14 and weeks 18-21 for the 20 cm depth, and weeks 1-15 and 19-22 for the 50 cm measurement depth. Fourth, the initial soil moisture values of CLM at 50 cm depth appear positively biased with a distance of approximately 0.05 cm$^3$/cm$^3$ to the areal-mean value of the soil water contents measured by the SPADE sensors on 01-03-2012 (first day of week 1). A smaller bias of 0.03 cm$^3$/cm$^3$ is observed at the 20 cm depth. The ENKF-AUG and EnKF-DUAL methods need a few days to recover from this erroneous initialization.

Table 8 lists the NSE and RMSE values of PMCMC, RRPF, EnKF-DUAL and EnKF-AUG for the CLM calibration (assimilation) period. We also list the performance of CLM without data assimilation (OpenLoop) using the mean soil moisture time series of many different realizations of the prior parameter distribution, and list in column with header "noParamUpdate" the RMSE and NSE values of the EnKF using state estimation only with CLM parameterizations drawn randomly from the prior parameter distribution. These results demonstrate that soil moisture assimilation enhances considerably the ability of CLM to predict the observed data. Compared

to open loop CLM simulation, the RMSE is reduced from 0.051, 0.031 and 0.069 to values of about 0.020, 0.012, and 0.016 (average) for the different data assimilation methods, respectively. Yet, the RMSE and NSE values of a CLM run with state estimation only (noParamUpdate) appear as good as those derived from joint parameter and state estimation using PMCMC, RRPF, EnkF-AUG and EnKF-DUAL. Overall, the best performance is observed for EnKF-AUG and EnKF-DUAL followed by PMCMC and RRPF.

We proceed in Figure 11 with traceplots of the $N = 100$ sampled trajectories of the saturated hydraulic conductivity ($\log_{10} k_s$ in m/s) at 50 cm depth (left column) and soil hydraulic parameter B at 50 cm depth (right column) during the 5-month assimilation period using (A-B) PMCMC (C-D) RRPF, (E-F) EnKF-AUG, and (G-H) EnKF-DUAL. The evolution of the ensemble mean $\log_{10} k_s$ and B values is separately indicated with the solid black line. The largest spread of the ensemble members is observed for EnKF-AUG and EnKF-DUAL and explained by the inflation method of equation (13) which inherits and sustains the prior parameter uncertainty.

The RRPF sampled trajectories of $\log_{10} k_s$ and $B$ exhibit a rather small uncertainty with PDF ś of these two parameters that appear well defined at all measurement times. This might explain the inferior performance of RRPF as detailed previously in Table 8. Overall, the two CLM parameters do not exhibit large temporal changes and converge to stable values in the last few weeks of the calibration period.

Figure 12 displays the observed (blue dots) and ensemble mean predicted soil moisture values by CLM (solid lines) at (A) 5, (B) 20, and (C) 50 cm depths during the evaluation period using PMCMC (black), PF (red), EnKF-AUG (green), and EnKF-DUAL (cyan). The soil moisture time series of the different data assimilation methods appear rather similar with largest differences observed at the 50 cm depth. In general, the PMCMC, RRPF, EnKF-AUG and EnKF-DUAL methods do not do a particularly good job in tracking the soil moisture

observations of the top soil layer. Indeed, the CLM soil moisture predictions derived from the different data assimilations are systematically biased, either underestimating (weeks 35-41 and 43-44) or overestimating (weeks 24-31 and 42) the observed soil moisture data during large parts of the evaluation data set. CLM much better tracks the soil moisture data of the 20 and 50 cm depth.

    Finally, Table 9 presents the NSE and RMSE values of PMCMC, RRPF, EnKF-AUG and EnKF-DUAL during

the 5-month evaluation period. We also list the performance of VIC-3L without data assimilation (OpenLoop) using the mean soil moisture time series derived from many different realizations of the prior parameter distribution, and display NSE and RMSE values of the EnFK using state estimation only (noParamUpdate) with CLM parameterizations drawn randomly from the prior parameter distribution. The results of this Table are in agreement with our findings for VIC-3L. Indeed, the RMSE values of the evaluation period are much higher than

their counterparts of the assimilation period. This is particularly evident for the 5 cm measurement depth where RMSE values have increased from 0.017-0.027 to 0.054-0.058. The deeper measurement depths do not appear to be as much affected, consistent with our findings from Fig. 12. The results also highlight the importance of joint CLM parameter and state estimation as state estimation alone (column noParamUpdate) results in significantly larger RMSE values during the evaluation period. This is most evident for the 50 cm measurement depth, where

the RMSE value of 0.050 of noParamUpdate is much larger than its value of 0.016-0.025 derived from PMCMC, RRPF, EnKF-AUG and EnKF-DUAL. Altogether, RRPF achieves the worst performance of all four parameter-state estimation methods during the evaluation period. PMCMC, EnKF-AUG and EnKF-DUAL provide rather similar RMSE and NSE values.

**6. Discussion**

In this study, we have evaluated the usefulness and applicability of four different data assimilation methods for joint parameter and state estimation of the VIC-3L and CLM land surface models using a 5-month calibration (assimilation) data set of distributed SPADE soil moisture measurements at 5, 20 and 50 cm depth in the Rollesbroich test site in the Eifel mountain range in western Germany. We used the EnKF with state augmentation or dual estimation, respectively, and the PF with a simple, statistically deficient, or more

sophisticated, MCMC-based parameter resampling method. The "calibrated" LSM models were tested using water content data from a 5-month evaluation period. The uniqueness of the present work resides in the application of these four joint or dual parameter and state estimation methods to real-world data.

Our results demonstrated that joint inference of the VIC-3L and CLM soil parameters improved considerably soil moisture characterization during the evaluation period compared to the mean water content predictions of an

open loop run derived via averaging of simulations of many different realizations drawn randomly from the prior parameter distribution. This is particularly true for CLM, the two deeper soil layers, and the EnKF-AUG and EnKF-DUAL methods (but followed closely by PMCMC). Despite this improvement in model performance over an open-loop simulation, VIC-3L and CLM do not adequately characterize soil moisture dynamics of the top layer (5 cm measurement depth) during the evaluation period (RMSE values of about 0.05 cm$^3$/cm$^3$). We posit

that these two models do not characterize adequately processes such as water infiltration, soil evaporation, and/or root water uptake (transpiration), which govern rapid variations in soil moisture storage in the top soil. VIC-3L also appeared deficient at 50 cm depth during the evaluation period with RMSE values of about 0.07 cm$^3$/cm$^3$ which are much larger than their counterparts of approximately 0.02 cm$^3$/cm$^3$ derived from CLM. These results favour the use of CLM which uses a more physics-based description of soil water movement, storage, and

associated hydrological fluxes at the Rollesbroich site.

The improvement in quality-of-fit of the VIC-3L and CLM models compared to an open-loop run does not necessarily imply that the estimated parameter values of VIC-3L and CLM characterize better the hydraulic properties and maximum baseflow velocity of the soils of the Rollesbroich experimental test site. Assimilation studies with synthetically generated data help to ascertain whether the model parameters converge properly to

their "true" values, yet this is difficult to confirm with real-world measurements. State estimation will, without doubt, help reduce the impact of epistemic errors and systematic biases of LSM input and forcing data on parameter inference during the assimilation period (e.g. Vrugt et al., 2005). But the calibrated parameter values derived with state estimation do not necessarily guarantee a consistent and adequate model performance during an independent evaluation period without state estimation. Indeed, without assimilation the simulated states may

diverge from their "measured" values and deteriorate model performance in an evaluation period. This begs the question which parameter values we should use to predict future system behaviour outside an assimilation period? Should we use parameter estimates derived with state estimation or should we use their values derived via batch calibration (optimization) without recursive adjustments to the state variables? This dilemma is illustrated further in Vrugt et al. (2006) by modelling of a subsurface tracer test using data from Yucca

Mountain, Nevada, USA. We conclude that the enhanced performance of VIC-3L and CLM during the evaluation period compared to our open-loop simulation is due to improved estimates of the initial states and the soil parameters.

In our implementation of the EnKF and PF, the VIC-3L and CLM parameters were assumed to be time-variant and their values updated jointly with the model states at each assimilation time step. The 5-month calibration period we used herein involves several large precipitation events, and as a consequence, the soil profile is rather wet. The resulting parameter estimates might therefore not be representative for dry periods with much lower moisture values of the soil profile. What is more, the assumption of spatial homogeneity might not characterize adequately the distributed soil properties of the Rollesbroich site and induce temporal variability in the VIC-3L and CLM parameters. Bias in model input and measurement errors of the forcing data also contribute to the temporal fluctuations of the estimated parameter values. These temporal parameter variations are meaningful in some cases as they can help diagnose structural model inadequacies and/or biases in model input and forcing data. Kurtz et al. (2012) successfully estimated a temporally-variant parameter with the EnKF, but these authors concluded that the algorithm needs a considerable spin-up period to "warm-up" to new parameter values. Vrugt et al. (2013) found considerable temporal non-stationarity in the parameters estimated by PMCMC as a result of the small time period used to calculate the acceptance probability of candidate particles. This finding is in agreement with the results of PMCMC in our paper. Of course, we could have assumed time-invariant parameters via a method such as SODA, yet this would have enhanced significantly computational requirements. Fortunately, parameters estimated via our implementation of the EnKF exhibit asymptotic properties during the assimilation period (e.g. see Shi et al. (2015)). This is particularly true for highly sensitive parameters. An example of this was parameter $D_{\mathrm{m}}$ of VIC-3L which quickly converged to values of around $1 - 2$ mm after assimilating just a handful of soil moisture observations.

It is difficult to assess whether the inferred VIC-3L and CLM parameter values will do a good job at predicting soil moisture dynamics at the different measurement depths during a much longer evaluation period with wet and dry conditions. As the estimated parameters represent apparent properties of the Rollesbroich site, one may expect their calibrated values not to change too much over time. We would need additional soil moisture data and/or other type of measurements to corroborate this. Nevertheless, the apparent parameter values derived herein improve characterization of soil moisture dynamics at the Rollesbroich site compared to a separate state estimation run with VIC-3L and CLM using parameters drawn randomly from the prior distribution, or open loop simulation using the ensemble mean model output of a large cohort of parameter vectors drawn randomly from the prior parameter distribution (initial parameter ensemble).

The different data assimilation methods (EnKF-AUG, EnKF-DUAL, RRPF and PMCMC) led to a rather similar performance of VIC-3L during the calibration and evaluation period. The only exception to this was the anomalous RMSE value of RRPF at the 50 cm measurement depth during the calibration period. This was explained by the slow convergence of the maximum baseflow velocity in RRPF. Our results for VIC-3L further demonstrated that the results of EnKF-AUG and EnKF-DUAL were equivalent for a 10% and 20% rainfall error. Moreover, the use of a larger value of the scaling $s$ in RRPF reduced considerably the RMSE values of VIC-3L in the calibration data period, particularly at the 50 cm measurement depth, whereas model performance was hardly improved during the evaluation period.

For CLM, larger differences were observed in the performance of the different data assimilation methods. This larger disparity among the methods is explained by the considerably larger number of soil layers (ten) used by CLM. This increased significantly the dimensionality of the parameter estimation problem. The overall best results at the 5, 20 and 50 cm measurement depths were observed for EnKF-AUG and EnKF-DUAL with RMSE

values that were somewhat smaller than their counterparts derived from PMCMC. This was true for both the calibration and evaluation periods. The RRPF exhibited the worst performance, in part determined by the use of a relatively small ensemble of $N = 100$ particles. The superiority of the EnKF-AUG and EnKF-DUAL methods for CLM is consistent with our expectations articulated previously in Section 3.1. The analysis step of the EnKF makes it much easier for EnKF-AUG and EnKF-DUAL to track the measured soil moisture dynamics, thereby promoting convergence in high-dimensional state-parameter spaces. PF-based methods, on the contrary, deteriorate in robustness and efficiency with larger dimensionality of the state-parameter space as they lack a state analysis step and approximate the transient state-parameter PDF via the particles' likelihoods. This likelihood is only a low-dimensional summary statistic of the distance between the forecasted and measured values of the states. Resampling with MCMC via the likelihood thus becomes increasingly more difficult in high-dimensional state-parameter spaces. For CLM, the PMCMC method still achieves comparable results to EnKF-AUG and EnKF-DUAL as the dimensionality of the state-parameter PDF of this model is only somewhat larger than its counterpart of VIC-3L. Of course, the use of a larger ensemble size makes it easier to characterize the transient state-parameter PDF, but at the expense of a significantly increased CPU-cost. For PMCMC, multiple different MCMC resampling steps can also enhance significantly the particle ensemble by allowing each particle trajectory to improve its likelihood. Yet, this deteriorates significantly the efficiency of implementation as each candidate particle requires a separate model evaluation of VIC-3L or CLM to determine its likelihood. Thus, for LSMs with relatively few state variables and model parameters, we expect the EnKF and PF methods to achieve a comparable performance. For larger dimensional state-parameter spaces we would recommend EnKF-AUG and EnKF-DUAL, unless one can afford a very large number of particles.

Finally, our results demonstrated that the differences between the soil moisture simulations of VIC-3L and CLM are much larger than the discrepancies among the four data assimilation methods. Overall, CLM performed better than VIC-3L, especially at 50 cm measurement depth. Of course, we cannot generalize this finding to other sites, but VIC-3L's rather poor characterization of soil moisture dynamics at 50 cm depth (systematic underestimation during first 2-3 months) warrants investigation into the use of a variable water table depth in this model to account for interactions between the variably-saturated soil domain and the groundwater reservoir of the Rollesbroich site. CLM simulates such interactions and the resulting variations in the water table depth affect soil water movement in the unsaturated zone.

## 7. Conclusions

In this study, we have evaluated the usefulness and applicability of four different data assimilation methods for joint parameter and state estimation of the Variable Infiltration Capacity Model (VIC-3L) and the Community Land Model (CLM) using a 5-month calibration (assimilation) period (March – July, 2012) of areal-averaged SPADE soil moisture measurements at 5, 20 and 50 cm depth in the Rollesbroich experimental test site in the Eifel mountain range in western Germany. This watershed is part of TERENO observatories and extensively monitored since 2011 to catalogue long-term ecological, social and economic impact of global change at regional level. We used the EnKF with state augmentation or dual estimation, respectively, and the PF with a simple, statistically deficient, or more sophisticated, MCMC-based parameter resampling method. The "calibrated" LSM models were tested using SPADE water content measurements from a 5-month evaluation period (August – December, 2012). The performance of the four different state and parameter estimation methods appeared rather

similar during the evaluation period with a slightly better performance of the augmentation and dual estimation methods, but followed closely by PMCMC and then RRPF. The differences between the soil moisture simulations of VIC-3L and CLM are much larger than the discrepancies among the four data assimilation methods. Overall, the best performance was observed for CLM. The large systematic underestimation of water storage at 50 cm depth by VIC-3L during the first few months of the evaluation period questions, in part, the validity of its fixed lower boundary condition at the bottom of the modelled soil domain. This approach ignores the movement of water into and out of the groundwater reservoir of the Rollesbroich site. CLM simulates interactions of the modelled soil domain with the Rollesbroich aquifer via the use of a variable water depth at the lower boundary.

**Appendix A: Parametrization of the VIC-3L Model**

The integrated water balance in the VIC-3L can be written as follows:

$$\partial S / \partial t = P - T - E - Q_\mathrm{d} - Q_\mathrm{b}, \tag{A1}$$

where $S$ [L] is storage, $t$ [T] denotes time, $\partial S / \partial t$ [LT$^{-1}$] signifies the change in water storage, and $P, T, E, Q_\mathrm{d}$, and $Q_\mathrm{b}$ [LT$^{-1}$] represent fluxes of precipitation, canopy transpiration, soil evaporation, direct runoff, and baseflow, respectively. Bare soil evaporation, $E$, is calculated using the equation of Francini and Pacciani (1991). The canopy transpiration flux, $T$, is equivalent to the total uptake of water by plant roots in our soil profile and is estimated following Blondin (1991) and Ducoudre et al. (1993) using the bulk equation of Monteith (1963). In this "single-leaf" approach, the canopy resistance is assumed to be a function of the minimum canopy resistance and environmental variables (factors) such as photosynthetically active radiation, ambient temperature, vapour pressure deficit, and soil moisture content. We refer to Wigmosta et al. (1994) for a detailed discussion of these four limiting variables, including their mathematical description and parameterization used herein. When it rains the leaves become covered with a thin film of water and the transpiration flux is suppressed temporarily until the intercepted water has evaporated at the potential rate derived from the Penman-Monteith equation (Shuttleworth, 2007). To calculate foliage storage the maximum canopy water storage is set to a multiple of 0.2 of the leaf area index (Dickinson, 1984). Direct runoff, $Q_\mathrm{d}$, reduces the amount of rainfall that can infiltrate in the top soil during wet conditions, and is calculated using (Liang et al., 1996):

$$Q_\mathrm{d} = \frac{1}{\Delta t} \begin{cases} P\Delta t - \left( z_1(\phi_1 - \theta_1) + z_2(\phi_2 - \theta_2) - (z_1\phi_1 + z_2\phi_2)\left(1 - \frac{P\Delta t + I_0}{I_\mathrm{max}}\right)^{(1+b)} \right) & \text{if } P\Delta t \le (I_\mathrm{max} - I_0) \\ P\Delta t - \left( z_1(\phi_1 - \theta_1) + z_2(\phi_2 - \theta_2) \right) & \text{otherwise,} \end{cases} \tag{A2}$$

where the triples $\{\theta_1, \phi_1, z_1\}$ and $\{\theta_2, \phi_2, z_2\}$ signify the volumetric moisture content [L$^3$L$^{-3}$], porosity [-], and depth [L] of the top layer of the soil and the next or second layer immediately below it, respectively, $I_0$ [L] and $I_\mathrm{max}$ [L] denote the actual and maximum moisture capacity of the soil, respectively, $\Delta t$ [L] signifies the integration time step that is used to solve numerically Equation (A1), and $b$ [-] is an unknown shape parameter that measures the spatial variability of the soil moisture capacity. Note that the integration time step, $\Delta t$, is often missing from Equation (A2) in VIC-manuals or literature publications. This is consistent if rainfall, $P$, is expressed in units of depth, say mm, but invalid in conjunction with Equation (A1) which requires as input precipitation rates. If the integration time step is set equivalent to the time unit of the measured precipitation rates then $\Delta t = 1$. This approach, however can introduce large numerical errors, particularly if the soil is close to

saturation. The dimensionless parameter $b$ is usually determined via calibration by fitting VIC-3L against a historical record of soil moisture observations and/or flux data.

The direct runoff in Equation (A2) is not only a function of the water saturation of the first layer, but also depends on the moisture content of the second underlying soil layer. To be able to track adequately the large storage variations of the top soil observed in experimental data, the first layer of VIC-3L must be taken rather small. Consequently, this top layer will saturate quickly in response to rainfall as it exhibits a rather negligible water holding capacity. Hence, VIC-3L uses the available storage of the first and second layer to determine the

excess precipitation, which is set equivalent to $Q_d$. If the rainfall depth exceeds the available moisture capacity of the soil, $(I_{max} - I_0)$, then the excess precipitation is removed via surface runoff. Otherwise, if $P\Delta t \leq (I_{max} - I_0)$, then a large fraction of the rainfall will infiltrate depending on the soil's available storage and the spatial variability of the moisture capacity within the grid cell. The values of $I_0$ and $I_{max}$ are estimated from [Zhao, 1992]:

$$I_0 = I_{max}\left(1 - (1 - A_s)^{(1/b)}\right) \tag{A3}$$

$$I_{max} = (1 + b)(z_1\phi_1 + z_2\phi_2) \quad , \tag{A4}$$

where $A_s$ [-] is the areal fraction of the grid cell that is saturated (infiltration capacity equal to $I_{max}$):

$$A_s = 1 - \left(1 - \frac{z_1\theta_1 + z_2\theta_2}{\phi_1 + \phi_2}\right)^{(b/(1+b))} \tag{A5}$$

The baseflow, $Q_b$, originates from the bottom (third) soil layer and is calculated using the formulation of the

1035 Arno model (Franchini and Pacciani, 1991):

$$Q_b = \begin{cases} \frac{D_S D_m}{W_S \phi_3}\theta_3 & \text{if } 0 \leq \theta_3 \leq W_s\phi_3 \\ \frac{D_S D_m}{W_S \phi_3}\theta_3 + \left(D_m - \frac{D_S D_m}{W_S}\right)\left(\frac{\theta_3 - W_S\phi_3}{\phi_3 - W_S\phi_3}\right)^2 & \text{otherwise} \end{cases} , \tag{A6}$$

where $D_m$ [LT$^{-1}$] is the maximum baseflow velocity, and $D_S$ and $W_S$ are dimensionless fractions of $D_m$ and the porosity of the third layer, $\phi_3$, respectively. The baseflow flux is linearly dependent on the water content of the third layer if $\theta_3 \leq W_s\phi_3$, and increases nonlinearly with water storage of the third layer if $\theta_3 \geq W_s\phi_3$.

Now we have discussed the different fluxes from the soil domain simulated by VIC-3L we can now write differential equations of the moisture dynamics in the individual soil layers (see also Liang et al., 1996).

$$\frac{\partial\theta_1}{\partial t}z_1 = P + Q_{1,2} - Q_d - R_1 - E$$

$$\frac{\partial\theta_2}{\partial t}z_2 = Q_{2,3} - Q_{1,2} - R_2 \quad , \tag{A7}$$

$$\frac{\partial\theta_3}{\partial t}z_3 = -Q_{2,3} - R_3 - Q_b$$

where $Q_{i,i+1}$ [LT$^{-1}$] is the vertical flux of water between two adjacent soil layers $i$ and $i + 1$, $R_i$ [LT$^{-1}$] signifies the root water uptake of the $i$th layer, and $i = \{1,2,3\}$. Downward fluxes are negative to be consistent with

1045 convention used in soil hydrology. The canopy transpiration flux is equal to the total water uptake by the plant roots, thus $T = R_1 + R_2 + R_3$. All three soil layers contain roots and thus contribute to transpiration in our application of VIC-3L to the Rollesbroich site. The vertical flux of water between two adjacent soil layers is

assumed to be equivalent to the hydraulic conductivity of the upper layer. VIC-3L computes the hydraulic conductivity of each soil layer using the formulation of Brooks and Corey (1988):

$$Q_{i,i+1} = -k_{s,i} \left( \frac{\theta_i - \theta_{r,i}}{\phi_i - \theta_{r,i}} \right)^{\beta_i} \quad (i = 1,2), \tag{A8}$$

where $k_{s,i}$ [LT$^{-1}$] and $\theta_{r,i}$ [L$^3$L$^{-3}$] signify the saturated hydraulic conductivity and the residual volumetric moisture content of the $i$th soil layer, respectively. The minus sign at the right-hand-side of equation (A8) matches the direction of the flux. The dimensionless exponent $\beta_i$ should be larger than 3.0.

The use of three soil layers by VIC-3L makes it difficult to describe accurately the vertical moisture distribution in the vadose zone. Indeed, VIC-3L cannot distinguish between saturated and partially-saturated areas in a given soil layer. As a consequence, the baseflow flux, $Q_b$, is made up of water from the unsaturated zone and the groundwater (Liang et al., 1996; Liang et al., 2003). Liang et al. (2003) developed a new parameterization, which considers explicitly effects of surface and groundwater interactions on soil moisture, transpiration, soil evaporation, runoff and recharge. This parameterization, coined VIC-ground, enhanced considerably water storage in the lower soil layer compared to VIC-3L.

**Appendix B: Parametrization of the CLM Model**

This Appendix summarizes the main equations of CLM which are used to describe variably-saturated water flow in the soil domain of our experimental catchment. The model uses a water balance formulation similar to Equation (A1) of Appendix A to simulate moisture storage and movement in the soil of each grid-cell of the application domain of interest. Yet, CLM includes a more exhaustive description of all the different processes that determine the water storage of the land surface. This includes canopy water, surface water, snow water, soil water, soil ice and water stored in the unconfined aquifer. In addition to surface and subsurface runoff, CLM also considers runoff from glaciers, wetlands and lakes.

Fluxes, $F$ [ML$^{-2}$T$^{-1}$], of ground evaporation, interception evaporation, and vegetation transpiration are calculated by CLM using the following general expression (Schwinger et al., 2010; Oleson et al., 2013):

$$F = \frac{\rho_a}{r_a}(q - q_a), \tag{B1}$$

where $\rho_a$ [ML$^{-3}$] is the density of air, $r_a$ [TL$^{-1}$] signifies the aerodynamic resistance, $q$ [MM$^{-1}$] is the specific humidity of the soil pores (for soil evaporation) or canopy (for vegetation transpiration and interception evaporation) or the saturated specific humidity of snow or surface water, and $q_a$ [MM$^{-1}$] denotes the specific humidity at atmospheric level if ground evaporation is calculated, or the saturated specific humidity within the canopy if canopy evapotranspiration is calculated. The values of $r_a$, $q$ and $q_a$ are based on Monin-Obukhov similarity theory (Schwinger et al., 2010; Oleson et al., 2013).

We use 10 soil layers (see Table 2) in CLM to solve for the vertical storage and movement of water. Whenever the index $i$ is used we mean 'for all $i \in \{1, \dots, 10\}$'. The saturated hydraulic conductivity, $k_{s,i}$ [LT$^{-1}$], saturated volumetric moisture content, $\theta_{s,i}$ [L$^3$L$^{-3}$], thermal conductivity, $\lambda_i$ [WL$^{-1}$K$^{-1}$], soil matric head at saturation, $\psi_{s,i}$ [L], and Clapp-Hornberger exponent, $B_i$ [-], of each soil layer are derived from built-in pedotransfer functions.

These functions use as inputs textural data (Clapp and Hornberger, 1978; Cosby et al., 1984) and/or the organic matter fraction (Lawrence and Slater, 2008) of each soil layer as follows:

$$\psi_{s,i} = -10(1 - f_{om,i})10^{(1.88-0.0131f_{sd,i})} - 10.3f_{om,i} \qquad \text{[mm]} \quad (B2)$$

$$B_i = (1 - f_{om,i})(2.91 + 0.159f_{cl,i}) + 2.7f_{om,i} \qquad \text{[-]} \quad (B3)$$

$$\theta_{s,i} = (1 - f_{om,i})(0.489 - 0.00126f_{sd,i}) + 0.9f_{om,i}, \qquad \text{[-]} \quad (B4)$$

$$k_{s,i} = \left(1 - f_{p,i}\right)\left[\frac{1-f_{om,i}}{0.0070556 \cdot 10^{(-0.884+1.53f_{sd,i})}} + \frac{f_{om,i}-f_{p,i}}{k_{s,om}}\right]^{-1} + f_{p,i}k_{s,om} \qquad \text{[mm/s]} \quad (B5)$$

where $f_{sd,i}$ and $f_{cl,i}$, and $f_{om,i}$ signify the fractions of sand, clay and organic matter, respectively, $f_{p,i}$ [-], denotes the fraction of connected organic matter, $k_{s,om}$ [mm/s], is the saturated hydraulic conductivity of organic soils. If the organic matter fraction, $f_{om,i}$, is smaller than 0.5, then $f_{p,i} = 0$, otherwise $f_{p,i} = 0.5f_{om,i}(f_{om,i} - 0.5)^{-0.139}$.

Vertical flow in the unsaturated zone is governed by rainfall infiltration, surface and subsurface runoff, root water uptake (canopy transpiration), and groundwater interactions. A modified Richards' equation is used to predict water storage and movement in the variably-saturated soils of the Rollesbroich site:

$$\frac{\partial \theta_i}{\partial t} = \frac{\partial}{\partial z}\left[k_i\left(\frac{\partial(\psi_i + z_i - C_i)}{\partial z}\right)\right] - R_i = \frac{\partial}{\partial z}\left[k_i\left(\frac{\partial(\psi_i - \psi_{e,i})}{\partial z}\right)\right] - R_i, \qquad (B6)$$

where $\theta_i$ [L³L⁻³], $\psi_i$ [L], $k_i$ [LT⁻¹], $z_i$ [L], and $\psi_{e,i}$ [L] denote the volumetric water content, matric head, hydraulic conductivity, depth, and equilibrium matric head of the $i^{th}$ soil layer, $C_i = \psi_{e,i} + z_i$, and $R_i$ [T⁻¹] is the loss of water via root water uptake (canopy transpiration). Note that Equation (B6) omits conveniently the evaporation flux from the first (top) layer. The hydraulic conductivity, $k_i$, of each layer depends on its moisture content, saturated hydraulic conductivity, and exponent $B$, and these values of the adjacent soil layer immediately below, with the exception of the bottom layer (Oleson et al., 2013; Han et al., 2014). The use of the constant $C_i$ in Equation (B6) allows CLM to simulate matric head variations below the water table. This modification maintains a hydrostatic equilibrium soil moisture distribution, and fixes a critical deficiency of the $\theta$-based formulation of Richards' equation (Zeng and Decker, 2009; Oleson et al., 2013).

The matrix head, $\psi_i$, and equilibrium matric head, $\psi_{e,i}$, of each soil layer are computed as follows:

$$\psi_i = \psi_{s,i}\left(\frac{\theta_i}{\theta_{s,i}}\right)^{-B_i} \qquad \text{and} \qquad \psi_{e,i} = \psi_{s,i}\left(\frac{\theta_{e,i}}{\theta_{s,i}}\right)^{-B_i}, \qquad (B7)$$

with

$$\theta_{e,i} = \theta_{s,i}\left(\frac{\psi_{s,i} + z_\nabla - z_i}{\psi_{s,i}}\right)^{(-1/B_i)}, \qquad (B8)$$

where $z_\nabla$ [L] is the depth of the water table.

The bottom boundary condition of Equation (B6) depends on the depth of the water table. This depth, $z_\nabla$, is calculated following Niu et al. (2007) and assumes the presence of an unconfined aquifer below the soil column. If the water table is within the modelled soil column (top 10 layers), then a constant water storage is assumed in the unconfined aquifer (soil column is saturated with water below water table) and a zero-flux bottom boundary condition is used. Recharge, $q_{rec}$ [LT⁻¹], to the unconfined aquifer is calculated as follows:

$$q_{\text{rec}} = -k_{\text{wt}} \left( \frac{-\psi_{\text{wt}}}{z_{\nabla} - z_{\text{wt}}} \right), \tag{B9}$$

where $k_{\text{wt}}$ [LT$^{-1}$], $\psi_{\text{wt}}$ [L], and $z_{\text{wt}}$ [L] signify the hydraulic conductivity, matric head, and depth of the layer that contains the groundwater table. Drainage, $q_{\text{drain}}$ [ML$^{-2}$T$^{-1}$], from the aquifer is calculated via a simple TOPMODEL-based (SIMTOP) scheme (Niu et al., 2005) using:

$$q_{\text{drain}} = 10 \sin(\varepsilon) \exp(-2.5 z_{\nabla}), \tag{B10}$$

where $\varepsilon$ [Rad] signifies the mean topographic slope of the respective grid cell. The change in the water table depth is then given by:

$$\Delta z_{\nabla} = \frac{(q_{\text{rec}} - q_{\text{drain}}) \Delta t}{S_{\text{y}}}, \tag{B11}$$

where $S_{\text{y}}$ [-] denotes the specific yield which depends on the properties of the soil.

**Acknowledgments**

We would like to thank the Terrestrial Environmental Observatories (TERENO) community for freely sharing with us the measurement data of the Rollesbroich experimental test site. The supercomputing centre of Forschungszentrum Jülich is acknowledged for their computational support and our access to the JUROPA cluster. The first author of this paper was funded by a stipend from the government of China. We are grateful to the two anonymous reviewers and editor Kurt Roth for the very careful and detailed evaluation of this manuscript.

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

Table 1. Description of the soil parameters of VIC-3L and CLM that are subject to inference with the different data assimilation methods using the 5-month soil moisture calibration data period of the Rollesbroich site. We list the symbol, unit, feasible range, perturbation, and domain of application of each parameter of VIC-3L and CLM. The column with header "perturbation" lists the statistical distributions that are used to create the initial parameter ensemble for each data assimilation algorithm. The notation $\mathcal{N}(a,b)$ signifies the univariate normal distribution with mean $a$ and standard deviation $b$, whereas $\mathcal{U}(a,b)$ denotes the univariate uniform distribution between $a$ and $b$. These perturbation distributions are centred on the best-guess parameter values of VIC-3L and CLM (see section 4.2) and define together the prior parameter distribution. This prior distribution honours textural measurements of each soil layer and its dispersion is in agreement with previously published studies.

| Model | Parameter | Description | Units | Ranges | Perturbation | Configuration |
|---|---|---|---|---|---|---|
| VIC-3L | $\log_{10}k_s$ | Saturated hydrologic conductivity | $\log_{10}$(m/s)$^\blacklozenge$ | [-7, -3] | $\mathcal{N}(0,1)$ | Layer |
| | $\beta$ | Exponent of Brooks-Corey drainage equation | - | [8, 30] | $\mathcal{U}(-5,5)$ | Layer |
| | $b$ | Infiltration shape parameter | - | $[10^{-3}, 0.8]$ | $\mathcal{U}(-0.1, 0.1)$ | Profile |
| | $D_m$ | Maximum baseflow velocity | mm/d | (0, 30] | $\mathcal{U}(-10, 10)$ | Profile |
| CLM | $f_{cl}$ | Clay fraction | - | [0.01, 1] | $\mathcal{U}(-0.1, 0.1)$ | Layer |
| | $f_{sd}$ | Sand fraction | - | [0.01, 1] | $\mathcal{U}(-0.1, 0.1)$ | Layer |
| | $f_{om}$ | Organic matter fraction | - | [0, 1] | $\mathcal{U}(-0.12, 0.12)$ | Layer |

[†]Note that the sand, clay, and organic matter fraction of each layer serve as input to pedotransfer functions in CLM which compute the hydraulic properties of each layer. See Equations (B2)-(B5) of Appendix B.

[♦]In the figures of this paper we conveniently use labels with units of m/s for $\log_{10}k_s$.

Table 2. Nodal depth, $z$, thickness, $\Delta z$, and depth at layer interface, $zh$, of the ten soil layers used by CLM.

| Layer $i$ | $z$ [m] | $\Delta z$ [m] | $zh$ [m] |
|---|---|---|---|
| 1 (top) | 0.0071 | 0.0175 | 0.0175 |
| 2 | 0.0279 | 0.0276 | 0.0451 |
| 3 | 0.0623 | 0.0455 | 0.0906 |
| 4 | 0.1189 | 0.0750 | 0.1655 |
| 5 | 0.2122 | 0.1236 | 0.2891 |
| 6 | 0.3661 | 0.2038 | 0.4929 |
| 7 | 0.6198 | 0.3360 | 0.8289 |
| 8 | 1.0380 | 0.5539 | 1.3828 |
| 9 | 1.7276 | 0.9133 | 2.2961 |
| 10 | 2.8646 | 1.5058 | 3.8019 |

Table 3. Summary of the different numerical experiments used in this paper for CLM and VIC-3L and their respective abbreviations used in the subsequent tables and figures.

| scenario description | Abbreviation |
|---|---|
| Open loop simulation | OpenLoop |
| EnKF with state estimation | noParamUpdate |
| EnKF with state augmentation | EnKF-AUG |
| EnKF with dual estimation | EnKF-DUAL |
| RRPF with ad-hoc parameter perturbations | RRPF |
| PMCMC | PMCMC |

Table 4. Calibration period: Values of the NSE and RMSE summary statistics of the quality of fit of VIC-3L for the Rollesbroich soil moisture observations at 5, 20, and 50 cm depth using the PMCMC, RRPF, EnKF-AUG and EnKF-DUAL data assimilation methods. For completeness, we also list the performance of the EnKF for state estimation only (noParamUpdate) using VIC-3L parameter values drawn randomly from the prior parameter distribution, and the performance of an open loop run of VIC-3L (OpenLoop) using the mean simulation of many different VIC-3L parameterizations drawn randomly from the prior parameter distribution (see Table 1 and section 4.2).

| Criteria | Soil depth | PMCMC | RRPF | EnKF-AUG | EnKF-DUAL | noParamUpdate | OpenLoop |
|---|---|---|---|---|---|---|---|
| NSE (-) | 5 cm | 0.82 | 0.73 | 0.80 | 0.82 | 0.89 | 0.33 |
| | 20 cm | 0.80 | 0.84 | 0.92 | 0.91 | 0.86 | -1.16 |
| | 50 cm | 0.27 | -11.77 | 0.69 | 0.58 | 0.91 | -26.65 |
| RMSE ($m^3/m^3$) | 5 cm | 0.019 | 0.023 | 0.020 | 0.019 | 0.015 | 0.036 |
| | 20 cm | 0.011 | 0.010 | 0.007 | 0.007 | 0.009 | 0.037 |
| | 50 cm | 0.021 | 0.088 | 0.014 | 0.016 | 0.008 | 0.129 |

Table 5. Evaluation period: Values of the NSE and RMSE summary statistics of the quality of fit of VIC-3L for the Rollesbroich soil moisture observations at 5, 20, and 50 cm depth using the calibrated parameter values and initial states derived from the PMCMC, RRPF, EnKF-AUG and EnKF-DUAL data assimilation methods. For completeness, we also list the performance of the EnKF using state estimation only (noParamUpdate) using VIC-3L parameter values drawn randomly from the prior parameter distribution, and the performance of an open loop run of VIC-3L (OpenLoop) using the mean simulation of many different VIC-3L parameterizations drawn randomly from the prior parameter distribution.

| Criteria | Soil depth | PMCMC | RRPF | EnKF-AUG | EnKF-DUAL | noParamUpdate | OpenLoop |
|---|---|---|---|---|---|---|---|
| NSE (-) | 5 cm | 0.39 | 0.39 | 0.39 | 0.39 | 0.35 | 0.36 |
| | 20 cm | 0.38 | 0.47 | 0.40 | 0.39 | -1.75 | -1.87 |
| | 50 cm | -10.33 | -8.41 | -10.54 | -11.33 | -26.83 | -32.96 |
| RMSE $(m^3/m^3)$ | 5 cm | 0.052 | 0.052 | 0.052 | 0.052 | 0.054 | 0.053 |
| | 20 cm | 0.026 | 0.024 | 0.026 | 0.026 | 0.055 | 0.056 |
| | 50 cm | 0.076 | 0.069 | 0.077 | 0.079 | 0.119 | 0.132 |

Table 6. RMSE values of VIC-3L for the Rollesbroich soil moisture measurements at 5, 20, and 50 cm depth using the EnKF with state AUGmentation or DUAL estimation during the calibration period. We also summarize the subsequent performance of the VIC-3L model using the calibrated parameter values and initial states derived from AUG and DUAL. The subscripts 10% and 20% signify the standard deviations of the measurements errors that are used to corrupt the hourly precipitation data.

| Period | Soil depth | EnKF-AUG_10% | EnKF-AUG_20% | EnKF-DUAL_10% | EnKF-DUAL_20% |
|---|---|---|---|---|---|
| Calibration (Assimilation) | 5 cm | 0.020 | 0.019 | 0.019 | 0.019 |
| | 20 cm | 0.007 | 0.007 | 0.007 | 0.007 |
| | 50 cm | 0.014 | 0.014 | 0.016 | 0.014 |
| Evaluation | 5 cm | 0.052 | 0.052 | 0.052 | 0.052 |
| | 20 cm | 0.026 | 0.025 | 0.026 | 0.025 |
| | 50 cm | 0.077 | 0.077 | 0.079 | 0.079 |

Table 7. RMSE values of VIC-3L for the Rollesbroich soil moisture observations at 5, 20, and 50 cm depth using data assimilation with RRPF during the calibration period. We also summarize the subsequent performance of the VIC-3L model using the calibrated parameter values and initial states derived from RRPF. The subscripts 0.01, 0.1, and 0.5 signify the value of the scaling factor $s$ of the multivariate normal distribution that is used to perturb the parameter values (importance density).

| Period | Soil depth | RRPF-0.01 | RRPF-0.1 | RRPF-0.5 |
|---|---|---|---|---|
| Calibration (Assimilation) | 5 cm | 0.025 | 0.023 | 0.015 |
| | 20 cm | 0.012 | 0.010 | 0.007 |
| | 50 cm | 0.113 | 0.088 | 0.037 |
| Evaluation | 5 cm | 0.053 | 0.052 | 0.056 |
| | 20 cm | 0.025 | 0.024 | 0.020 |
| | 50 cm | 0.119 | 0.069 | 0.071 |

Table 8. Calibration period: Values of the NSE and RMSE summary statistics of the quality of fit of CLM for the Rollesbroich soil moisture measurements at 5, 20, and 50 cm depth with the PMCMC, RRPF, EnKF-AUG and EnKF-DUAL data assimilation methods. For completeness, we also list the performance of the EnKF for state estimation only (noParamUpdate) using CLM parameter values drawn randomly from the prior parameter distribution, and the performance of an open loop run of CLM (OpenLoop) using the mean simulation of many different CLM parameterizations drawn randomly from the prior parameter distribution.

| Statistic | Soil depth | PMCMC | RRPF | EnKF-AUG | EnKF-DUAL | noParamUpdate | OpenLoop |
|---|---|---|---|---|---|---|---|
| NSE (-) | 5 cm | 0.63 | 0.63 | 0.82 | 0.85 | 0.72 | -0.31 |
| | 20 cm | 0.73 | 0.23 | 0.94 | 0.95 | 0.98 | -0.57 |
| | 50 cm | 0.50 | -0.26 | 0.85 | 0.86 | 0.47 | -6.90 |
| RMSE ($m^3/m^3$) | 5 cm | 0.027 | 0.027 | 0.019 | 0.017 | 0.024 | 0.051 |
| | 20 cm | 0.013 | 0.022 | 0.006 | 0.006 | 0.004 | 0.031 |
| | 50 cm | 0.017 | 0.028 | 0.009 | 0.009 | 0.018 | 0.069 |

Table 9. Evaluation period: NSE and RMSE values for the Rollesbroich soil moisture measurements at 5, 20, and 50 cm depth using CLM. The initial states and parameter values used by the PMCMC, RRPF, EnKF-AUG and EnKF-DUAL data assimilation methods originate from the 5-month calibration data period. For completeness, we also list the performance of the EnKF using state estimation (noParamUpdate) using CLM parameter values drawn randomly from the prior parameter distribution, and the performance of an open loop run of CLM (OpenLoop) using the mean simulation of many different CLM parameterizations drawn randomly from the prior parameter distribution.

| Criteria | Soil depth | PMCMC | RRPF | EnKF-AUG | EnKF-DUAL | noParamUpdate | OpenLoop |
|---|---|---|---|---|---|---|---|
| NSE (-) | 5cm | 0.26 | 0.23 | 0.32 | 0.33 | -0.19 | -0.14 |
| | 20cm | 0.39 | 0.21 | 0.44 | 0.46 | 0.24 | -0.11 |
| | 50cm | 0.35 | -0.23 | 0.51 | 0.42 | -3.87 | -4.58 |
| RMSE ($m^3/m^3$) | 5cm | 0.057 | 0.058 | 0.055 | 0.054 | 0.072 | 0.071 |
| | 20cm | 0.026 | 0.029 | 0.025 | 0.024 | 0.031 | 0.035 |
| | 50cm | 0.018 | 0.025 | 0.016 | 0.017 | 0.050 | 0.053 |

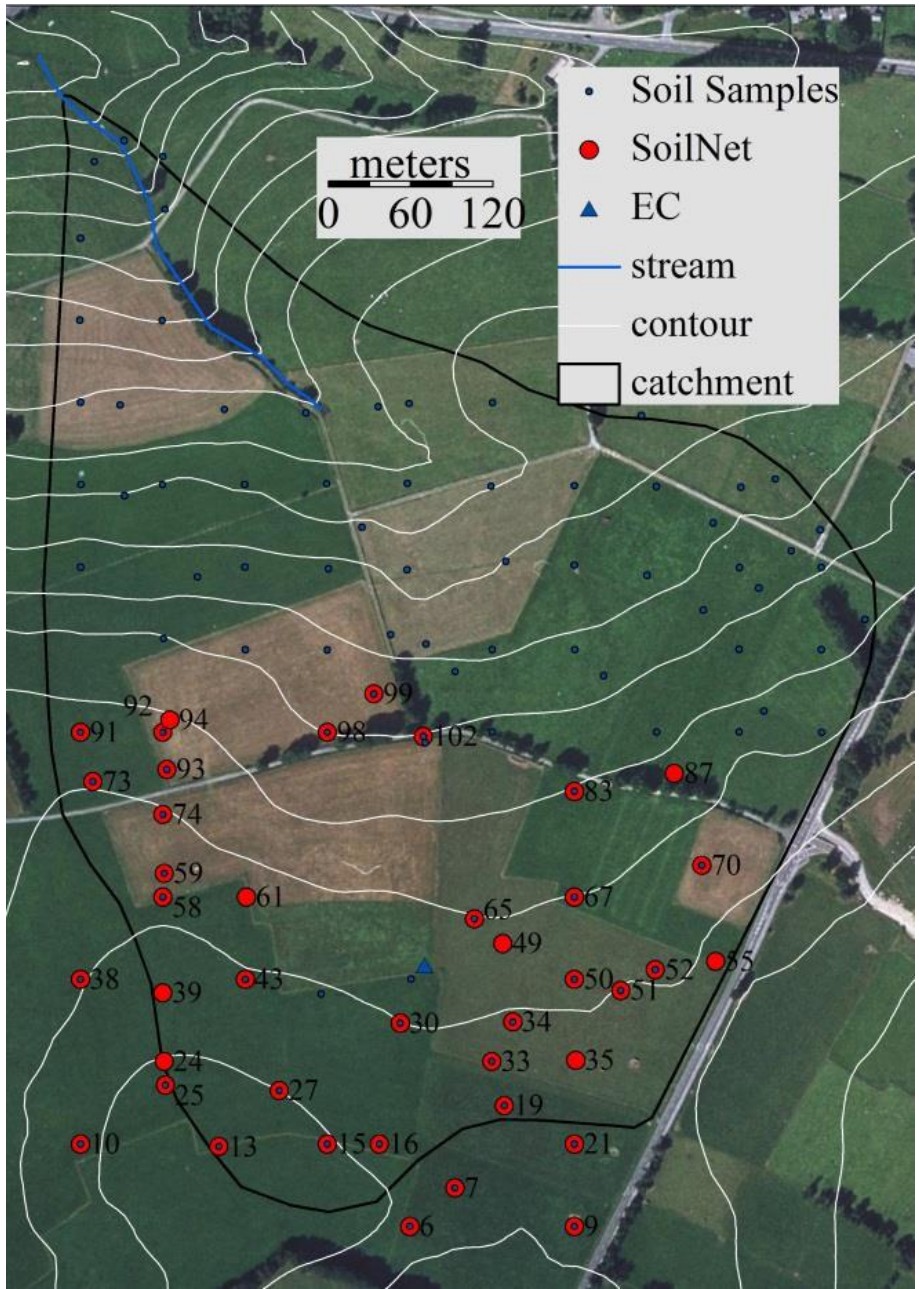

Figure 1. Aerial photograph of the 270,000 m$^2$ Rollesbroich experimental test site near the city of Rollesbroich in the Eifel mountain range, western Germany (photo is taken from Qu et al., (2014)). The solid black line signifies the outer perimeter of our site and is determined in part by topographic gradients except for the Rollesbroich Straße which acts as border in the East-Southeast part of our domain. The small blue dots characterize locations within the watershed where soil samples were taken. The larger red dots are locations of the sensor network where soil moisture and temperature were recorded at depths of 5, 20, and 50 cm. The blue triangle symbolizes the eddy covariance tower.

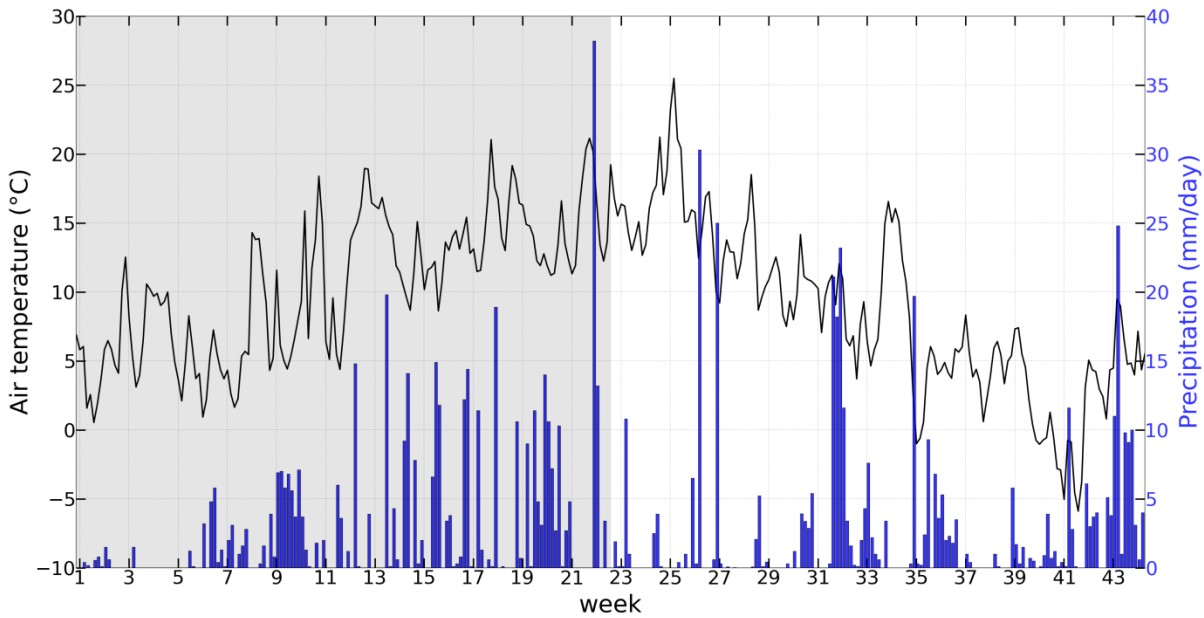

Figure 2. Historical records of daily mean air temperature (solid black line; left *y*-axis) and precipitation (blue bars; right *y*-axis) in the period from March 1, 2012 to December 31, 2012 for the Rollesbroich experimental test site in the Eifel mountain range in western Germany. The grey region demarcates the 5-month assimilation period (March 1, 2012 to July 31, 2012) which is used for VIC-3L and CLM calibration using joint parameter and state estimation. The subsequent 5-month period between August 1, 2012 and December 31, 2012 serves as our evaluation period to verify the performance of the calibrated VIC-3L and CLM models without state estimation.

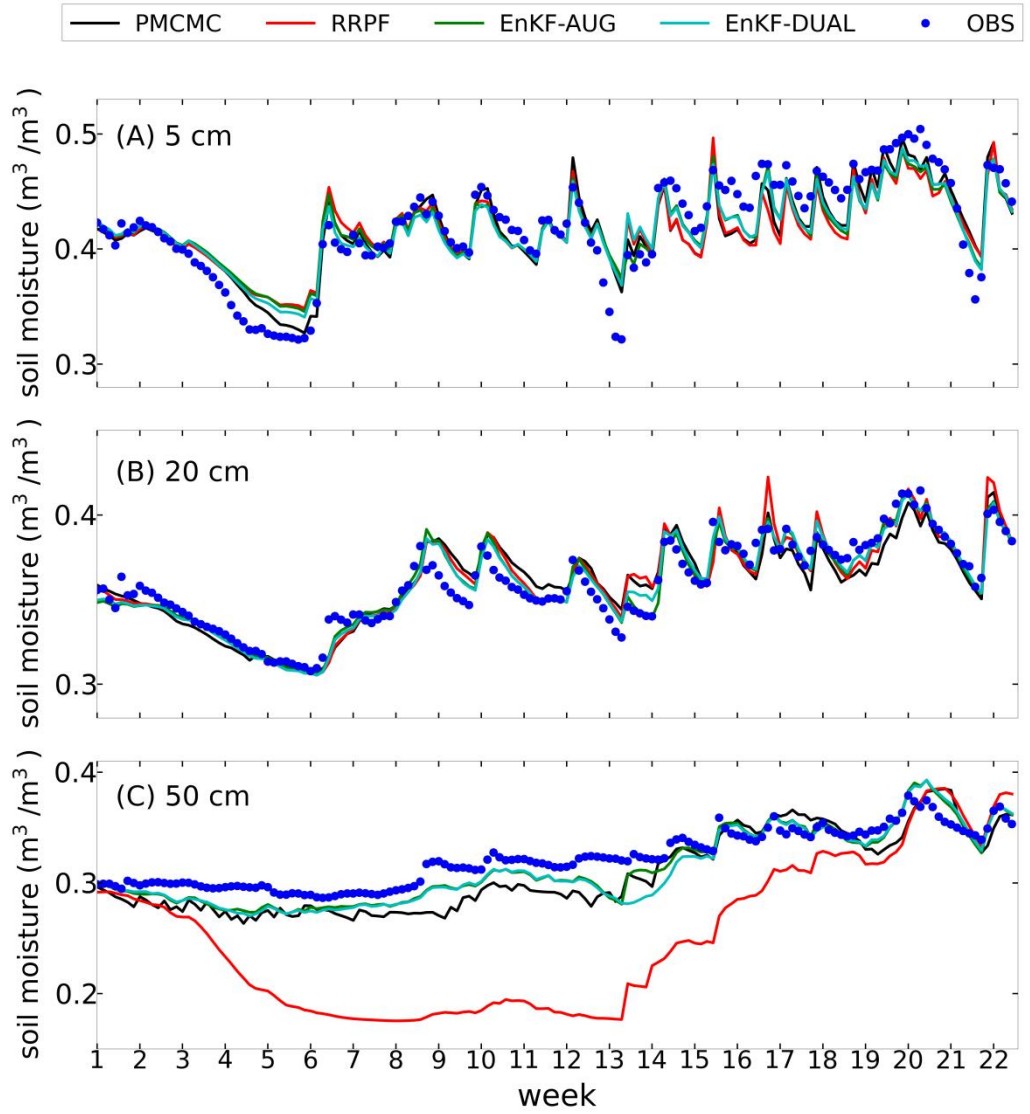

Figure 3. Assimilation period: Observed (blue dots) and VIC-3L predicted time series (solid lines) of soil moisture content at depths of (A) 5, (B) 20, and (C) 50 cm in the Rollesbroich site. Colour coding is used to differentiate between the results of PMCMC [black], RRPF [red], EnKF-AUG [green], and EnKF-DUAL [cyan]. The first days of week 1 and 22 are 01-03-2012 and 26-07-2012, respectively.

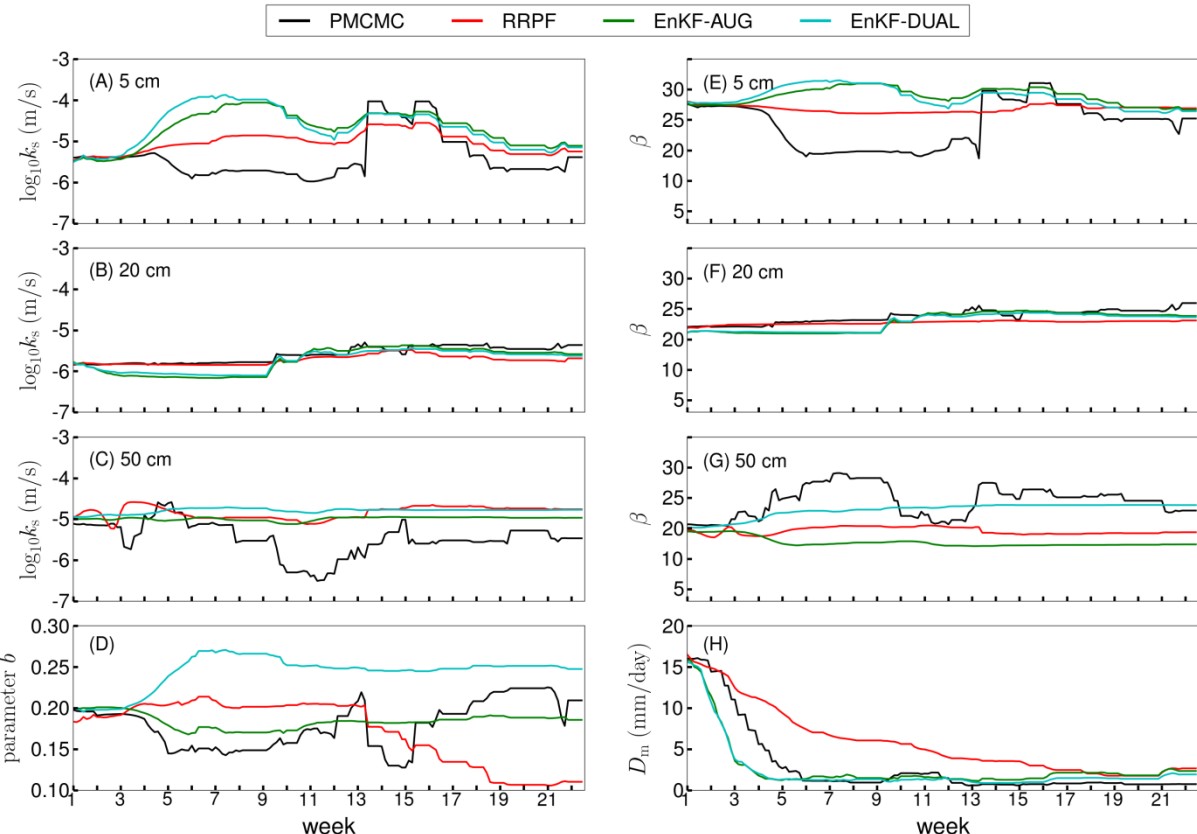

Figure 4. Traceplots (solid lines) of the VIC-3L parameters, Saturated hydraulic conductivity ($\log_{10}k_s$ in m/s) at (A) 5 cm, (B) 20 cm, and (C) 50 cm depth, (D) $b$, $\beta$ at (E) 5 cm, (F) 20 cm, and (G) 50 cm depth, and (H) the maximum baseflow velocity, $D_m$, in mm/day during the 5-month assimilation period. Colour coding is used to differentiate between the results of PMCMC [black], RRPF [red], EnKF-AUG [green], and EnKF-DUAL [cyan]. The first days of week 1 and 22 are 01-03-2012 and 26-07-2012, respectively.

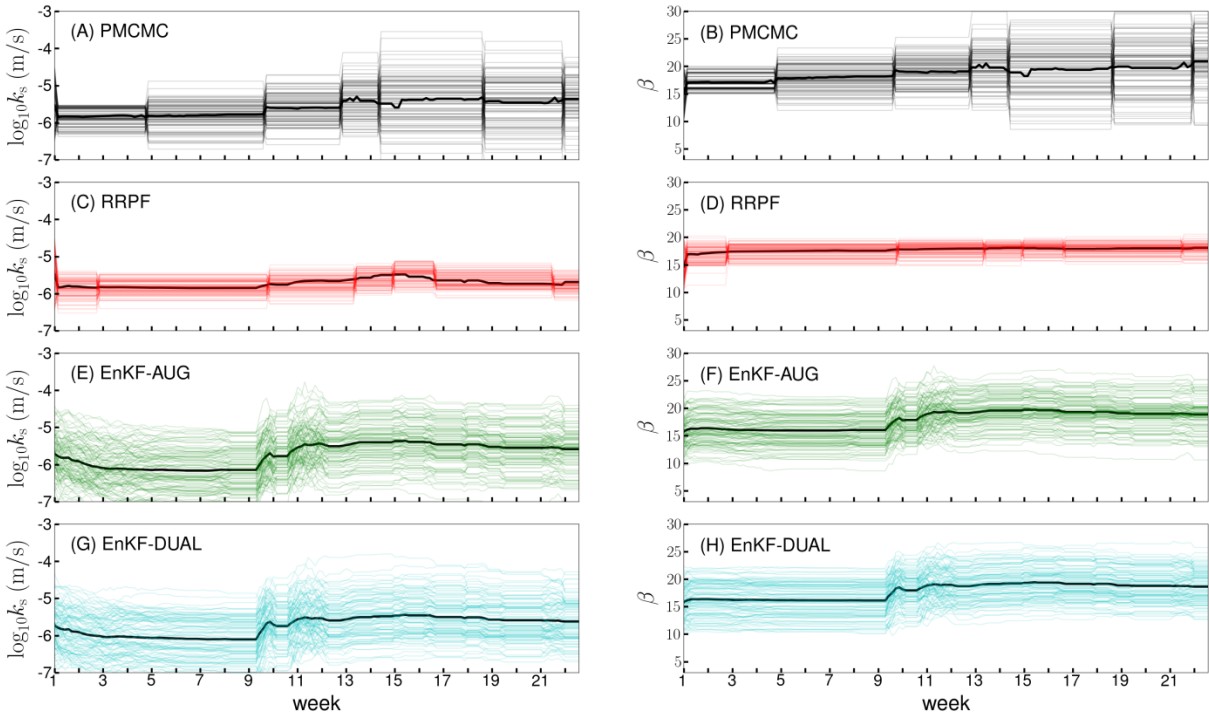

Figure 5. Sampled trajectories of the $N = 100$ ensemble members of the saturated hydraulic conductivity ($\log_{10} k_s$ in m/s) at 20 cm depth (left column) and parameter $\beta$ (right column) of VIC-3L during the 5-month assimilation period of week 1 to 22 using (A-B) PMCMC [grey] (C-D) RRPF [red], (E-F) EnKF-AUG [green], and (G-H) EnKF-DUAL [cyan]. The trajectory of the ensemble mean is separately indicated in each panel using the solid black line.

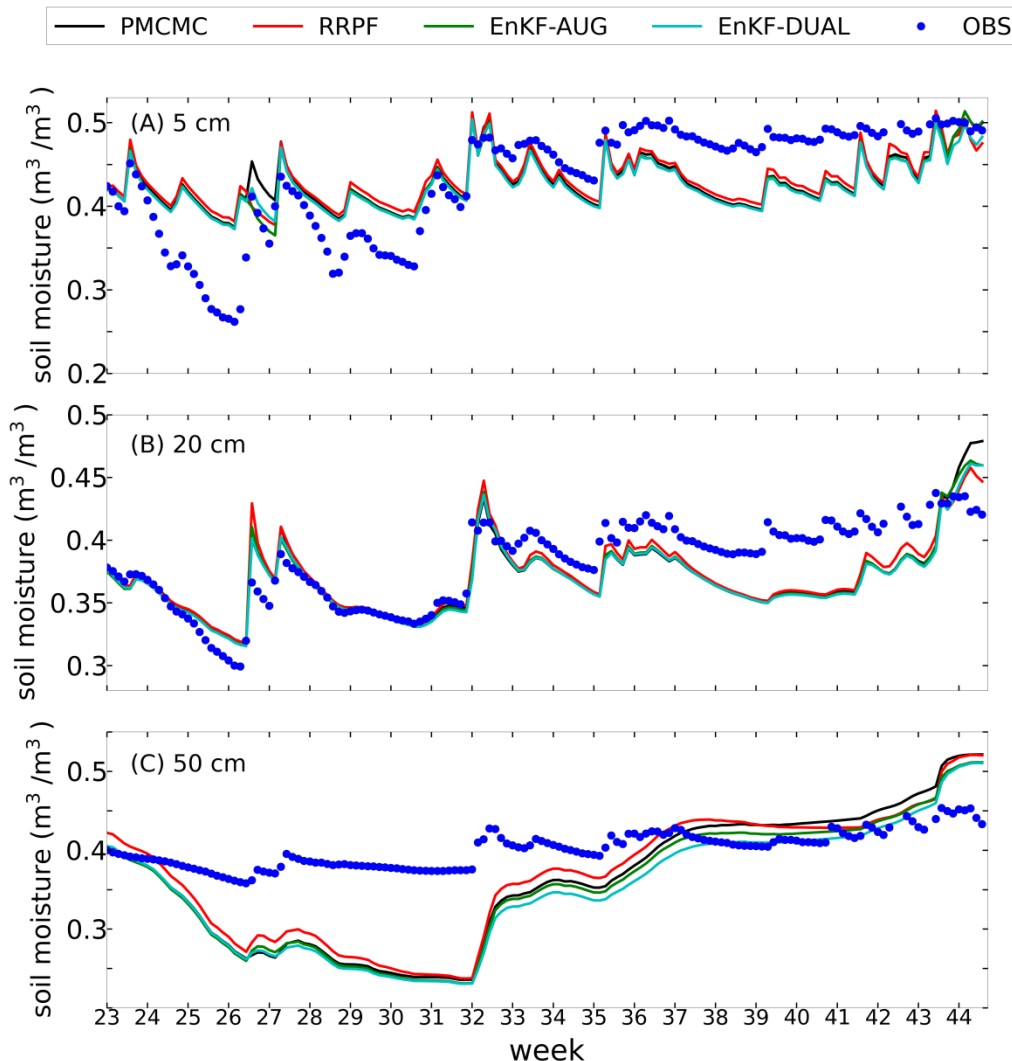

Figure 6. Evaluation period: Observed (blue dots) and VIC-3L simulated time series (solid lines) of soil moisture content at depths of (A) 5 cm, (B) 20 cm, and (C) 50 cm in the Rollesbroich site. Colour coding is used to differentiate between the results of PMCMC [black], RRPF [red], EnKF-AUG [green], and EnKF-DUAL [cyan]. The first days of week 23 and 44 are 01-08-2012 and 26-12-2012, respectively.

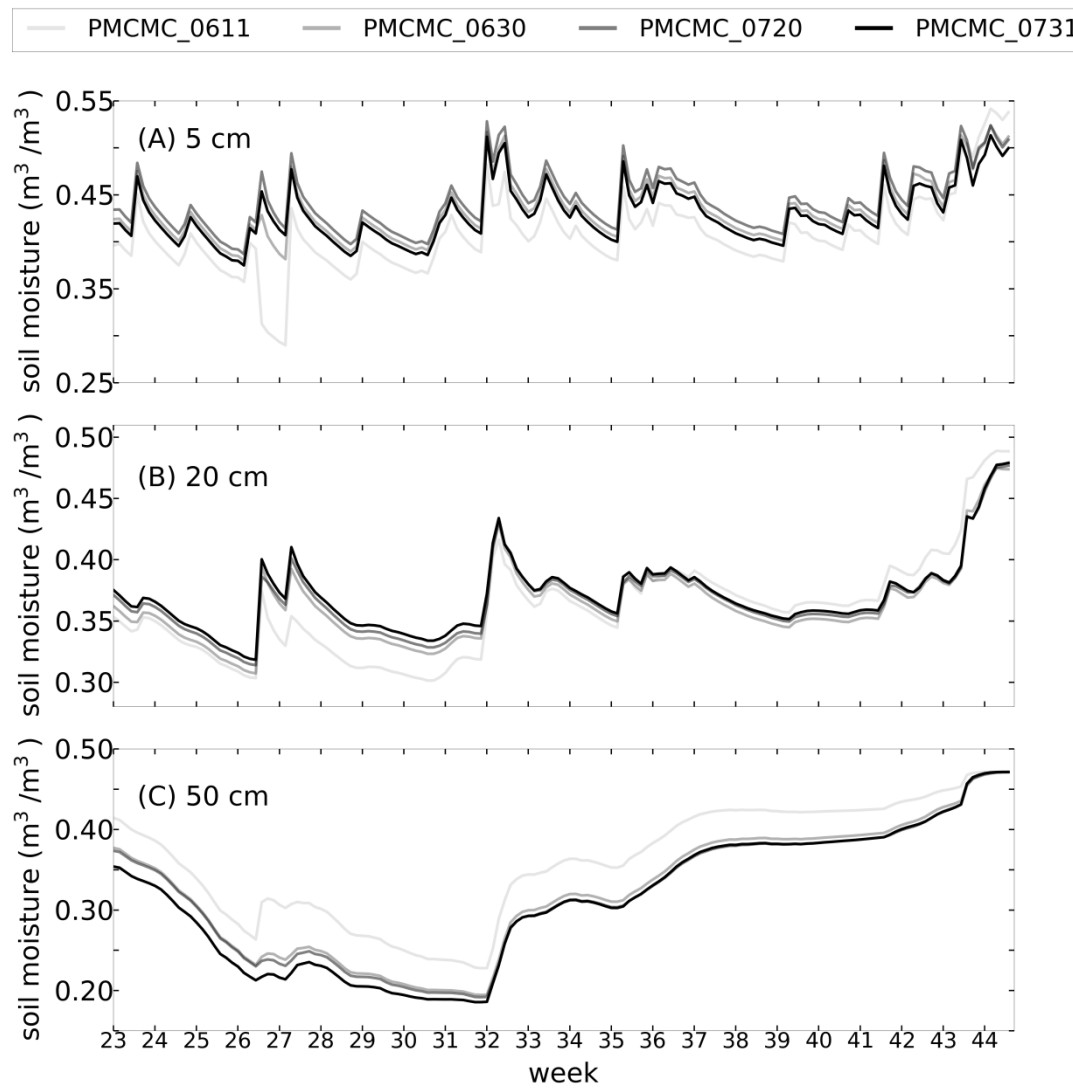

Figure 7. Evaluation period: VIC-3L simulated volumetric moisture contents at (A) 5 cm, (B) 20 cm, and (C) 50 cm depth in the soil of the Rollesbroich experimental test site using parameter values derived from PMCMC via assimilation periods ending on 06-11 [platinum], 06-30 [silver], 07-20 [grey] and 07-31 [black], respectively. For PMCMC_0611, PMCMC_0630 and PMCMC_0720, the soil moisture state on 01-08-12, the first day of the 5-month evaluation period, was derived from VIC-3L simulation using the analysis state and parameter values of the last day of the assimilation period.

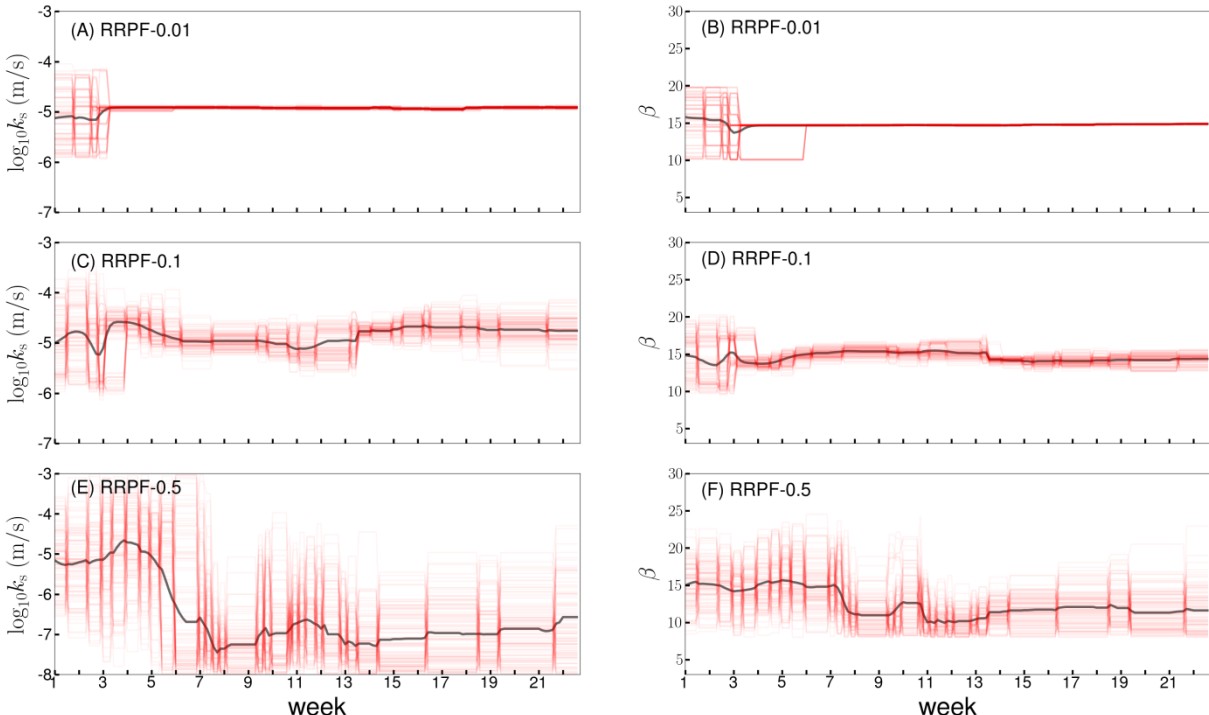

Figure 8. Sampled trajectories of the $N = 100$ ensemble members (solid red lines) of the saturated hydraulic conductivity ($\log_{10}k_s$ in m/s) at 50 cm depth (left column) and parameter $\beta$ (right column) of VIC-3L during the 5-month assimilation period of week 1 to 22 using (A-B) RRPF-0.01, (C-D) RRPF-0.1, and (E-F) RRPF-0.5. The ensemble mean is separately indicated in each panel with the solid grey line.

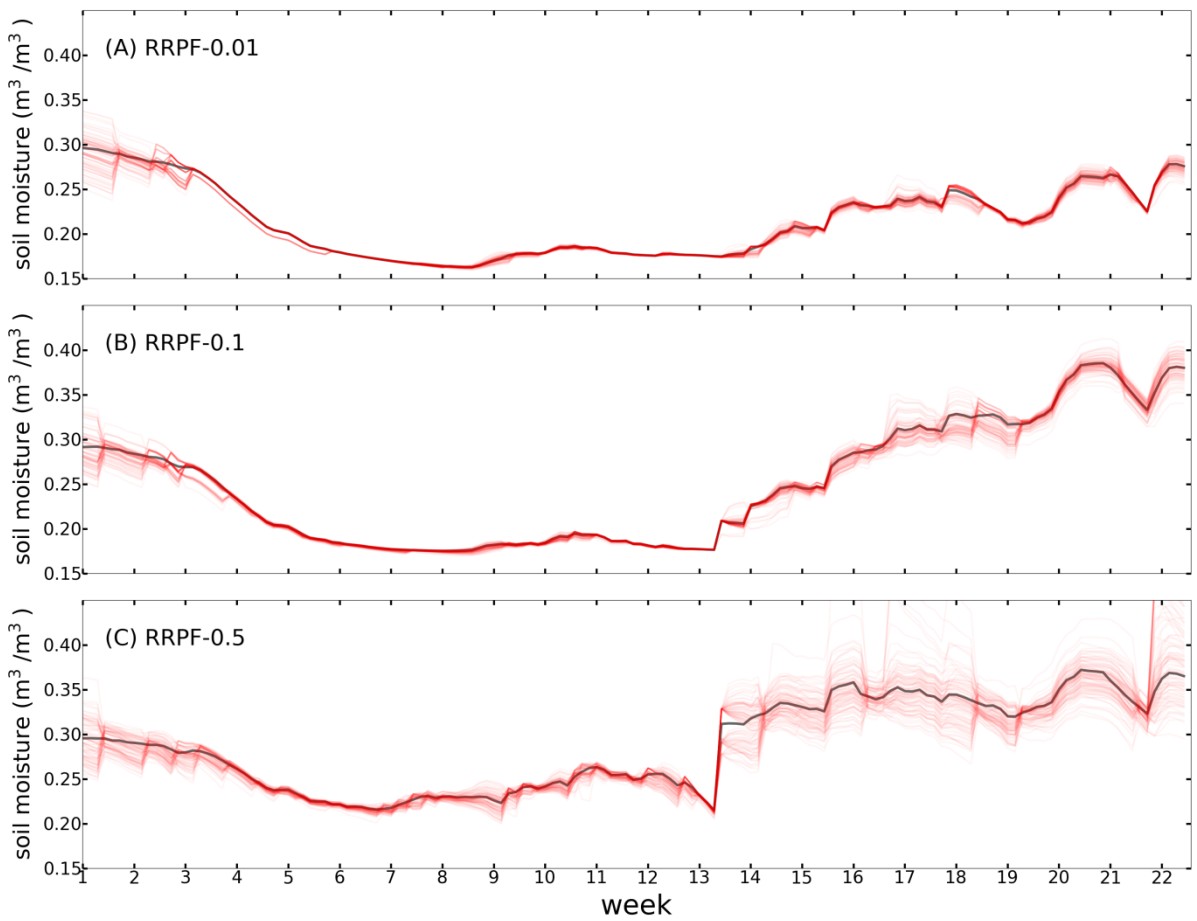

Figure 9. Soil moisture trajectories of the $N = 100$ ensemble members at 50 cm depth for the 5-month assimilation period (week 1 to 22) of the Rollesbroich site using VIC-3L and (A) RRPF-0.01, (B) RRPF-0.1, and (C) RRPF-0.5. The solid black line signifies the ensemble mean soil moisture prediction.

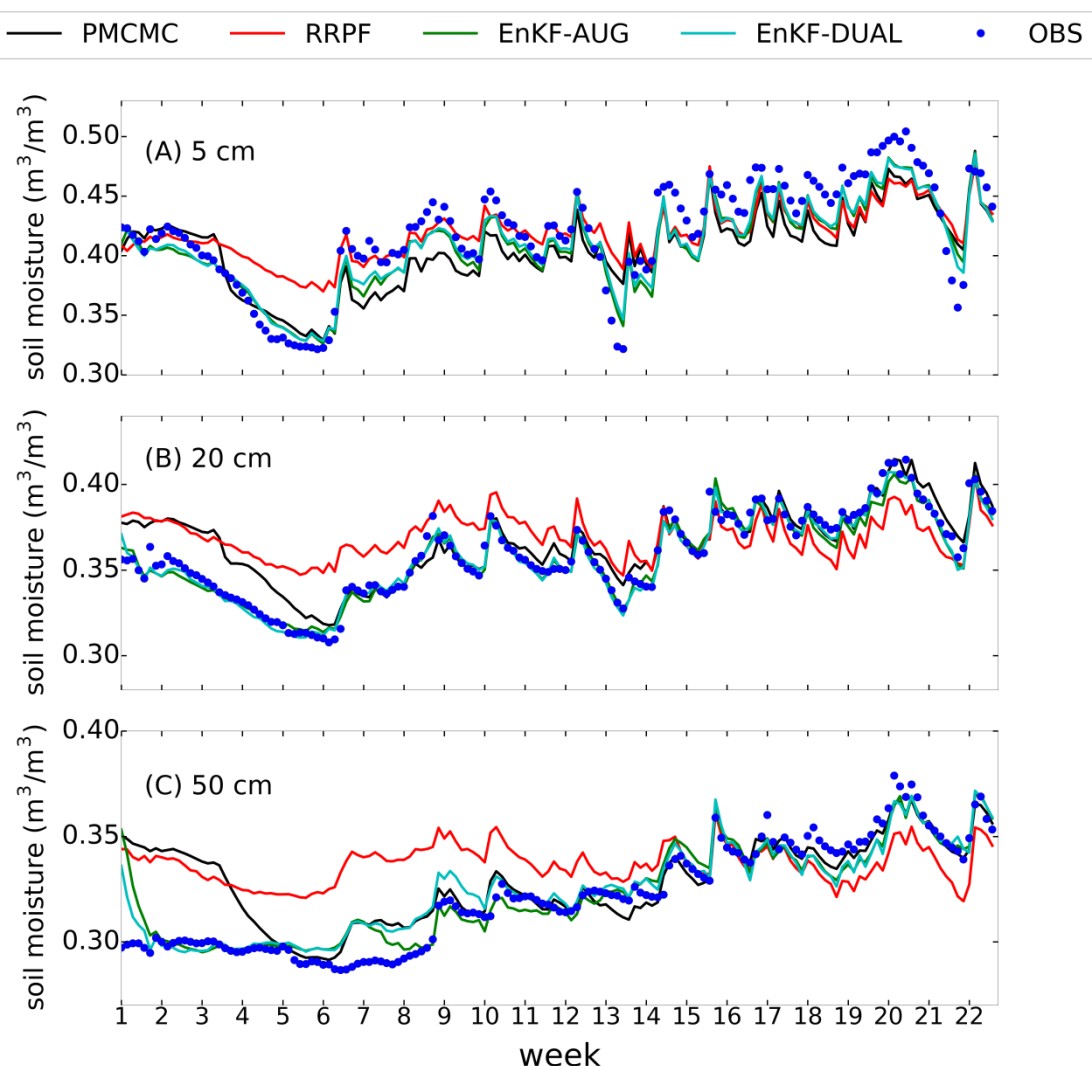

Figure 10. CLM predicted time series of soil moisture content at (A) 5 cm, (B) 20 cm, and (C) 50 cm depth during the 5-month calibration period using PMCMC [black], RRPF [red], EnKF-AUG [green], and EnKF-DUAL [cyan]. The first day of week 1 is 01-03-2012 and week 22 starts with 26-07-2012.

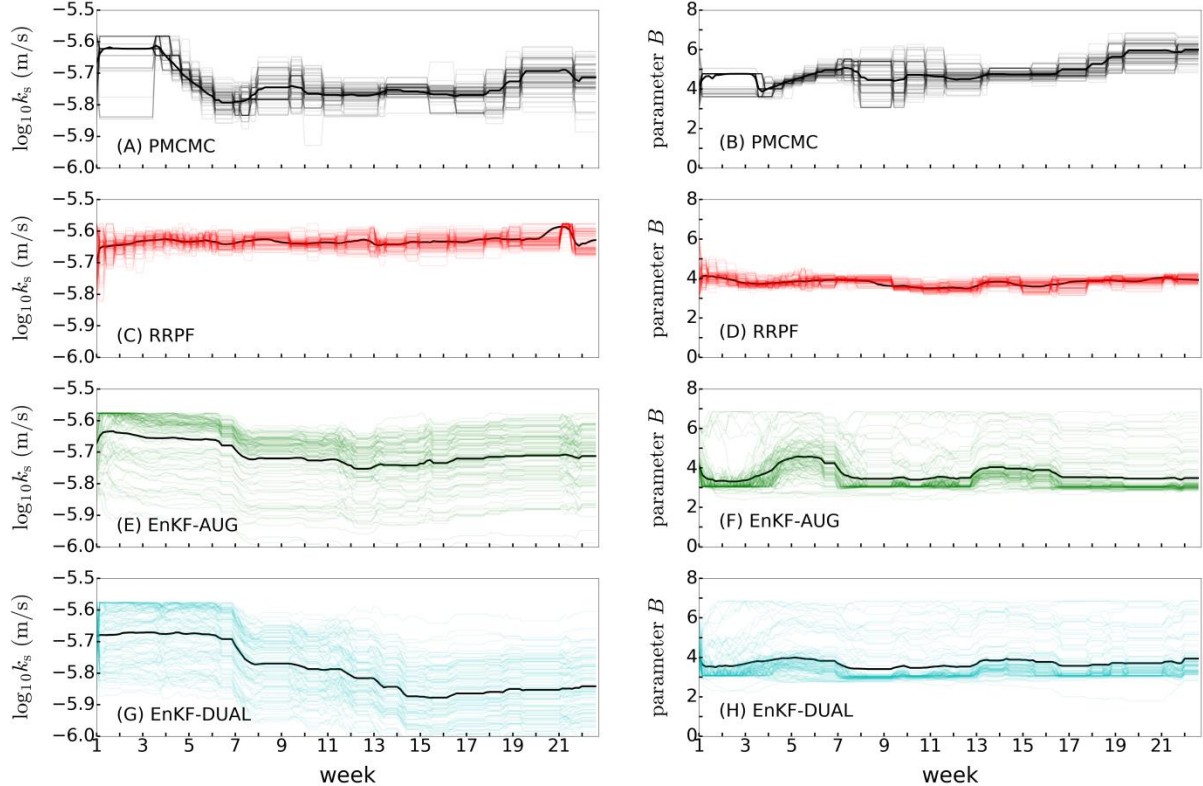

Figure 11. Sampled trajectories of the $N = 100$ ensemble members of the saturated hydraulic conductivity ($\log_{10}k_s$ in m/s) at 50 cm depth (left column) and soil hydraulic parameter $B$ at 50 cm depth (right column) of CLM during the 5-month assimilation period of week 1 to 22 using (A-B) PMCMC [grey] (C-D) RRPF [red], (E-F) EnKF-AUG [green], and (G-H) EnKF-DUAL [cyan]. The solid black line signifies the evolution of the ensemble mean values of $\log_{10}k_s$ and $B$. Please note that $\log_{10}k_s$ (in $\log_{10}$(m/s)) and parameter $B$ are derived from the sand, clay, and organic matter fractions of each soil layer, which are estimated during the assimilation period.

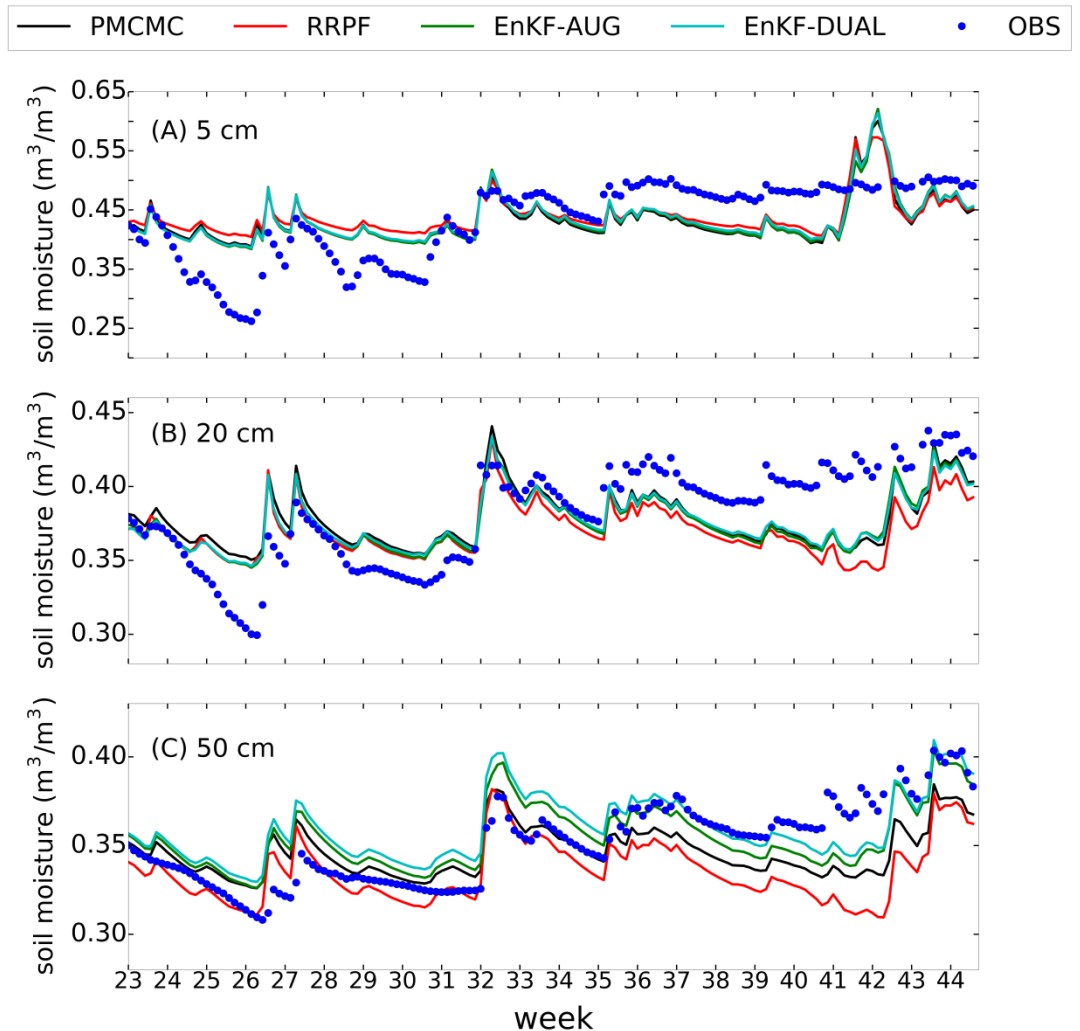

Figure 12. Traceplots of soil moisture contents simulated by CLM during the evaluation period at (A) 5 cm, (B) 20 cm, and (C) 50 cm depth in the Rollesbroich site using the calibrated parameter values derived from PMCMC [black], RRPF [red], EnKF-AUG [green], and EnKF-DUAL [cyan]. The measured moisture data are separately indicated in each panel with the solid blue dots. The first day of week 23 is 01-08-2012 and the last day of week 44 is 02-01-2013.