# Peer review of "State and Parameter Estimation of Two Land Surface Models Using the Ensemble Kalman Filter and the Particle Filter"

_Hydrology and Earth System Sciences, 2016_

## Referee Comment (RC1) · Anonymous Referee #1 · 22 Mar 2016

The manuscript addresses an important issue in soil hydrology: The application of data assimilation methods to real world data, especially when estimating not only states, but also parameters. The authors state to address three main questions: (1) the performance of the data assimilation methods on Land Surface Models with real world data in general, (2) the differences in performance due to different data assimilation methods and (3) difference in performance due to different Land Surface Models. The study finds small differences due to data assimilation methods and large differences based on different Land Surface Models.

These findings can give valuable insight for the applicability of data assimilation methods on Land Surface Models. But, this requires an adequate discussion of the

used measurements, data assimilation methods and finally the results.

I have 3 major comments regarding each of these discussions, which fall short off answering the stated main questions enough. Additionally I have one major comment on the quality of the explanations for employed methods and models.

**General comments:**

**1. Discussion of the used measurements:** Since data assimilation combines observations and measurements based on their uncertainties, a good description of these uncertainties can be important for good performance. Because of this, I would appreciate a detailed discussion of the measurements, their uncertainties and the implications of the assumptions made.

Especially discuss the following points:

Page 10, Lines 20-23: " . . . and a soil moisture and soil temperature sensor network (with measurements at 5, 20 and 50cm depth) are installed, amongst others. Soil moisture time series at 41 locations are being recorded."

What kind of soil moisture sensors are installed? Please discuss possible uncertainties in the data.

Page 10, Lines 24-35: "In this work, the Rollesbroich site is modeled as a single point and the data of the soil sensor network are averaged to calculate areal averages of soil moisture content at 5cm, 20cm and 50cm depth."

Please discuss the importance and implications of this assumption. What are the expected impacts of heterogeneity?

Page 29, Figure 3: The figure shows higher water contents closer to the surface. Please mention this and discuss reasons and implications.

Page 11 Lines 34-36: "The soil moisture observation error is assumed to be normally distributed with mean equal to 0 and standard deviation equal to 0.02m3/m3, for both VIC-3L and CLM." Please discuss why you assume this uncertainty, especially since it is a mean of 41 values.

Page 11 Lines 33-34: "Precipitation was perturbed were perturbed by multiplicative error N(1,0.1) to represent the uncertainty of measured precipitation at the site." Please give a reason for this error. What is the assumed error for evaporation?

**2. Discussion of the assimilation methods:** Since the methods combine observations and models based on their uncertainties, the representation of the model uncertainties can be important. Please discuss the following points in more detail:

Page 2, Lines 13-15: "This approach allows for joint estimation of the states and parameters while taking into explicit consideration model structural error and forcing data errors (Liu and Gupta, 2007)." This is correct, but it is not to the point, since the authors later set the model error to zero (see Page 11, Lines 36-37) and hence do not consider model structural errors. Please discuss this.

Page 6, Line 25 (Eq. 25): Please mention that the way R (and $y_t^i$) is described, you assume uncorrelated measurement errors.

Page 7, Line 25 (Eq. 33): Please describe the implications of employing this method. How does the performance of the filter depend on the choice of initial uncertainty?

Page 9, Line 13. How did you choose the tuning parameter s? How does it influence the performance? Please include the choice of s for the PF and initial uncertainties of the EnKF when comparing different methods.

Page 15 Line 32: You state: "It is not surprising that the EnKF is more efficient and effective than the PF." I would follow this statement in case of strictly Gaussian distributions and linear measurement operators. In those cases the EnKF is expected to outperform the PF. Nevertheless, when dealing with non-linear processes that challenge the Gaussian assumption, the better performance is not clear at all.

Page 165 Line 33: You state that EnKF and PF "differ fundamentally in their analysis step". I would disagree and argue that the analysis step is similar. The only difference is that the EnKF updates it's posterior based on the Gaussian assumption of the distributions, while the PF drops this assumption.

Page 25, Table 1: The EnKF assumes Gaussian distributions. What is the reason that you sample for all but one parameter from an initial uniform distribution? What are the implications for the chosen inflation method?

Page 16, Lines 1-2: You state: "The value of the likelihood does generally not say anything about how close the forecasted variables are to their measured counterparts." I disagree, the likelihood yields information about the distance.

**3. Discussion of results:** Since the results show a wealth of information. I would appreciate a detailed discussion. Especially consider the following points in detail and incorporate them into your conclusion:

Page 29, Figure 3: The large deviations in the Particle Filter might hint at filter inbreeding in the states. You observe the deviation but do not discuss it and actually exclude the possibility of inbreeding by investigating the parameters: "A too narrow spread of ensemble members would lead to filter divergence. For the state augmentation (AUG) and dual estimation (DUAL), the spread of the ensemble members is kept large enough during the whole assimilation period as the ensemble inflation method helped to keep adequate ensemble spread. RRPF and MCMCPF also have enough ensemble spread because of parameter perturbation and MCMCPF resampling." (Page 13 Lines 15-19). Please address the question of adequate ensemble spread in the states by actually showing the ensemble of states there.

Page 30 Figure 4: During the calibration period the filter without parameter estimation performs better. Please discuss possible reasons.

Page 31 Figure 5: Parameter b estimated by MCMC shows a large difference to the other methods. But MCMC does perform approximately as well as the other filters (Figure 7, Page 33). Why is there no difference? Please discuss.

Page 32 Figure 6: Initial parameter uncertainties are the same for PF and EnKF but at time 0 the ensemble spreads are different. Please explain.
You only show the two parameters with the least change over time. Give a reason or show the one with the smallest and the one with the largest changes.

Page 34, Figure 8: There is basically no difference for the prediction of the water content whether there are parameters estimated or not. Please discuss why this is the case.

Page 37 Figure 11: You do not show the parameters from the augmented state.

Instead you show parameters derived from those in a non-linear way. Please discuss this.

Figures 7, 8, 12, 13: Especially water content of the top layer can almost not be represented by either model, although improved with state of the art data assimilation methods.

Please discuss this including the representation of the physics in the models and implications of the perfect model assumption.

The difference in the assimilation methods is small. Please discuss if this difference is significant. Do you expect the same results for other applications? What is the influence of specific filter settings on the performance?

**4. Please improve the explanations on the Particle Filter and the two LSMs.**

Page 7 Line 32 - Page 8 Line 11: I do understand Particle Filters, but the given explanation is not clear. Especially clarify your description of transition and proposal densities.

Page 8 Lines 21-30: You describe SIR and RR. What is the reason to choose RR?

Page 7 Lines 31-32: "The particle filter was first suggested in the research area of object recognition, robotics and target tracking (Arulampalam et al., 2002)." The PF was actually mentioned earlier (Gordon et al., 1993).

For this study the understanding of the different LSM models is important, since the main difference in performance is attributed to different models. Because of that a good description is necessary. Although I am not an expert on LSMs, I noticed the following:

Page 17, Line 15 (Eq. A2): The equation describes the soil water movement in the top two layers. To me it was not clear if this description is valid for both layers individually.

[Figure]

If so, why is the precipitation P and evaporation E the same?

Page 18, Line 1 (Eq. A7): The dimensions are inconsistent: You add [LT-1], [] and [L]. $i_m$ should be $I_m$ (or is not introduced).

Please also explain the distinction of the cases $P + I < I_m$ and $P + I > I_m$.

Page 19, Lines 18-22: the Richards equation is formulated for a continuum. Please explain the modifications and the implications of applying it to layers. Please give the extent of these layers.

**Specific comments:**

Page 5, Line 36: "Commonly used data assimilation algorithms are EnKF, PF and variants of them." 4D-Var is also commonly used.

Page 6, Line 19: "H [...] is the identity matrix if y refers to in-situ ground measurements available at all grid cells." This is only the case if the same quantity as the state is observed. Otherwise the quantity has to be transferred. Additionally mention that H has to be linear for the EnKF.

Page 6-7: You do not mention the use of a damping factor (Hendricks Franssen and Kinzelbach, 2008) for the parameters. Is there a specific reason you do not employ it?

Page 25, Table 1: It is not clear which parameters are estimated for each layer individually and which are estimated for the entire profile.

Page 11 Lines 11-12: "The parameters of the other layers were updated with help of the calculated spatial covariances in case of EnKF."

This statement implies that the parameters were not estimated with the PF? Please clarify.

Page 11 Line 30: "EnKF with state updating only was tested for (2)."
Why did you not test state updating with the PF?

Page 12 Lines 2-10: Why do you need 2 different characterizations of the uncertainty. What is the additional gain by showing NSE?

Page 12 Line 9-10: "A NSE value equal to 1 and RMSE equal to 0 imply a perfect prediction." This is wrong. A RMSE equal to the measurement uncertainty is perfect.

**Technical corrections:**

Page 6, Line 8: "$v_{t-1}$ to model error at time step t" should be $v_t$

Page 8, Line 9 (Eq. 35): Sign not printed (displayed as empty square).

Page 9, Lines 6-13 and Page 9, Line 31-Page 10, Line 13: Algorithms are formatted differently.

Page 11 Lines 33: " . . . was perturbed were perturbed . . ."

Page 13, Line 3: "Even although" should be "even though"?

Page 17, Line 9: Introduce Leaf Area Index, before using the abbreviation LAI.

Page 18, Line 1 (Eq. A7): Maybe choose another letter than I since it could be confused with 1 in the exponent.

Page 18, Line 25: "soil matric potential $\psi_i$ [L]" is actually the matric head. In soil physics the matric head is typically described with $h_m$, while potentials are described with $\psi$.

Page 19, Line 23: "Niu (Niu et al.,2007)" should be Niu et al. (2007).

Consistently write $D_m$, $k_s$ with subscripted characters.

---

## Author Comment (AC1) · 3 Jun 2016

*The manuscript addresses an important issue in soil hydrology: The application of data assimilation methods to real world data, especially when estimating not only states, but also parameters. The authors state to address three main questions: (1) the performance of the data assimilation methods on Land Surface Models with real world data in general, (2) the differences in performance due to different data assimilation methods and (3) difference in performance due to different Land Surface Models. The study finds small differences due to data assimilation methods and large differences based on different Land Surface Models. These findings can give valuable insight for the applicability of data assimilation methods on Land Surface Models. But, this requires an adequate discussion of the used measurements, data assimilation methods and finally the results.*

*I have 3 major comments regarding each of these discussions, which fall short off answering the stated main questions enough. Additionally I have one major comment on the quality of the explanations for employed methods and models.*

**Reply: We thank the reviewer for pointing out the contribution of our work. We will revise the manuscript taking into account the comments.**

*Page 10, Lines 20-23: "... and a soil moisture and soil temperature sensor network (with measurements at 5, 20 and 50cm depth) are installed, amongst others. Soil moisture time series at 41 locations are being recorded." What kind of soil moisture sensors are installed? Please discuss possible uncertainties in the data.*

**Reply: The SPADE soil water content probes (sceme.de GmbH i.G., Horn-Bad Meinberg, Germany; (*Hübner et al*., 2009)) were installed at 5 cm, 20 cm and 50 cm depth along a vertical profile. The SPADE probe is a ring oscillator and the frequency of the oscillator is a function of the dielectric permittivity of the surrounding medium, which is strongly dependent on the soil water content because of the high permittivity of water ($\approx$80F/m) as compared to mineral soil solids ($\approx$2-9F/m), and air ($\approx$1F/m). The SPADE probe was calibrated according to the procedure outlined in [*Qu et al*., 2014]. The possible uncertainties in the soil moisture data are related to imperfect contact of the sensors with the soil, imperfection of the model which relates the sensor response and dielectric permittivity and imperfection of the model which relates dielectric permittivity and soil moisture. The measurement error is assumed to be $0.02cm^3/cm^3$. We will modify the text in the revised version of the manuscript to clarify this.**

*Page 10, Lines 24-35: "In this work, the Rollesbroich site is modeled as a single point and the data of the soil sensor network are averaged to calculate areal averages of soil moisture content at 5cm, 20cm and 50cm depth." Please discuss the importance and implications of this assumption. What are the expected impacts of heterogeneity?*

**Reply: Data assimilation experiments with land surface models are generally conducted for large scales, especially when remote sensing data are assimilated. Therefore it is important to evaluate the model performance at a larger scale. The Rollesbroich site has an area of $0.27km^2$, which is a very small catchment. *Qu et al.* (2014) described the statistics of soil properties for soil samples taken in the Rollesbroich catchment (see Table 1). We can see that soil texture shows a relatively limited variation. In our work only vertical heterogeneity is considered. In this case, heterogeneity does not seem to be very strong and we do not face a special challenging upscaling case for the land surface model. This will be clarified in the revised version of the manuscript.**

**Table 1. Descriptive statistics of soil properties on the basis of 273 soil samples in the Rollesbroich catchment. Table is from *Qu et al*., (2014).**

| | | Clay % | Sand % | Silt % | Bulk Density (g/cm$^3$) | Carbon Content (g/kg) | Porosity (cm$^3$/cm$^3$) |
|---|---|---|---|---|---|---|---|
| 5cm | mean | 18.99 | 19.90 | 61.10 | 0.94 | 54.47 | 0.65 |
| | std | 2.00 | 3.82 | 3.79 | 0.12 | 15.82 | 0.05 |
| 20cm | mean | 18.03 | 20.76 | 61.20 | 1.28 | 34.08 | 0.52 |

| | | | | | | |
|---|---|---|---|---|---|---|
| | std | 1.99 | 4.03 | 3.46 | 0.15 | 16.84 | 0.05 |
| 50cm | mean | 16.50 | 22.00 | 61.50 | 1.52 | 11.22 | 0.43 |
| | std | 2.40 | 5.68 | 4.53 | 0.16 | 6.01 | 0.06 |

*Page 29, Figure 3: The figure shows higher water contents closer to the surface. Please mention this and discuss reasons and implications.*

**Reply: The Rollesbroich catchment is a wet site with a yearly average precipitation around 1200mm. Regular precipitation events cause a wet surface layer. In addition, porosity of the upper soil layer is higher than for the deeper soil layers. This causes that during wet conditions soil moisture content is higher for the upper soil layer than for the deeper layer. It implies that at this site often we have a drainage flux from the top soil towards the aquifer (and drainage channels). This will be clarified in the revised version of the manuscript.**

*Page 11 Lines 34-36: "The soil moisture observation error is assumed to be normally distributed with mean equal to 0 and standard deviation equal to $0.02m^3/m^3$, for both VIC-3L and CLM." Please discuss why you assume this uncertainty, especially since it is a mean of 41 values.*

**Reply: We admit that $0.02m^3/m^3$ is a little larger than the uncertainty of the mean soil moisture content averaged over the 41 values. A larger observation error elevates potential problems with filter inbreeding. In addition, it adds flexibility in case of the presence of an observation bias or model structural error. We will add an explanation in the paper.**

*Page 11 Lines 33-34: "Precipitation was perturbed were perturbed by multiplicative error N(1,0.1) to represent the uncertainty of measured precipitation at the site." Please give a reason for this error. What is the assumed error for evaporation?*

**Reply: In the Rollesbroich catchment, precipitation was measured by a tipping bucket. Therefore only a measurement error was assumed, which is typically around 10% of the measured value [*Hodgkinson et al*., 2004]. In this work the variables which govern evapotranspiration (incoming shortwave and longwave radiation, air temperature, relative humidity, wind speed), were not perturbed. An explanation will be added in this paper.**

*Page 2, Lines 13-15: "This approach allows for joint estimation of the states and parameters while taking into explicit consideration model structural error and forcing data errors (Liu and Gupta, 2007)." This is correct, but it is not to the point, since the authors later set the model error to zero (see Page 11, Lines 36-37) and hence do not consider model structural errors. Please discuss this.*

**Reply: Yes, model structural error is not considered in our work, but parameter uncertainties and forcing uncertainties are considered and we assume that these capture in this case the model uncertainty. However, we agree that it can be expected that we have other model structural errors, for example in relation to the representation of photosynthesis. We will revise this part in our manuscript.**

*Page 6, Line 25 (Eq. 25): Please mention that the way R (and $y_t^i$) is described, you assume uncorrelated measurement errors.*

**Reply: We will add this information as suggested by the reviewer.**

*Page 7, Line 25 (Eq. 33): Please describe the implications of employing this method. How does the performance of the filter depend on the choice of initial uncertainty?*

**Reply: Filter inbreeding is a problem associated with EnKF. The ensemble spread may narrow down in the course of parameter estimation so that most of the ensemble members would become very close to the ensemble mean value, which is called filter inbreeding and which might cause filter divergence [*Franssen and Kinzelbach*, 2008; *Han et al.*, 2014; *Whitaker and Hamill*, 2012]. The approach according Eq. 33 has been proven to be an efficient method to avoid filter**

inbreeding [*Han et al.*, 2014; *Whitaker and Hamill*, 2012]. We will test the effect of initial uncertainties on the performance of EnKF, e.g. increasing the forcing error from 10% to 20%. The additional simulation results will be explained in the manuscript. If larger changes would be observed, a more detailed discussion will be included.

*Page 9, Line 13. How did you choose the tuning parameter s? How does it influence the performance? Please include the choice of s for the PF and initial uncertainties of the EnKF when comparing different methods.*

**Reply:** **The optimal tuning parameter *s* is hardly known in applications [*Yan et al.*, 2015]. It was set to 0.01 in some applications [*DeChant and Moradkhani*, 2012; *Plaza et al.*, 2012]. In our work, to keep particle spread, *s* was set to 0.1. We will test other values for parameter *s*, like 0.01, to see how it influences the performance. The additional simulation results will be explained in the manuscript. If larger changes would be observed, a more detailed discussion will be included.**

*Page 15 Line 32: You state: "It is not surprising that the EnKF is more efficient and effective than the PF." I would follow this statement in case of strictly Gaussian distributions and linear measurement operators. In those cases the EnKF is expected to outperform the PF. Nevertheless, when dealing with non-linear processes that challenge the Gaussian assumption, the better performance is not clear at all.*

**Reply:** **We added an explanation already in the manuscript why we think EnKF outperforms PF, even although the PF is in theory more suited for non-linear processes and non-Gaussian statistics. We will re-evaluate this part and try to formulate more precisely.**

*Page 16 Line 33: You state that EnKF and PF "differ fundamentally in their analysis step". I would disagree and argue that the analysis step is similar. The only difference is that the EnKF updates it's posterior based on the Gaussian assumption of the distributions, while the PF drops this assumption.*

**Reply:** **We still think that EnKF and PF differ fundamentally in their analysis step. In EnKF, the difference of the forecasted output variable(s) and corresponding observed value(s) (the part (y-Hx)) is used directly to update states and parameters. However, in PF, not the difference but the likelihood is calculated and the individual particles are not corrected towards the states, but only weighted differently in correspondence with the likelihood.**

*Page 25, Table 1: The EnKF assumes Gaussian distributions. What is the reason that you sample for all but one parameter from an initial uniform distribution? What are the implications for the chosen inflation method?*

**Reply:** **We want to compare EnKF and PF starting from the same prior distribution in order to make a more meaningful comparison. It is right that EnKF assumes a Gaussian distribution, but the PF not. We believe that assuming an initial uniform distribution is a neutral assumption good for comparing EnKF and PF. We will add an explanation in the paper.**

*Page 16, Lines 1-2: You state: "The value of the likelihood does generally not say anything about how close the forecasted variables are to their measured counterparts." I disagree, the likelihood yields information about the distance.*

**Reply:** **The likelihood yields information about the relative, but not the absolute distance between forecasted variables and measurements. We can say that particles with higher weights are closer to the measurements but we cannot tell how close they are to the measurements.**

*Discussion of results: Since the results show a wealth of information. I would appreciate a detailed discussion. Especially consider the following points in detail and incorporate them into your conclusion:*

*Page 29, Figure 3: The large deviations in the Particle Filter might hint at filter inbreeding in the states. You observe the deviation but do not discuss it and actually exclude the possibility of inbreeding by investigating the parameters: "A too narrow spread of ensemble members would lead to filter divergence. For the state augmentation (AUG) and dual estimation (DUAL), the spread of the ensemble members is kept large enough during the whole assimilation period as the ensemble inflation method helped to keep adequate ensemble spread. RRPF and MCMCPF also have enough ensemble spread because of parameter perturbation and MCMCPF resampling." (Page 13 Lines 15-19). Please address the question of adequate ensemble spread in the states by actually showing the ensemble of states there.*

**Reply: We will show the evolution of the state ensemble in our manuscript and address this question.**

*Page 30 Figure 4: During the calibration period the filter without parameter estimation performs better. Please discuss possible reasons.*

**Reply: When only states (soil moisture content) are assimilated, states are updated directly by observations. However, when states and parameters are updated jointly, the nonlinear relation between states and parameters is considered which may introduce inconsistency. We will further discuss these results in the paper.**

*Page 31 Figure 5: Parameter b estimated by MCMC shows a large difference to the other methods. But MCMC does perform approximately as well as the other filters (Figure 7, Page 33). Why is there no difference? Please discuss.*

**Reply: *Demaria et al.* (2007) evaluated the sensitivity and identifiability of ten parameters which control surface and subsurface runoff in the VIC model for four U.S. watersheds along a hydroclimatic gradient. They found that parameter b is crucial in a dry environment, while its impact on model performance is not significant in wet sites. They concluded that parameter b plays a key role in partitioning rainfall into soil moisture and surface runoff in dry environments. [*Liang and Guo*, 2003] and [*Atkinson et al.*, 2002] have a similar conclusion. In our work, the Rollesbroich catchment is very wet, even though parameter b estimated by MCMC shows a large difference with other methods, it shows small impact on the soil moisture content for layer 1 and layer 2. An explanation will be added in the paper.**

*Page 32 Figure 6: Initial parameter uncertainties are the same for PF and EnKF but at time 0 the ensemble spreads are different. Please explain. You only show the two parameters with the least change over time. Give a reason or show the one with the smallest and the one with the largest changes.*

**Reply: Figure 6 shows the evolution of the parameter ensemble from time step 1 but not time step 0. At time step 0, the ensemble spreads are the same, but at time step 1, the parameter ensemble is updated by PF or EnKF, and the ensemble spreads between EnKF and PF differ. We will show the evolution of the parameter ensemble from time step 0 onwards in the revised version of the manuscript. We think that the saturated hydraulic conductivity $\log_{10} k_s$ and the model parameter $\beta$ are the two most important parameters for the VIC Model for the Rollesbroich catchment, so they are shown in the figure.**

*Page 34, Figure 8: There is basically no difference for the prediction of the water content whether there are parameters estimated or not. Please discuss why this is the case.*

**Reply: Predictions of soil moisture content for layer 2 and layer 3 (in the verification period) improved significantly for the case of parameter estimation. Concerning the soil moisture**

content of layer 1, the RMSE value of the open loop run is $0.053\text{m}^3/\text{m}^3$, which is already quite close to the observed values. In addition, the soil moisture content for the upper layer is strongly driven by single precipitation events. We will extend the discussion of these results.

*Page 37 Figure 11: You do not show the parameters from the augmented state. Instead you show parameters derived from those in a non-linear way. Please discuss this.*

**Reply: We showed on purpose the soil hydraulic parameters $k_s$ and B, which are in CLM calculated from soil texture. We believe that displaying these parameters is more meaningful than soil texture.**

*Figures 7, 8, 12, 13: Especially water content of the top layer can almost not be represented by either model, although improved with state of the art data assimilation methods. Please discuss this including the representation of the physics in the models and implications of the perfect model assumption.*

**Reply: We will provide additional discussion of the results and discuss reasons for the larger deviations in the fit.**

*The difference in the assimilation methods is small. Please discuss if this difference is significant. Do you expect the same results for other applications? What is the influence of specific filter settings on the performance?*

**Reply: The performance of the four data assimilation algorithms in which states and parameters are jointly estimated does not differ very much in our study. Nevertheless, the small difference in performance between EnKF and PF based algorithms indicates that PF is also an efficient data assimilation algorithm for problems of this size. It can be expected that larger ensemble numbers can improve the performance of EnKF and PF based algorithms. For MCMCPF, multiple MCMC resampling steps can also help improve performance. Given the CPU-intensity of the calculations a larger comparison is beyond the scope of this work. In our work, all algorithms are evaluated with real-world data, so we think our results are meaningful for other applications. We expect that for example with more unknowns (i.e., 2D and 3D-applications) EnKF-based algorithms will perform better than PF, as PF will become extremely CPU-intensive and needs many more particles. We will evaluate the impact of filter settings on the results of this paper. However, it will be difficult to evaluate the significance of the difference between the DA-algorithms on the basis of this single study.**

*Please improve the explanations on the Particle Filter and the two LSMs. Page 7 Line 32 - Page 8 Line 11: I do understand Particle Filters, but the given explanation is not clear. Especially clarify your description of transition and proposal densities.*

**Reply: We will revise this part in our manuscript.**

*Page 8 Lines 21-30: You describe SIR and RR. What is the reason to choose RR?*

**Reply: RR is developed from SIR by [*Liu and Chen*, 1998]. RR is one of the most popular methods for PF to reduce particle degeneration. For RR, the variance of particles is smaller than the one given by the SIR scheme. Moreover, RR is computationally cheaper than SIR. This explanation will be added in the revised version of the manuscript.**

*Page 7 Lines 31-32: "The particle filter was first suggested in the research area of object recognition, robotics and target tracking (Arulampalam et al., 2002)." The PF was actually mentioned earlier (Gordon et al., 1993).*

**Reply: This will be corrected in the revised version.**

*For this study the understanding of the different LSM models is important, since the main difference in performance is attributed to different models. Because of that a good description is necessary. Although I am not an expert on LSMs, I noticed the following: Page 17, Line 15 (Eq. A2): The equation describes the soil water movement in the top two layers. To me it was not clear if this description is valid for both layers individually. If so, why is the precipitation P and evaporation E the same?*

**Reply: This will be corrected in the revised version.**

*Page 18, Line 1 (Eq. A7): The dimensions are inconsistent: You add [LT-1], [] and [L]. $i_m$ should be $I_m$ (or is not introduced). Please also explain the distinction of the cases $P+I < I_m$ and $P+I > I_m$.*

**Reply: $i_m$ is $I_m$. We will revise this part in our manuscript. When $P+I > I_m$, the upper soil layers (layer 1 and layer 2) will be saturated. When $P+I < I_m$, the upper soil layers are assumed unsaturated, and infiltration capacity is variable, as clarified in Eq. A9.**

*Page 19, Lines 18-22: the Richards equation is formulated for a continuum. Please explain the modifications and the implications of applying it to layers. Please give the extent of these layers.*

**Reply: The unmodified Richards equation is:**

**The soil water flux *q* can be described by Darcy's law:**

$$q = -k\left(\frac{\partial(\psi+z)}{\partial z}\right) \tag{1}$$

**For one-dimensional vertical water flow in soils, the conservation of mass is stated as:**

$$\frac{\partial\theta}{\partial t} = -\frac{\partial q}{\partial z} - E = \frac{\partial}{\partial z}\left[k\left(\frac{\partial(\psi+z)}{\partial z}\right)\right] - E \tag{2}$$

*E is the ET loss. Zeng and Decker* **(2009) note that this equation cannot maintain the hydrostatic equilibrium soil moisture distribution because of the truncation errors of the finite-difference numerical scheme. They show that this deficiency can be overcome by subtracting the equilibrium state as:**

$$\frac{\partial\theta}{\partial t} = \frac{\partial}{\partial z}\left[k\left(\frac{\partial(\psi+z-C)}{\partial z}\right)\right] - E \tag{3}$$

**Where C is a constant hydraulic potential above the water table $z_\nabla$:**

$$C = \psi_E + z \tag{4}$$

**Substitution of equation (4) into equation (3) yields the modified Richards equation (B7):**

$$q = -k\left(\frac{\partial(\psi-\psi_E)}{\partial z}\right) \quad \text{and} \quad \frac{\partial\theta}{\partial t} = -\frac{\partial q}{\partial z} - E = \frac{\partial}{\partial z}\left[k\left(\frac{\partial(\psi-\psi_E)}{\partial z}\right)\right] - E \tag{5}$$

**Implication of this modified method: Richards equation (2) used a θ-based solution which cannot account for the variation of ψ below water table because θ is constant (at saturated value)**

while $\psi$ varies temporally and spatially, which leads to the failure to maintain the hydrostatic equilibrium soil moisture distribution. However, the modified Richards equation in which constant hydraulic potential C is explicitly subtracted at each time step can fix this deficiency. Details about the implementation of the modified method can be seen in [*Zeng and Decker*, 2009].

Below we explain the application of the modifications to layers. These details would not be presented in the revised manuscript and would be referred to the CLM-manual (Oleson et al., 2013) where more details can be found. Table 2 shows the soil layer definition in CLM for $N_{levsoi}$ =10 layers.

Table 2. Soil layer definition where soil moisture is calculated in CLM. Layer node depth($z$), thickness($\Delta z$), and depth at layer interface($z_h$) for 10 soil layers. Unit is meter.

| Layer $i$ | $z$ | $\Delta z$ | $z_h$ |
|---|---|---|---|
| 1 (top) | 0.0071 | 0.0175 | 0.0175 |
| 2 | 0.0279 | 0.0276 | 0.0451 |
| 3 | 0.0623 | 0.0455 | 0.0906 |
| 4 | 0.1189 | 0.0750 | 0.1655 |
| 5 | 0.2122 | 0.1236 | 0.2891 |
| 6 | 0.3661 | 0.2038 | 0.4929 |
| 7 | 0.6198 | 0.3360 | 0.8289 |
| 8 | 1.0380 | 0.5539 | 1.3828 |
| 9 | 1.7276 | 0.9133 | 2.2961 |
| 10 | 2.8646 | 1.5058 | 3.8019 |

Numerical solution of equation (5) to soil layer $i$:

$$\Delta z_i \frac{\Delta \theta_i}{\Delta t} = -q_{i-1}^{t+1} + q_i^{t+1} - e_i \qquad (6)$$

where $t$ is the time step, $\Delta z_i$ is soil layer thickness, $\Delta \theta_i = \theta_i^{t+1} - \theta_i^t$, $q_i$ is the outgoing flux of water from layer $i$ to layer $i+1$, $q_{i-1}$ is the incoming flux of water from layer $i-1$ to layer $i$ and $e_i$ is layer-averaged ET loss. The water fluxes ($q_{i-1}^{t+1}$ and $q_i^{t+1}$) in equation (6) are linearized about $\theta$ using Taylor series expansion which results in a general tridiagonal equation set of the form [*Oleson et al.*, 2013]:

$$r_i = a_i \Delta \theta_{i-1} + b_i \Delta \theta_i + c_i \Delta \theta_{i+1} \qquad (7)$$

$a_i$, $b_i$, $c_i$ and $r_i$ will be discussed under different conditions below. This tridiagonal equation set is solved over $i=1,…, N_{levsoi}+1$ where $i= N_{levsoi}+1$ is a virtual layer representing the aquifer.

When $i=1$, the boundary condition is the infiltration rate, and

$a_i = 0$

$b_i = \frac{\partial q_i}{\partial \theta_i} - \frac{\Delta z_i}{\Delta t}$

$c_i = \frac{\partial q_i}{\partial \theta_{i+1}}$

$r_i = q_{infl}^{t+1} - q_i^t + e_i$

$q_{infl}^{t+1}$ is the infiltration into the soil which is partitioned between water input flux (sum of precipitation reaching the ground and melt water from snow), surface runoff, and surface water storage.

**When $i=2, \ldots, N_{levsoi} - 1$,**

$$a_i = -\frac{\partial q_{i-1}}{\partial \theta_{i-1}}$$

$$b_i = \frac{\partial q_i}{\partial \theta_i} - \frac{\partial q_{i-1}}{\partial \theta_i} - \frac{\Delta z_i}{\Delta t}$$

$$c_i = \frac{\partial q_i}{\partial \theta_{i+1}}$$

$$r_i = q_{i-1}^t - q_i^t + e_i$$

**For the lowest soil layer ($i = N_{levsoi}$), the bottom boundary condition depends on the depth of the water table. If the water table is within the soil column, a zero flux boundary condition is applied ($q_i^t = 0$) and**

$$a_i = -\frac{\partial q_{i-1}}{\partial \theta_{i-1}}$$

$$b_i = -\frac{\partial q_{i-1}}{\partial \theta_i} - \frac{\Delta z_i}{\Delta t}$$

$$c_i = 0$$

$$r_i = q_{i-1}^t + e_i$$

**And for the aquifer layer $i = N_{levsoi} + 1$:**

$$a_i = 0 \quad b_i = -\frac{\Delta z_i}{\Delta t} \quad c_i = 0 \quad r_i = 0$$

**If water table is below the soil column, for $i = N_{levsoi}$:**

$$a_i = -\frac{\partial q_{i-1}}{\partial \theta_{i-1}}$$

$$b_i = \frac{\partial q_i}{\partial \theta_i} - \frac{\partial q_{i-1}}{\partial \theta_i} - \frac{\Delta z_i}{\Delta t}$$

$$c_i = \frac{\partial q_i}{\partial \theta_{i+1}}$$

$$r_i = q_{i-1}^t - q_i^t + e_i$$

**And for the aquifer layer $i = N_{levsoi} + 1$:**

$$a_i = -\frac{\partial q_{i-1}}{\partial \theta_{i-1}}$$

$$b_i = -\frac{\partial q_{i-1}}{\partial \theta_i} - \frac{\Delta z_i}{\Delta t}$$

$c_i = 0$

$r_i = q_{i-1}^t$

**Upon solution of the tridiagonal equation set, soil water content is updated as:**

$$\theta_i^{t+1} = \theta_i^t + \Delta\theta_i \Delta z_i \qquad\qquad\qquad (8)$$

*Specific comments: Page 5, Line 36: "Commonly used data assimilation algorithms are EnKF, PF and variants of them." 4D-Var is also commonly used.*

**Reply: We admit that 4D-Var is also commonly used. The sentence will be adapted to account for this, and we will specify shortly in which contexts the different methods are normally used.**

*Page 6, Line 19: "H [...] is the identity matrix if y refers to in-situ ground measurements available at all grid cells." This is only the case if the same quantity as the state is observed. Otherwise the quantity has to be transferred. Additionally mention that H has to be linear for the EnKF.*

**Reply: We will revise and extend this part in our manuscript.**

*Page 6-7: You do not mention the use of a damping factor (Hendricks Franssen and Kinzelbach, 2008) for the parameters. Is there a specific reason you do not employ it?*

**Reply: The filter inbreeding problem could be reduced with help of a damping factor, which limits the intensity of the perturbation of the parameters [*Franssen and Kinzelbach*, 2008]. In our work, the inflation algorithm proposed by *Whitaker and Hamill* (2012) was applied to the ensemble of parameters to reduce filter inbreeding.**

*Page 25, Table 1: It is not clear which parameters are estimated for each layer individually and which are estimated for the entire profile.*

**Reply: We will clarify this in table 1.**

*Page 11 Lines 11-12: "The parameters of the other layers were updated with help of the calculated spatial covariance in case of EnKF." This statement implies that the parameters were not estimated with the PF? Please clarify.*

**Reply: Parameters are also estimated in PF. In PF, each particle includes a state vector and a parameter vector. But the weight vector is calculated only by the state vector. Both the state vector and parameter vector are resampled with help of the weight vector. We will reformulate the text to make this clearer.**

*Page 11 Line 30: "EnKF with state updating only was tested for (2)." Why did you not test state updating with the PF?*

**Reply: In our work, parameters are perturbed to generate particles for PF, which means each particle includes a state vector and a parameter vector. After calculating the weight vector for the state vector, the resampling is (automatically) applied for both state vector and parameter vector. In this context, it is not possible to only update states. If we would have deterministic parameters, it would be possible to only update states with PF, but then not a comparison with EnKF could be made.**

*Page 12 Lines 2-10: Why do you need 2 different characterizations of the uncertainty. What is the additional gain by showing NSE?*

**Reply: Because RMSE values are usually affected by both a mean bias and random variations, NSE is added as another measure [*Han et al.*, 2012]. NSE values represent the correlation between the estimation and the observation.**

*Page 12 Line 9-10: "A NSE value equal to 1 and RMSE equal to 0 imply a perfect prediction." This is wrong. A RMSE equal to the measurement uncertainty is perfect.*

**Reply: Thanks, we admit that for a prediction in the verification phase we cannot expect a better result than a RMSE equal to the measurement uncertainty but still an RMSE equal to 0 would be the best result. We will modify the sentence to accommodate this result.**

**All Technical corrections are applied in our revised manuscript.**

**Reference**

Atkinson, S. E., R. A. Woods, and M. Sivapalan (2002), Climate and landscape controls on water balance model complexity over changing timescales, *Water Resources Research*, *38*(12), doi:10.1029/2002wr001487.

DeChant, C. M., and H. Moradkhani (2012), Examining the effectiveness and robustness of sequential data assimilation methods for quantification of uncertainty in hydrologic forecasting, *Water Resources Research*, *48*, doi:10.1029/2011wr011011.

Demaria, E. M., B. Nijssen, and T. Wagener (2007), Monte Carlo sensitivity analysis of land surface parameters using the Variable Infiltration Capacity model, *Journal of Geophysical Research-Atmospheres*, *112*(D11), 15, doi:10.1029/2006jd007534.

Franssen, H. J. H., and W. Kinzelbach (2008), Real-time groundwater flow modeling with the Ensemble Kalman Filter: Joint estimation of states and parameters and the filter inbreeding problem, *Water Resources Research*, *44*(9), doi:10.1029/2007wr006505.

Han, X., H. J. H. Franssen, C. Montzka, and H. Vereecken (2014), soil moisture and soil properties estimation in the community Land Model with synthetic brightness temperature, *Water Resource Research*, doi:10.1002/2013WR014586.

Han, X., X. Li, H. J. H. Franssen, H. Vereecken, and C. Montzka (2012), Spatial horizontal correlation characteristics in the land data assimilation of soil moisture, *Hydrol. Earth Syst. Sci.*, *16*(5), 1349-1363, doi:10.5194/hess-16-1349-2012.

Hodgkinson R. A., T. J. Pepper, and D. W. Wilson (2004), Evaluation of tipping bucket rain gauge performance and data quality, Environment Agency, Rio House, Waterside Drive, Aztec West, Almondsbury, Bristol, BS32 4UD, ISBN:1844323242.

Hübner, C., R. Cardell-Oliver, R. Becker, K. Spohrer, K. Jotter, and T. Wagenknecht (2009), Wireless soil moisture sensor networks for environmental monitoring and vineyard irrigation: Helsinki University of Technology, 1, 408-415.

Liang, X., and J. Z. Guo (2003), Intercomparison of land-surface parameterization schemes: sensitivity of surface energy and water fluxes to model parameters, *Journal of Hydrology*, *279*(1-4), 182-209, doi:10.1016/s0022-1694(03)00168-9.

Liu, and R. Chen (1998), Sequential Monte Carlo methods for dynamic systems, *Journal of the American Statistical Association*, *93*(443), 1032-1044, doi:10.2307/2669847.

Oleson, K., et al. (2013), Technical description of version 4.5 of the Community Land Model (CLM), NCAR Technical Note NCAR/TN-503+STR, 422, doi:10.5065/D6RR1W7M.

Plaza, D. A., R. De Keyser, G. J. M. De Lannoy, L. Giustarini, P. Matgen, and V. R. N. Pauwels (2012), The importance of parameter resampling for soil moisture data assimilation into hydrologic models using the particle filter, *Hydrol. Earth Syst. Sci.*, *16*(2), 375-390, doi:10.5194/hess-16-375-2012.

Qu, B. H. R., H. J. A., M. G., P. Y. A., and V. H. (2014), Effects of soil hydraulic properties on the spatial variability of soil water content: evidence from sensor network data and inverse modeling, *Vadose Zone Journal*, *13*(12), doi:10.2136/vzj2014.07.0099.

Whitaker, J. S., and T. M. Hamill (2012), Evaluating Methods to Account for System Errors in Ensemble Data Assimilation, *Monthly Weather Review*, *140*(9), 3078-3089, doi:10.1175/mwr-d-11-00276.1.

Yan, H., C. M. DeChant, and H. Moradkhani (2015), Improving Soil Moisture Profile Prediction With the Particle Filter-Markov Chain Monte Carlo Method, *Ieee Transactions on Geoscience and Remote Sensing*, *53*(11), 6134-6147, doi:10.1109/tgrs.2015.2432067.

Zeng, X., and M. Decker (2009), Improving the numerical solution of soil moisture-based Richards equation for land models with a deep or shallow water table, *Journal of Hydrometeorology*, *10*, 308-319, doi:10.1175/2008JHM1011.1.

---

## Referee Comment (RC2) · Anonymous Referee #2 · 14 Jul 2016

The manuscript presents different data assimilation methods for a joint estimation of soil moisture states and model parameters for the VIC hydraulic model and the Community Land Model. The models were tuned and evaluated at a single site and the main objectives include the advantages of the joint state and parameter estimation incorporating real data from the field, performance of the DA methods as well as the different land surface schemes.

**General comments**

The topic is interesting for the scientific community and the paper is clearly structured and well written. I agree for the most part with referee #1, who emphasizes shortcomings with respect to the presentation and discussion of the given objectives, to which I will add only a few more comments.

Furthermore, I have a comment regarding the usefulness of the presented data assimilation techniques for land surface modeller. In my opinion, the merits of a joint state and parameter estimation should not only be discussed with respect to DA schemes updating states only, but also with respect to alternative methods like conventional bayesian interference, e.g. (Yang et al., 2008), as well as the issue of optimizing time-invariant parameter vs. time-variant parameter, which has been intensely studied in the group of the authors (Vrugt et al., 2005, 2013). Therefore a discussion of the following (which might be too obvious for the authors to mention) can help to increase the significance for a broader community: What are the advantages of the joint parameter estimation versus optimizing time-invariant parameter? There seem to be shortcomings as time-variant parameter may be highly dependent on the end of the training sequence, especially when it ended shortly after a large precipitation event, like in this study. Will the parameter converge in the given training data set of 5 months? Vrugt et al. (2013) show that time-variant parameter can exhibit considerable non-stationarity, which is caused by changing sensitivity of the target variable on the parameters. Is there a difference/advantage of the joint estimation with time-variant parameter in terms of equifinality/identifiability of the parameter?

[Figure]

**Specific comments**

- p.10, ll.28ff: For me there seems to be no need to show the spin-up time series (Figure 2). Precipitation and temperature of the assimilation and verification period seem to be enough.

- p.11: What is the reason of choosing July 31 as the date to switch from assimilation to verification period? This choice seems to be critical for me, as the parameter of the final time step are chosen for the verification period. What would be the impact, if e.g. July 20 would have been chosen, as Figure 5 suggests, that some parameter for the MCMCPF method were significantly different?

- p.11: state updating only: How does the model then learn for the verification period? How are the parameter chosen in this case? Please describe this more clearly.

- p.11, ll.34ff, Table 1: soil moisture observation errors and parameter perturbations are given by normal and uniform distributions and corresponding ranges, means and standard deviations are given with numbers without further reasoning. As a comprehensive set of soil moisture measurements and soil core data is available, I would assume, the range of perturbation is related to the measured distributions, but I did not see a hint in the text. Referee #1 already addressed this issue related to measurement uncertainty and spatial heterogeneity, and the authors gave detailed reply, but I still miss, how the prior distributions and measurement uncertainties are related to the measured pdfs. It is surprising for me, that the uncertainty of the soil moisture measurements related to spatial heterogeneity is smaller than the given instrument uncertainty of $\pm 0.02 m^3 m^{-3}$.

- p 13. ll3-5: You state: *"Even although the soil moisture time series for the state augmentation and dual estimation method are very similar, the temporal evolution of their parameter values are different"*. This hints at the issue of equifinality and

identifiability of the parameters with respect to the time series to be predicted. Please discuss this problem.

**Technical corrections**

- Figures 3,7,9,12: Legend: "OBS" were coded with 2 dots. Please make use of different line types for a better discrimination between the displayed series. Especially red and green will be indistinguishable for many readers

**References**

Vrugt, J. A., Diks, C. G. H., Gupta, H. V., Bouten, W., and Verstraten, J. M.: Improved treatment of uncertainty in hydrologic modeling: Combining the strengths of global optimization and data assimilation, Water Resour. Res., 41, W01 017, doi:10.1029/2004WR003059, 2005.

Vrugt, J. A., ter Braak, C. J., Diks, C. G., and Schoups, G.: Hydrologic data assimilation using particle Markov chain Monte Carlo simulation: Theory, concepts and applications, Adv. Water Resour., 51, 457–478, doi:10.1016/j.advwatres.2012.04.002, 35th Year Anniversary Issue, 2013.

Yang, J., Reichert, P., Abbaspour, K., Xia, J., and Yang, H.: Comparing uncertainty analysis techniques for a SWAT application to the Chaohe Basin in China, J. Hydrol., 358, 1–23, doi:10.1016/j.jhydrol.2008.05.012, 2008.

---

## Author Comment (AC2) · 2 Aug 2016

*The manuscript presents different data assimilation methods for a joint estimation of soil moisture states and model parameters for the VIC hydraulic model and the Community Land Model. The models were tuned and evaluated at a single site and the main objectives include the advantages of the joint state and parameter estimation incorporating real data from the field, performance of the DA methods as well as the different land surface schemes.*

*The topic is interesting for the scientific community and the paper is clearly structured and well written. I agree for the most part with referee #1, who emphasizes shortcomings with respect to the presentation and discussion of the given objectives, to which I will add only a few more comments.*

**Reply:** **We thank the reviewer for pointing out the contribution of our work. We will revise the manuscript taking into account the comments.**

*Furthermore, I have a comment regarding the usefulness of the presented data assimilation techniques for land surface modeller. In my opinion, the merits of a joint state and parameter estimation should not only be discussed with respect to DA schemes updating states only, but also with respect to alternative methods like conventional Bayesian interference, e.g. (Yang et al., 2008), as well as the issue of optimizing time-invariant parameter vs. time-variant parameter, which has been intensely studied in the group of the authors (Vrugt et al., 2005, 2013). Therefore a discussion of the following (which might be too obvious for the authors to mention) can help to increase the significance for a broader community: What are the advantages of the joint parameter estimation versus optimizing time-invariant parameter? There seem to be shortcomings as time-variant parameter may be highly dependent on the end of the training sequence, especially when it ended shortly after a large precipitation event, like in this study. Will the parameter converge in the given training data set of 5 months? Vrugt et al. (2013) show that time-variant parameter can exhibit considerable non-stationarity, which is caused by changing sensitivity of the target variable on the parameters. Is there a difference/advantage of the joint estimation with time-variant parameter in terms of equifinality/identifiability of the parameter?*

**Reply:** **Yes, we agree that there are other relevant methods for parameter estimation/calibration of hydrologic models, for example Bayesian recursive estimation [Thiemann et al., 2001], particle swarm optimization [Scheerlinck et al., 2009] and differential evolution adaptive metropolis [Vrugt and Ter Braak, 2011]. However these methods require in general a large number of model evolutions, which is often prohibitive for large scale land surface models. We refer therefore in the revised version of the manuscript shortly to alternative methods and point to the limitations of those methods.**

**Generally, parameters are time variant when jointly estimated with state variables as they are updated at each assimilation time step. It is true that time-variant parameters may be dependent on the end of the training sequence, especially for the parameters which are very sensitive to model forcings. The fact that we replace heterogeneous soil properties and soil moisture content for a given area by spatially homogeneous values, also introduces temporal variability in the effective parameters that are estimated in this study. In this context, it can be expected that estimated parameters show temporal evolution. Uncertainties and errors in model forcings and model structural errors will introduce additional temporal fluctuation of estimated parameter values. In a batch calibration approach, these temporal parameter variations will be averaged out and parameters are estimated which on average perform better over the period of consideration. The advantage of sequential data assimilation is that parameter estimation is faster whereas temporal parameter variations in some cases are meaningful. Kurtz et al. (2012) were successful in estimating a temporal variable parameter with EnKF, but concluded that the algorithm needs time to adjust to new parameter values. Vrugt et al. (2013) found considerable temporal non-stationarity in parameters estimated by McMC-PF.**

**This will be discussed in the revised version of the manuscript. We will point to both limitations and advantages of estimating temporally variable parameters, which depend on the data assimilation algorithm used, and also on different types of errors being involved. It is important**

to notice that especially for EnKF, parameters converged towards more stable values at the end of the assimilation period. .

*P.10, ll.28ff: For me there seems to be no need to show the spin-up time series (Figure 2). Precipitation and temperature of the assimilation and verification period seem to be enough.*

**Reply: We will revise this part in our revised manuscript.**

*P.11: What is the reason of choosing July 31 as the date to switch from assimilation to verification period? This choice seems to be critical for me, as the parameters of the final time step are chosen for the verification period. What would be the impact, if e.g. July 20 would have been chosen, as Figure 5 suggests, that some parameter for the MCMCPF method were significantly different?*

**Reply: For 2013, there are issues with a large number of sensors in the area and the mean soil moisture content would have to be estimated from fewer (and different) sensors. We started the assimilation in March 2012 as in the winter before soil moisture content readings were affected by soil freezing and therefore unreliable (at least in February). We will test the impact of the choice of the last assimilation day on the parameter estimation with the MCMCPF method and discuss this issue in the revised manuscript.**

*P.11: state updating only: How does the model then learn for the verification period? How are the parameter chosen in this case? Please describe this more clearly.*

**Reply: when only the state is updated in the assimilation period, the model gets more accurate initial state conditions in the verification period. We would indeed expect that an improved characterization of initial states has some positive impact during the first weeks, but vanishes over time. We will address this point more clearly in the revised manuscript.**

*P.11, ll.34ff, Table 1: soil moisture observation errors and parameter perturbations are given by normal and uniform distributions and corresponding ranges, means and standard deviations are given with numbers without further reasoning. As a comprehensive set of soil moisture measurements and soil core data is available, I would assume, the range of perturbation is related to the measured distributions, but I did not see a hint in the text. Referee #1 already addressed this issue related to measurement uncertainty and spatial heterogeneity, and the authors gave detailed reply, but I still miss, how the prior distributions and measurement uncertainties are related to the measured pdfs. It is surprising for me, that the uncertainty of the soil moisture measurements related to spatial heterogeneity is smaller than the given instrument uncertainty of $0.02 m^3/m^3$.*

**Reply: For the CLM model parameters, the parameter perturbations are taken from Han et al. (2014), and for the model parameter perturbations for VIC, we refer to Demaria et al. (2007) and Troy et al. (2008). Also measurements were available at the Rollesbroich site to estimate parameter uncertainty. In particular, soil texture measurements are available. If we calculate the uncertainty of the mean soil texture based on those data, we get very small uncertainties. The range of parameter perturbations should be large enough to have enough spread among the state ensemble members, which helps for better assimilation performance. In this case, the uncertainty has to be increased in order to fit the data. This is related to the fact that ultimately soil hydraulic parameters, and not soil texture, are important for calculating water and energy fluxes in the soil. The pedotransfer functions which are used to relate soil texture and soil hydraulic parameters are also subject to uncertainty. We therefore did not use directly the uncertainty of the soil texture estimated from the measurements, but increased it. In the revised version of the manuscript we will give details on this important point.**

**Qu et al. (2014) calculated the root mean square error (RMSE) associated with soil water content estimation, which is 0.026 $m^3/m^3$, after the two-step calibration procedure for this catchment. The uncertainty of the mean soil moisture content, assuming a Gaussian distribution, from 41 measurements in this catchment was $\frac{0.026}{\sqrt{41}}$ $m^3/m^3$. However, in our study we used**

$0.02\text{m}^3/\text{m}^3$ as observation error, because a larger observation error elevates problems with filter inbreeding. We found also in this case that the small measurement error estimated from the data was too small for our purposes and we will add discussion in the revised version of the manuscript to discuss this.

*P 13. ll3-5: You state: "Even although the soil moisture time series for the state augmentation and dual estimation method are very similar, the temporal evolution of their parameter values are different". This hints at the issue of equifinality and identifiability of the parameters with respect to the time series to be predicted. Please discuss this problem.*

**Reply:** **Equifinality is handled by both methods because not a single best solution is calculated but an ensemble of different solutions, which are all compatible with the measurement data. The ensemble mean values are plotted. The updating of the parameters follows for both methods the same general tendency. However, as the reviewer stresses, the ensemble mean values also differ for the two assimilation methods. We believe that in this case differences are related to the assimilation methods. The land surface model is ran twice in EnKF in case of dual estimation but only once for the augmentation approach. Model structure errors and biases "contribute" to different extents to parameter updating by these two data assimilation methods. Therefore the temporal evolution of parameter values is different. Discussion will be added in the paper.**

*Figures 3,7,9,12: Legend: "OBS" were coded with 2 dots. Please make use of different line types for a better discrimination between the displayed series. Especially red and green will be indistinguishable for many readers*

**Reply:** **We will revise this in the revised manuscript.**

**Reference**

Demaria, E. M., B. Nijssen, and T. Wagener (2007), Monte Carlo sensitivity analysis of land surface parameters using the Variable Infiltration Capacity model, Journal of Geophysical Research-Atmospheres, 112(D11), 15, doi:10.1029/2006jd007534.

Han, X., H. J. H. Franssen, C. Montzka, and H. Vereecken (2014), Soil moisture and soil properties estimation in the community Land Model with synthetic brightness temperature, Water Resource Research, doi:10.1002/2013WR014586.

Kurtz, W., H. J. H. Franssen, and H. Vereecken (2012), Identification of time-variant river bed properties with the ensemble Kalman filter, Water Resource Research, 48, W10534, doi:10.1029/2011WR011743.

Qu, W., H. R. Bogena, J. A. Huisman, G. Martinez, Y. A. Pachepsky, and H. Vereecken (2014), Effects of soil hydraulic properties on the spatial variability of soil water content: evidence from sensor network data and inverse modeling, Vadose Zone Journal, 13(12), doi:10.2136/vzj2014.07.0099.

Scheerlinck, K., V. R. N. Pauwels, H. Vernieuwe, and B. De Baets (2009), Calibration of a water and energy balance model: Recursive parameter estimation versus particle swarm optimization, Water Resource Research, 45, W10422, doi:10.1029/2009WR008051.

Shi, Y., K. J. Davis, F. Zhang and C. J. Duffy, and X. Yu (2015), Parameter estimation of a physically-based land surface hydrologic model using an ensemble Kalman filter: A multivariate real-data experiment, Advances in Water Resources, doi: 10.1016/j.advwatres.2015.06.009.

Thiemann, M., M. Trosset, H. Gupta, and S. Sorooshian (2001), Bayesian recursive parameter estimation for hydrologic models, Water Resource Research, 37(10), 2521–2535.

Troy, T. J., E. F. Wood, and J. Sheffield (2008), An efficient calibration method for continental-scale land surface modeling, Water Resources Research, 44(9), doi:10.1029/2007wr006513.

Vrugt, J. A., and C. J. F. Ter Braak (2011), DREAM(D): An adaptive Markov Chain Monte Carlo simulation algorithm to solve discrete, noncontinuous, and combinatorial posterior parameter estimation problems, Hydrology Earth System Sciences, 15(12), 3701–3713.

Vrugt, J. A., C. J. F. ter Braak, C. G. H. Diks, and G. Schoups (2013), Hydrologic data assimilation using particle Markov chain Monte Carlo simulation: Theory, concepts and applications, Advances in Water Resources, 51, 457-478, doi:10.1016/j.advwatres.2012.04.002.

---

## Referee Report (RR1)

I appreciate the detailed answers and the additionally performed simulations to investigate the raised issues. However, I still have few comments regarding answers, which were inconsistent or not sufficient to me. I would like to see them clarified:

**Major comments:**
1. The expected impact of heterogeneity is now explained in line 376-380: "Qu et al. (2014) described the statistics of soil properties for soil samples taken in the Rollesbroich catchment. Soil texture showed a relatively limited variation. In our work only vertical heterogeneity is considered. In this case, heterogeneity does not seem to be very strong and we do not face a challenging upscaling case for the land surface model." However, Qu et al. (2014) seem to state that there is considerable heterogeneity, e.g.: "Spatial variability of the measured soil water content was higher at the 50-cm depth than the 5- and 20-cm depths, as indicated by the temporal dynamics of the standard deviation of soil water content presented in Fig. 2 (bottom panel). We attribute this to the pedological situation (shallow soil above consolidated bedrock) in which the highly variable stone content in the subsoil leads to considerable spatial variability of soil water content at the 50-cm depth."
Furthermore, the revised discussion also states (line 622-624): "The fact that we replace heterogeneous soil properties and soil moisture content for a given area by spatially homogeneous values, also introduces temporal variability in the effective parameters that are estimated in this study."
To me this sounds like heterogeneity is a rather important challenge for upscaling. Please clarify.
In this context (if heterogeneity is important) I didn't apprehend why the representation of photosynthesis was considered the most noteworthy model structural error (line 451-454): "The model error was set to zero assuming that uncertainty was captured by uncertain parameters and model forcings. However, we agree that it can be expected that we have other model structural errors, for example in relation to the representation of photosynthesis."

2. I appreciate that the authors investigate the ensemble inflation method (line 549-552): "The effect of initial uncertainties on the performance of EnKF with the ensemble inflation method is also tested with the VIC-3L model. The forcing error was increased from 10% to 20%. Table 6 shows the RMSE values for soil moisture content characterization in the assimilation and verification periods. The difference between the results for 10% or 20% perturbation of the forcings is very limited, for both variants of the EnKF-method." However, I suspect that there is a misunderstanding. How can you assess the effect of the initial uncertainties of the parameters, by changing the forcing error and not the initial uncertainties?
As a side note: The inflation method keeps the parameter uncertainties constant. However, you now also state that "parameter uncertainty decreased" (line 636). How could you attest this?

**Minor comments:**
3. Table 5: In original manuscript Figure 8, there was basically no difference for the prediction of the water content of the first layer, whether there are parameters estimated or not. The authors clarified this: "Predictions of soil moisture content for layer 2 and layer 3 (in the verification period) improved significantly for the case of parameter estimation. Concerning the soil moisture content of layer 1, the RMSE value of the open loop run is 0.053m3/m3, which is already quite close to the observed values. In addition, the soil moisture content for the upper layer is strongly driven by single precipitation events. We extended the discussion of these results (line 533-536): "In the verification period, the RMSE values of the scenario noParamUpdate are close to the RMSE values of the open loop run. If soil parameters were updated during the assimilation period, the RMSE values for soil moisture characterization were reduced. More specifically, the four methods show a RMSE improvement of about 54% and 42% for the second and third model layer (compared with the open loop run)." "
The part of the answer ("Concerning the soil moisture content of layer 1, the RMSE value of the open loop

run is 0.053m3/m3, which is already quite close to the observed values. In addition, the soil moisture content for the upper layer is strongly driven by single precipitation events.") was added to the results for CLM instead of VIC-3L. Please correct. Furthermore, to me this statement doesn't seem entirely consistent with the discussion (line 675-682): "In the verification period soil moisture of the top layer cannot be represented perfectly by the two LSM's, in spite of parameter updating with state of the art data assimilation methods. Table 5 and table 9 illustrate that the RMSE values of the four joint state and parameter assimilation methods are similar for both models, which means that both models have larger errors for the top layer. There is a number of reasons for the larger soil moisture mismatches for the upper layer: (i) the memory effect from initial conditions, very well identified at the beginning of the verification period, is smaller for the upper soil layer, as this layer is more affected by precipitation events and evaporation; (ii) these soil moisture changes make that it is also more affected by model structural errors, for example concerning evaporation processes."

4. Figure 4: In original manuscript Figure 5, Parameter b estimated by MCMC showed a large difference to the other methods. But MCMC performed approximately as well as the other filters. The authors clarify this in their response: "Demaria et al. (2007) evaluated the sensitivity and identifiability of ten parameters which control surface and subsurface runoff in the VIC model for four U.S. watersheds along a hydroclimatic gradient. They found that parameter b is crucial in a dry environment, while its impact on model performance is not significant for wet sites. They concluded that parameter b plays a key role in partitioning rainfall into soil moisture and surface runoff in dry environments. [Liang and Guo, 2003] and [Atkinson et al., 2002] reached a similar conclusion. In our work, as the Rollesbroich catchment is very wet, even though parameter b estimated by MCMC shows a large difference with other methods, it shows small impact on the soil moisture content for layer 1 and layer 2. In the revised manuscript, all experiments of VIC-3L were done again. Evolution of parameter b estimated by MCMC was more reasonable (figure 4)."
Figure 4 now shows a similar value for b estimated with MCMC. What was changed to achieve the new results?

5. Figure 5: In original manuscript Figure 6 parameter spreads at time 0 seemed different. The authors clarified this: "In original manuscript figure 6 shows the evolution of the parameter ensemble from time step 1 but not time step 0. At time step 0, the ensemble spreads are the same, but at time step 1, the parameter ensemble is updated by PF or EnKF, and the ensemble spreads between EnKF and PF differ. We showed the evolution of the parameter ensemble from time step 0 onwards in the revised version of the manuscript (figure 5, figure 8 and figure 11)."
Now MCMC, AUG and DUAL seem to have the same initial spread, but to me PF still seems to have a different spread?

6. Equation A8: The authors missed to explain the inconsistent dimensions (you add $[LT^{-1}]$, $[]$ and $[L]$). Please clarify or correct the equation.

7. Figure 11: I agree that the shown parameters are more meaningful than the estimated soil texture. I still think it is worth to mention (e.g. in the caption), that the shown parameters are not the directly estimated ones.

8. Line 768: If you do mean the soil matric potential and not the matric head, the dimension is not $[L]$, but $[E\ L^{-3}]$.

9. Concerning the reply about RMSE: "Thanks, we admit that for a prediction in the verification phase we cannot expect a better result than a RMSE equal to the measurement uncertainty but still an RMSE equal to 0 would be the best result."

A RMSE equal to 0 is not the best result. It would mean, that the model perfectly describes the measurement noise, but not the actual state.

---

## Referee Report (RR2)

The authors have revised and reformulated their manuscript in large parts. This did increase the quality. However, I cannot understand why this was not done before submission. Theses late changes increase the work for the reviewers.

On top of this, the new manuscript partly revokes at least one change from the previous version.
I had asked before:
*Page 11 Lines 34-36: "The soil moisture observation error is assumed to be normally distributed with mean equal to 0 and standard deviation equal to $0.02m^3/m^3$, for both VIC-3L and CLM." Please discuss why you assume this uncertainty, especially since it is a mean of 41 values.*
To which the authors answered:
*Reply: We admit that $0.02m^3 3/m^3$ is a little larger than the uncertainty of the mean soil moisture content averaged over the 41 values. A larger observation error elevates potential problems with filter inbreeding. In addition, it adds flexibility in case of the presence of an observation bias or model structural error. This was added in the revised version of the manuscript (line 448-451):*
"We admit that $0.02m^3/m^3$ is a little larger than the uncertainty of the mean soil moisture content averaged over the 41 values. A larger observation error elevates potential problems with filter inbreeding. In addition, it adds flexibility in case of the presence of an observation bias or model structural error."
Now it reads (Page 16, Lines 573-576): "The measurement errors of the soil moisture observations are assumed to be zero-mean Gaussian with standard deviation, $\sigma = 0.02$ m$^3$/m$^3$. This results in $\mathbf{R} = 4 \cdot 10^{-4}\mathbf{I}_m$ in equations (4) and (17), respectively. The value of $\sigma$ is set larger than its default of say 0.01 m$^3$/m$^3$ to compensate, at least in part, for the lack of use of an explicit error model."
I might be a little pedantic about this and the chosen error of 0.02 m$^3$/m$^3$ seems appropriate, as explained by the authors. But, the error of the mean of 41 measurements with individual errors of 0.02 m$^3$/m$^3$ is 0.003 m$^3$/m$^3$. I think it is a different message if the measurement uncertainty was increased by a factor of 2 or by a factor of over 6. Please correct this.
Due to the rewriting of almost the complete manuscript, these changes go unnoticed (and at least in this instance have also altered the given statement). Hence, I ask the authors to answer all the previously asked questions, that resulted in changes of the text, again.

I have also comments regarding two previous questions, that were not answered sufficient to me:

Regarding my previous question 4: I had asked, what was changed to achieve the different results for the VIC-3L parameter b with MCMC (in the first manuscript the value had deviated strongly from the results by the other assimilation methods).
The authors answered: "We redid all the experiments including the generation of the initial ensemble of parameters."
What exactly did you change in the generation of the initial ensemble of parameters? Was it e.g. just the seed for the generation of random numbers? If this is the case, I think it is an important information, that the seed had to be tuned to achieve a consistent result with the other methods and should be mentioned in the manuscript. Please clarify this.

Regarding my previous question 2: I had asked about the impact of different initial parameter uncertainties (in the new manuscript called *prior parameter distribution*) for the estimation with the EnKF in the context of the chosen inflation method. The authors answer "[...] that this uncertainty was not underestimated [...]" and refer to the paper by Whitaker and Hamill (2012).
However, the authors have chosen a special case of the inflation method proposed by Whitaker and Hamill (2012): They keep the parameter spread constant, while Whitaker and Hamill (2012) actually scale the relaxation to the prior spread with a tuning parameter. In their case the best results were not obtained with

the special case of keeping the spread constant. In my understanding, this choice additionally leads to a direct permanent coupling of the parameter uncertainty at each time step to the initially chosen uncertainty. Due to this introduced coupling, I think, that the choice of the initial parameter uncertainty (*prior parameter distribution*) is important. Thus, it might be not enough to ensure that the initial parameter uncertainty is not underestimated. It is possible that the uncertainty was chosen too large and that a smaller uncertainty could actually improve the results.

Please discuss this or explain where I'm wrong.

---

## Author Response (AR2)

**Editor Decision: Publish subject to revisions (further review by Editor and Referees) (27 Nov 2016) by Prof. Dr. Kurt Roth**

*Comments to the Author:*

*The manuscript improved significantly. This is also attested by the reviewers. Still, some important questions pointed out by the reviewers remain open and need to be addressed. Please go through them carefully. Since the raised issues are beyond the mere technical, I'll keep the reviewers involved also in the next, hopefully final iteration.*

**Reply:** **Thanks. We revised the manuscript taking into account the comments of the reviewers. In addition, we reformulated in our new manuscript almost all the sentences. We believe that it strongly improves the readability of the manuscript. Our results and conclusions are however not modified.**

**As the text was edited thoroughly we didn't submit a marked-up manuscript version because you would see that entire paper is red.**

**Reviewer I**

*I appreciate the detailed answers and the additionally performed simulations to investigate the raised issues. However, I still have few comments regarding answers, which were inconsistent or not sufficient to me. I would like to see them clarified:*

*Major comments:*

*1. The expected impact of heterogeneity is now explained in line 376-380: "Qu et al. (2014) described the statistics of soil properties for soil samples taken in the Rollesbroich catchment. Soil texture showed a relatively limited variation. In our work only vertical heterogeneity is considered. In this case, heterogeneity does not seem to be very strong and we do not face a challenging upscaling case for the land surface model." However, Qu et al. (2014) seem to state that there is considerable heterogeneity, e.g.: "Spatial variability of the measured soil water content was higher at the 50-cm depth than the 5- and 20-cm depths, as indicated by the temporal dynamics of the standard deviation of soil water content presented in Fig. 2 (bottom panel). We attribute this to the pedological situation (shallow soil above consolidated bedrock) in which the highly variable stone content in the subsoil leads to considerable spatial variability of soil water content at the 50-cm depth."*

*Furthermore, the revised discussion also states (line 622-624): "The fact that we replace heterogeneous soil properties and soil moisture content for a given area by spatially homogeneous values, also introduces temporal variability in the effective parameters that are estimated in this study." To me this sounds like heterogeneity is a rather important challenge for upscaling. Please clarify. In this context (if heterogeneity is important) I didn't apprehend why the representation of photosynthesis was considered the most noteworthy model structural error (line 451-454): "The model error was set to zero assuming that uncertainty was captured by uncertain parameters and model forcings. However, we agree that it can be expected that we have other model structural errors, for example in relation to the representation of photosynthesis."*

**Reply: We thank the reviewer for pointing out the improvement of our manuscript. It is true that in spite of a limited spatial variability in soil texture characteristics the spatial variability in soil moisture content is not so small. This could be related to the influence of groundwater and the presence of a drainage system, but also to variations in soil hydraulic properties although texture is quite homogeneous. We decided therefore to reformulate the sentences cited to (line 541-551):**

*"In this work, we conveniently assume the soil-land-surface domain of the Rollesbroich site to be homogeneous and characterized by areal average values of soil moisture content at 5, 20 and 50 cm depth. In other words, we consider only vertical variations in soil water storage. Common LSM data assimilation experiments published in the literature usually involve application to much larger spatial scales, especially when remote sensing data are used. Hence, it is important to evaluate the LSM performance for a site where heterogeneities are neglected. Qu et al. (2014) investigated the geostatistical properties of the soils of the Rollesbroich test site. This work demonstrated a rather small spatial variability of the soil texture. This does not suggest, however that we can ignore spatial variations in the measured soil moisture values. Indeed, the standard deviations of soil moisture vary between 0.04 and 0.07 $cm^3/cm^3$ depending on the actual soil layer. This spatial heterogeneity of the soil moisture data documents variability in the soil hydraulic properties, and complicates the application and upscaling of LSMs."*

**We take the simple or biased representation of photosynthesis as an example of model structure error, but it doesn't mean that the representation of photosynthesis is the only most noteworthy model structural error. Models are assemblies of assumptions and simplifications and thus inevitably imperfect approximations to the complex reality. We would say all these simple parameterizations and mathematical implementations (e.g., spatial and temporal discretization) can lead to model structure error. We reformulated the sentences in the new manuscript (line 578-585):**

*"Also, we account crudely for errors in LSM model formulation via parameter uncertainty (discussed next) and the use of a stochastic description of the precipitation record of the Rollesbroich site. The hyetograph of each ensemble member is derived by multiplying the measured hourly precipitation rates of the tipping bucket with multipliers drawn from a unit-mean normal distribution with standard deviation of 0.10. This is equivalent to a heteroscedastic error of 10% of the observed precipitation (Hodgkinson et al., 2004). Forcing variables which govern evapotranspiration (incoming shortwave and longwave radiation, air temperature, relative humidity, and wind speed) were not corrupted. Of course, the prior parameter distribution and precipitation forcing do not account for errors in the photosynthesis module."*

*2. I appreciate that the authors investigate the ensemble inflation method (line 549-552): "The effect of initial uncertainties on the performance of EnKF with the ensemble inflation method is also tested with the VIC-3L model. The forcing error was increased from 10% to 20%. Table 6 shows the RMSE values for soil moisture content characterization in the assimilation and verification periods. The difference between the results for 10% or 20% perturbation of the forcings is very limited, for both variants of the EnKF-method." However, I suspect that there is a misunderstanding. How can you assess the effect of the initial uncertainties of the parameters, by changing the forcing error and not the initial uncertainties? As a side note: The inflation method keeps the parameter uncertainties constant. However, you now also state that "parameter uncertainty decreased" (line 636). How could you attest this?*

**Reply: The reviewer asked about the impact of initial uncertainty (without further specification) on the performance of the ensemble inflation method, and we interpreted this as the role of the uncertainty of the model forcings, as this uncertainty was less well defined compared to the uncertainty of the soil texture. In the experiments also the uncertainty of texture was taken into account and we thought that this uncertainty was not underestimated and therefore did not consider increasing its perturbation. In addition, we thought the question was more about the inflation method itself. When Whitaker and Hamill (2012) proposed the inflation method we used in our study, experiments were conducted to account for background errors not accounted for by the first-guess ensemble, which in their study included both sampling error and model error. Their tests with large and small ensembles, with and without model error, suggested that this inflation method is well suited to account for unrepresented assimilation errors such as sampling error. In our study, we further found that the effect of forcing error on this method was limited. Therefore we think that this method is a well-built methodology and broadly used in data assimilation research. We admit that the "parameter uncertainty decreased" (line 638) is confusing. The inflation method kept the parameter spread constant. We deleted this statement in the revised version to avoid the misunderstanding.**

*Minor comments:*

*3. Table 5: In original manuscript Figure 8, there was basically no difference for the prediction of the water content of the first layer, whether there are parameters estimated or not. The authors clarified this: "Predictions of soil moisture content for layer 2 and layer 3 (in the verification period) improved significantly for the case of parameter estimation. Concerning the soil moisture content of layer 1, the RMSE value of the open loop run is 0.053m3/m3, which is already quite close to the observed values. In addition, the soil moisture content for the upper layer is strongly driven by single precipitation events. We extended the discussion of these results (line 533-536): "In the verification period, the RMSE values of the scenario noParamUpdate are close to the RMSE values of the open loop run. If soil parameters were updated during the assimilation period, the RMSE values for soil moisture characterization were reduced. More specifically, the four methods show a RMSE improvement of about 54% and 42% for the second and third model layer (compared with the open loop run)."*

*The part of the answer ("Concerning the soil moisture content of layer 1, the RMSE value of the open loop run is $0.053m^3/m^3$, which is already quite close to the observed values. In addition, the soil moisture content for the upper layer is strongly driven by single precipitation events.") was added to the results for CLM instead of VIC-3L. Please correct. Furthermore, to me this statement doesn't seem entirely consistent with the discussion (line 675-682): "In the verification period soil moisture of the*

*top layer cannot be represented perfectly by the two LSM's, in spite of parameter updating with state of the art data assimilation methods. Table 5 and table 9 illustrate that the RMSE values of the four joint state and parameter assimilation methods are similar for both models, which means that both models have larger errors for the top layer. There is a number of reasons for the larger soil moisture mismatches for the upper layer: (i) the memory effect from initial conditions, very well identified at the beginning of the verification period, is smaller for the upper soil layer, as this layer is more affected by precipitation events and evaporation; (ii) these soil moisture changes make that it is also more affected by model structural errors, for example concerning evaporation processes."*

**Reply: We revised almost all the text in the revised version. We discussed this issue in the discussion (line 872-876):**

***"Despite this improvement in model performance over an open-loop simulation, VIC-3L and CLM do not adequately characterize soil moisture dynamics of the top layer (5 cm measurement depth) during the evaluation period (RMSE values of about 0.05 $cm^3/cm^3$). We posit that these two models do not characterize adequately processes such as water infiltration, soil evaporation, and/or root water uptake (transpiration), which govern rapid variations in soil moisture storage in the top soil."***

*4. Figure 4: In original manuscript Figure 5, Parameter b estimated by MCMC showed a large difference to the other methods. But MCMC performed approximately as well as the other filters. The authors clarify this in their response: "Demaria et al. (2007) evaluated the sensitivity and identifiability of ten parameters which control surface and subsurface runoff in the VIC model for four U.S. watersheds along a hydroclimatic gradient. They found that parameter b is crucial in a dry environment, while its impact on model performance is not significant for wet sites. They concluded that parameter b plays a key role in partitioning rainfall into soil moisture and surface runoff in dry environments. [Liang and Guo, 2003] and [Atkinson et al., 2002] reached a similar conclusion. In our work, as the Rollesbroich catchment is very wet, even though parameter b estimated by MCMC shows a large difference with other methods, it shows small impact on the soil moisture content for layer 1 and layer 2. In the revised manuscript, all experiments of VIC-3L were done again. Evolution of parameter b estimated by MCMC was more reasonable (figure 4)." Figure 4 now shows a similar value for b estimated with MCMC. What was changed to achieve the new results?*
**Reply: We redid all the experiments including the generation of the initial ensemble of parameters.**

*5. Figure 5: In original manuscript Figure 6 parameter spreads at time 0 seemed different. The authors clarified this: "In original manuscript figure 6 shows the evolution of the parameter ensemble from time step1 but not time step 0. At time step 0, the ensemble spreads are the same, but at time step 1, the parameter ensemble is updated by PF or EnKF, and the ensemble spreads between EnKF and PF differ. We showed the evolution of the parameter ensemble from time step 0 onwards in the revised version of the manuscript (figure 5, figure 8 and figure 11)." Now MCMC, AUG and DUAL seem to have the same initial spread, but to me PF still seems to have a different spread?*
**Reply: For figure 5, we checked the source datasets for the figure. PF has the same initial parameter ensemble at time 0, but when I plotted the figure, PF started still from time step 1. This has been modified in the new version of the manuscript.**

*6. Equation A8: The authors missed to explain the inconsistent dimensions (you add [LT-1], [] and [L]).Please clarify or correct the equation.*
**Reply: Thanks. We corrected the unit in the revised manuscript.**

*7. Figure 11: I agree that the shown parameters are more meaningful than the estimated soil texture. I still think it is worth to mention (e.g. in the caption), that the shown parameters are not the directly estimated ones.*
**Reply: The text was added in the caption of figure 11 in the revised manuscript:**

***"Please note that $log_{10}k_s$ (in $log_{10}(m/s)$) and parameter B are derived from the sand, clay, and organic matter fractions of each soil layer, which are estimated during the assimilation period."***

*8. Line 768: If you do mean the soil matric potential and not the matric head, the dimension is not [L], but [EL-3].*

**Reply: We corrected this to "soil matric head". Confusing is that the technical description of CLM 3.5 [*Oleson et al.*, 2004] refers to it as soil water matric potential.**

*9. Concerning the reply about RMSE: "Thanks, we admit that for a prediction in the verification phase we cannot expect a better result than a RMSE equal to the measurement uncertainty but still an RMSE equal to 0 would be the best result." A RMSE equal to 0 is not the best result. It would mean, that the model perfectly describes the measurement noise, but not the actual state.*

**Reply: Thanks, you are right. But in reality we don't know the actual state values, so we still think that a smaller RMSE value indicates a better prediction (line 641-642):**

**"*Larger values of the NSE and smaller values of the RMSE are preferred as they indicate a better LSM performance.*"**

**Reference**

Crow, W. T., D. Ryu, and J. S. Famiglietti (2005), Upscaling of field-scale soil moisture measurements using distributed land surface modeling, *Advances in Water Resources*, *28*(1), 1-14, doi:10.1016/j.advwatres.2004.10.004.

Oleson, K. W., et al. (2004), Technical Description of the Community Land Model (CLM), *NCAR Technical Note NCAR/TN-461+STR*, 186.

Qu, W., H. R. Bogena, J. A. Huisman, G. Martinez, Y. A. Pachepsky, and H. Vereecken (2014), Effects of soil hydraulic properties on the spatial variability of soil water content: evidence from sensor network data and inverse modeling, *Vadose Zone Journal*, 13(12), doi:10.2136/vzj2014.07.0099.

Vereecken, H., R. Kasteel, J. Vanderborght, and T. Harter (2007), Upscaling hydraulic properties and soil water flow processes in heterogeneous soils: A review, *Vadose Zone Journal*, *6*(1), 1-28, doi:10.2136/vzj2006.0055.

Whitaker, J. S., and T. M. Hamill (2012), Evaluating Methods to Account for System Errors in Ensemble Data Assimilation, *Monthly Weather Review*, *140*(9), 3078-3089, doi:10.1175/mwr-d-11-00276.1.

Xiao, J. F., K. J. Davis, N. M. Urban, K. Keller, and N. Z. Saliendra (2011), Upscaling carbon fluxes from towers to the regional scale: Influence of parameter variability and land cover representation on regional flux estimates, *Journal of Geophysical Research-Biogeosciences*, *116*, doi:10.1029/2010jg001568.

**Reviewer II**

*As I already reviewed the first version of the discussion paper as reviewer #2, I will focus here on the reply of the authors and the changes made in this version of the manuscript.*

*I appreciate the efforts undertaken by the authors to revise this manuscript, which improved substantially at least from my point of view. All my comments were addressed properly, also the readability of the figures is much better now.*

Reply: **Thanks for pointing out the improvement of our manuscript.**

*I have only some words to say about the comment and discussion about temporal non-stationary parameters: The discussion introduced in ll.620-639 in the revised manuscript generally tackles this issue well, but does not really answer the question about the predictive power of such parameters. Similar with the editor, it is not quite clear for me, how parameters, which are meant to change in time, can be used for predictions. In case, that there is a theoretical reason for this, the temporal evolution should then enter the model equations to improve the predictions. It is reasonable, that the parameter may evolve in time as a consequence of using neglecting spatial heterogeneity, as described in ll. 622-625, but then we get the right answer for the wrong reasons, and predictive power is not enhanced. Although this issue is important at least from my perspective, I see that the authors follow a different focus in the paper, which is elaborated thoroughly and presented in a straightforward manner. I would appreciate when the authors could tackle this point as well, but I can accept their decision to work on that aspect not in great detail.*

Reply: **Thanks. Since our focus of this manuscript is to compare the four methods of joint assimilation of states and parameters in two land surface models and the manuscript is already long, we think that we will not work on this aspect in great detail. This is a difficult issue and we believe that estimated soil hydraulic parameters will also do a quite good job in other time periods, probably because the effective parameter values for the larger grid cell would not change so much (this is a guess). However, in case exhaustive high quality measurements would be available over a large area this could be further tested and it could for example be detected if over a year effective (estimated) parameters would show a typical yearly cycle which would be relatively similar for different years. Such a yearly cycle of effective soil hydraulic parameters could also be used then for other years. This remains however speculative and now we tend to think that estimated effective parameters for a larger grid cell are certainly not perfect under all conditions, but would in general lead to better predictions, also under different meteorological conditions. We included this small discussion on the quality of the effective parameters also in the discussion (line 917-925):**

*"It is difficult to assess whether the inferred VIC-3L and CLM parameter values will do a good job at predicting soil moisture dynamics at the different measurement depths during a much longer evaluation period with wet and dry conditions. As the estimated parameters represent effective properties of the Rollesbroich site, one may expect their calibrated values not to change too much over time. We would need additional soil moisture data and/or other type of measurements to corroborate this. Nevertheless, the effective parameter values derived herein improve characterization of soil moisture dynamics at the Rollesbroich site compared to a separate state estimation run with VIC-3L and CLM using parameters drawn randomly from the prior distribution, or open loop simulation using the ensemble mean model output of a large cohort of parameter vectors drawn randomly from the prior parameter distribution (initial parameter ensemble)."*

*ll.636-638: "For highly identifiable parameters, parameter uncertainty decreased and parameters converged fast. So we think that joint estimation of states and time-variant parameters with data assimilation still shows great potentials in terms of identifiability of parameters." I do not understand the point. When parameters were highly identifiable, then the method works well. That is clear to me, but what is the "great potentials in terms of identifiability"? Do you mean to distinguish between such parameters which can be constrained well and others which do not? It would be more interested to hear about the behaviour, when parameters are not well identifiable as it is typically the case for complex LSMs. Please clarify this.*

**Reply:** **We meant to emphasize the capability of joint estimation of states and time-variant parameters with data assimilation in parameter estimation. We revised almost all the text in the revised version and we discussed this issue in the new manuscript (line 898-916):**

*"In our implementation of the EnKF and KF, the VIC-3L and CLM parameters were assumed to be time-variant and their values updated jointly with the model states at each assimilation time step. The 5-month calibration period we used herein involves several large precipitation events, and as a consequence, the soil profile is rather wet. The resulting parameter estimates might therefore not be representative for dry periods with much lower moisture values of the soil profile. What is more, the assumption of spatial homogeneity might not characterize adequately the distributed soil properties of the Rollesbroich site and induce temporal variability in the VIC-3L and CLM parameters. Bias in model input and measurement errors of the forcing data also contribute to the temporal fluctuations of the estimated parameter values. These temporal parameter variations are meaningful in some cases as they can help diagnose structural model inadequacies and/or biases in model input and forcing data. Kurtz et al. (2012) successfully estimated a temporally-variant parameter with the EnKF, but these authors concluded that the algorithm needs a considerable spin-up period to "warm-up" to new parameter values. Vrugt et al. (2013) found considerable temporal non-stationarity in the parameters estimated by PMCMC as a result of the small time period used to calculate the acceptance probability of candidate particles. This finding is in agreement with the results of PMCMC in our paper. Of course, we could have assumed time-invariant parameters via a method such as SODA, yet this would have enhanced significantly computational requirements. Fortunately, parameters estimated via our implementation of the EnKF exhibit asymptotic properties during the assimilation period (e.g. see Shi et al. (2015)). This is particularly true for highly sensitive parameters. An example of this was parameter Dm of VIC-3L which quickly converged to values of around 1 − 2 mm after assimilating just a handful of soil moisture observations."*

---

## Author Response (AR3)

**Editor Decision: Reconsider after major revisions (further review by Editor and Referees) (10 May 2017) by Kurt Roth**

*Comments to the Author*:

*Foreseeably, both reviewers were disturbed/confused by the late and extensive rewriting of the manuscript and not amused by the additional workload.*

**Reply:** **We agree that it was unfortunate to make so many editorial changes in such a late stage of the review process. We apologize for this. One of our co-authors missed the submission and the first round of the review process (major revision). During the second round (minor revision) he made a great effort editing the manuscript to make it a very nice / well written and structured paper. But only texts were changed and our results and conclusions were unchanged. We apologize again for the additional workload.**

*One reviewer found the manuscript good enough to publish with some comments at the discretion of the authors. These concerned: (i) dropping "joint" from the title and (ii) the use of the term "effective".*

**Reply:** **We dropped "Joint" from the title. We replaced "effective" with "apparent" where applicable.**

*The other reviewer submitted a report and proposed major revision in order to be able to check the responses. In particular, the request is to mark all essential modifications that may go unnoticed in the typical late-stage reading of a manuscript. An example of such a modification, which I completely agree is noteworthy and which has escaped my notice, is given in the reviewer's report. I ask you to*

*1. go carefully through the manuscript in its transition to the latest version and mark all the key modifications*

*2. address carefully the reviewer's questions, also the unanswered ones from the previous iteration (you need not agree with the reviewer, but you must comment on the raise issues).*

**Reply:** **We put a marked-up manuscript version which marks all the key modifications for the responses. We also addressed carefully all the reviewer questions including his/her former questions in the last two revisions.**

*Given the real long time this manuscript has been in the making and the iterative process, I want this to be the last round with one of two possible outcomes: publish or submit as a new manuscript.*

**Reply:** **We are confident that this is indeed the last revision round. We would like also to re-stress that in the last revision round, although many editorial changes were made, the content did not change.**

**Reviewer I**

The authors have revised and reformulated their manuscript in large parts. This did increase the quality. However, I cannot understand why this was not done before submission. Theses late changes increase the work for the reviewers.

**Reply: We apologize for the additional workload. One of our co-authors missed the submission and the first round of the review process (major revision). During the second round (minor revision) he made a great effort editing the manuscript to make it a very nice / well written and structured paper. We appreciate his work, because we believe the manuscript improved. Only texts were changed and our results and conclusions were unchanged. But we are sorry for the inconvenience caused by the extensive rewriting.**

On top of this, the new manuscript partly revokes at least one change from the previous version. I had asked before:

*Page 11 Lines 34-36: "The soil moisture observation error is assumed to be normally distributed with mean equal to 0 and standard deviation equal to 0.02m3/m3, for both VIC-3L and CLM." Please discuss why you assume this uncertainty, especially since it is a mean of 41 values.*

To which the authors answered:

Reply: We admit that $0.02m^3/m^3$ is a little larger than the uncertainty of the mean soil moisture content averaged over the 41 values. A larger observation error elevates potential problems with filter inbreeding. In addition, it adds flexibility in case of the presence of an observation bias or model structural error. This was added in the revised version of the manuscript (line 448-451):

*"We admit that $0.02m^3/m^3$ is a little larger than the uncertainty of the mean soil moisture content averaged over the 41 values. A larger observation error elevates potential problems with filter inbreeding. In addition, it adds flexibility in case of the presence of an observation bias or model structural error."*

Now it reads (Page 16, Lines 573-576): "The measurement errors of the soil moisture observations are assumed to be zero-mean Gaussian with standard deviation $\sigma = 0.02m^3/m^3$. This results in $\mathbf{R} = 4 \cdot 10^{-4}\mathbf{I}_m$ in equations (4) and (17), respectively. The value of $\sigma$ is set larger than its default of say 0.01 $m^3/m^3$ to compensate, at least in part, for the lack of use of an explicit error model."

I might be a little pedantic about this and the chosen error of $0.02m^3/m^3$ seems appropriate, as explained by the authors. But, the error of the mean of 41 measurements with individual errors of $0.02m^3/m^3$ is $0.003m^3/m^3$. I think it is a different message if the measurement uncertainty was increased by a factor of 2 or by a factor of over 6. Please correct this.

**Reply: We agree with what you said. We revised the text in our manuscript (line 574-577):**

**"*The measurement errors of the soil moisture observations are assumed to be zero-mean Gaussian with standard deviation, $\sigma = 0.02 \ m^3/m^3$. This results in $R = 4 \cdot 10^{-4}I_m$ in equations (4) and (17), respectively. We admit that $0.02m^3/m^3$ is clearly larger than the uncertainty of the mean soil moisture content averaged over the 41 values. A larger observation error alleviates potential problems with filter inbreeding.*"**

Due to the rewriting of almost the complete manuscript, these changes go unnoticed (and at least in this instance have also altered the given statement). Hence, I ask the authors to answer all the previously asked questions that resulted in changes of the text, again.

**Reply:** **We answered all your previously asked questions that resulted in changes of the text again in this response (see below).**

I have also comments regarding two previous questions that were not answered sufficient to me:

Regarding my previous question 4: I had asked, what was changed to achieve the different results for the VIC-3L parameter b with MCMC (in the first manuscript the value had deviated strongly from the results by the other assimilation methods).

The authors answered: "We redid all the experiments including the generation of the initial ensemble of parameters."

What exactly did you change in the generation of the initial ensemble of parameters? Was it e.g. just the seed for the generation of random numbers? If this is the case, I think it is important information that the seed had to be tuned to achieve a consistent result with the other methods and should be mentioned in the manuscript. Please clarify this.

**Reply:** **Yes, we changed the random seed to get the new results for all the methods, not only for the MCMC method. In our work all the methods used the same random seed to ensure they have the same initial ensemble of parameters. We clarify this point in the new manuscript (line 593-594):**

*"This initial parameter ensemble is the same for all the assimilation methods."*

Regarding my previous question 2: I had asked about the impact of different initial parameter uncertainties (in the new manuscript called prior parameter distribution) for the estimation with the EnKF in the context of the chosen inflation method. The authors answer "[...] that this uncertainty was not underestimated [...]" and refer to the paper by Whitaker and Hamill (2012).

However, the authors have chosen a special case of the inflation method proposed by Whitaker and Hamill (2012): They keep the parameter spread constant, while Whitaker and Hamill (2012) actually scale the relaxation to the prior spread with a tuning parameter. In their case the best results were not obtained with the special case of keeping the spread constant. In my understanding, this choice additionally leads to a direct permanent coupling of the parameter uncertainty at each time step to the initially chosen uncertainty.

Due to this introduced coupling, I think, that the choice of the initial parameter uncertainty (prior parameter distribution) is important. Thus, it might be not enough to ensure that the initial parameter uncertainty is not underestimated. It is possible that the uncertainty was chosen too large and that a smaller uncertainty could actually improve the results.

Please discuss this or explain where I'm wrong.

**Reply:** **We agree with "I think, that the choice of the initial parameter uncertainty (prior parameter distribution) is important. Thus, it might be not enough to ensure that the initial parameter uncertainty is not underestimated. It is possible that the uncertainty was chosen too large and that a smaller uncertainty could actually improve the results." However, Whitaker and Hamill (2012) used a climate model to initialize ensemble weather predictions and they only**

updated states with EnKF. In our work, we did data assimilation in combination with land surface models, and we updated both states and parameters. If parameter estimation is included in the data assimilation, results are more prone to filter inbreeding especially for a small ensemble size. However, the reviewer is right that if ensemble spread is maintained during data assimilation, it is more important that initial ensemble spread is correct. However, we believe that we could control the ensemble spread well as the applied uncertainty on soil hydraulic parameters and meteorological forcings is reasonable.

To discuss this issue, we added the text in the revised manuscript (line 381-385):

"*This method promotes a parameter spread that is in agreement with the width of the prior parameter distribution, and is particularly important to avoid a strong underestimation of ensemble variance and associated filter inbreeding in applications with relatively small ensemble sizes. As the spread is kept artificially constant, it cannot be assessed properly how data assimilation affects reduction of prediction uncertainty. In addition, it is important that the initial ensemble spread is adequate. This is a drawback of the applied inflation.*"

**Previous comments from reviewer I in the first review round**

*Page 10, Lines 20-23: "... and a soil moisture and soil temperature sensor network (with measurements at 5, 20 and 50cm depth) are installed, amongst others. Soil moisture time series at 41 locations are being recorded." What kind of soil moisture sensors are installed? Please discuss possible uncertainties in the data.*

**Reply:** The SPADE soil water content probes (sceme.de GmbH i.G., Horn-Bad Meinberg, Germany; (*Hübner et al.*, 2009)) were installed at 5 cm, 20 cm and 50 cm depth along a vertical profile. The SPADE probe is a ring oscillator and the frequency of the oscillator is a function of the dielectric permittivity of the surrounding medium, which is strongly dependent on the soil water content because of the high relative permittivity of water ($\approx$80) as compared to mineral soil solids ($\approx$2-9), and air ($\approx$1). The SPADE probe was calibrated according to the procedure outlined in (*Qu et al.*, 2014). The possible uncertainties in the soil moisture data are related to imperfect contact of the sensors with the soil, imperfection of the model which relates the sensor response and dielectric permittivity and imperfection of the model which relates dielectric permittivity and soil moisture. The measurement error is assumed to be 0.02cm$^3$/cm$^3$. We added the text in the revised version of the manuscript to clarify this (line 524-532):

*"Water content data are measured at 41 different locations (see Figure 1) using SPADE soil moisture probes (sceme.de GmbH i.G., Horn-Bad Meinberg, Germany) (Hübner et al., 2009) installed at 5 cm, 20 cm and 50 cm depth along a vertical profile. The SPADE probe is a ring oscillator and the frequency of the oscillator is a function of the dielectric permittivity of the surrounding medium, which depends strongly on local soil water content because of the high relative permittivity of water ($\approx$ 80) as compared to mineral soil solids ($\approx$ 2-9), and air ($\approx$ 1). The SPADE probe was calibrated following the procedure outlined in (Qu et al., 2014). The soil moisture measurements are subject to several sources of error. This includes an inadequate contact of the sensors with the surrounding soil, and structural imperfections of the equations which relate the sensor response to the dielectric permittivity, and this permittivity to soil moisture."*

*Page 10, Lines 24-35: "In this work, the Rollesbroich site is modeled as a single point and the data of the soil sensor network are averaged to calculate areal averages of soil moisture content at 5cm, 20cm and 50cm depth." Please discuss the importance and implications of this assumption. What are the expected impacts of heterogeneity?*

**Reply:** Data assimilation experiments with land surface models are generally conducted for large scales, especially when remote sensing data are assimilated. Therefore it is important to evaluate the land surface model performance for a site where heterogeneities are neglected. The Rollesbroich site has an area of 0.27km$^2$, which is a very small catchment. *Qu et al.* (2014) described the statistics of soil properties for soil samples taken in the Rollesbroich catchment (see Table 1). Their work demonstrated a rather small spatial variability of the soil texture. This does not suggest, however that we can ignore spatial variations in the measured soil moisture values. Indeed, the standard deviations of soil moisture vary between 0.04 and 0.07 cm$^3$/cm$^3$ depending on the actual soil layer. This spatial heterogeneity of the soil moisture data documents variability in the soil hydraulic properties, and complicates the application and upscaling of land surface models.

This was added in the revised manuscript (line 542-552):

*"In this work, we conveniently assume the soil-land-surface domain of the Rollesbroich site to be homogeneous and characterized by areal average values of soil moisture content at 5, 20 and 50 cm depth. In other words, we consider only vertical variations in soil water storage. Common LSM data assimilation experiments published in the literature usually involve application to much larger spatial scales, especially when remote sensing data are used. Hence, it is important to evaluate the LSM performance for a site where heterogeneities are neglected. Qu et al. (2014) investigated the geostatistical properties of the soils of the Rollesbroich test site. This work demonstrated a rather*

*small spatial variability of the soil texture. This does not suggest, however that we can ignore spatial variations in the measured soil moisture values. Indeed, the standard deviations of soil moisture vary between 0.04 and 0.07 cm³/cm³ depending on the actual soil layer. This spatial heterogeneity of the soil moisture data documents variability in the soil hydraulic properties, and complicates the application and upscaling of LSMs.*"

**Table 1. Descriptive statistics of soil properties on the basis of 273 soil samples in the Rollesbroich catchment. Table is from *Qu et al*., (2014).**

| | | Clay % | Sand % | Silt % | Bulk Density (g/cm³) | Carbon Content (g/kg) | Porosity (cm³/cm³) |
|---|---|---|---|---|---|---|---|
| **5cm** | **mean** | **18.99** | **19.90** | **61.10** | **0.94** | **54.47** | **0.65** |
| | **std** | **2.00** | **3.82** | **3.79** | **0.12** | **15.82** | **0.05** |
| **20cm** | **mean** | **18.03** | **20.76** | **61.20** | **1.28** | **34.08** | **0.52** |
| | **std** | **1.99** | **4.03** | **3.46** | **0.15** | **16.84** | **0.05** |
| **50cm** | **mean** | **16.50** | **22.00** | **61.50** | **1.52** | **11.22** | **0.43** |
| | **std** | **2.40** | **5.68** | **4.53** | **0.16** | **6.01** | **0.06** |

*Page 29, Figure 3: The figure shows higher water contents closer to the surface. Please mention this and discuss reasons and implications.*

**Reply: Text was added in the revised manuscript (line 653-663):**

"*Figure 3 displays the observed (blue dots) and VIC-3L predicted soil moisture values (solid lines) at (A) 5, (B) 20, and (C) 50 cm depths using PMCMC (black), RRPF (red), EnKF-AUG (green), and EnKF-DUAL (cyan). As the Rollesbroich test site experiences a yearly average precipitation of more than about 1000 mm it is not surprise that the upper soil layer at 5 cm is rather wet with volumetric soil moisture contents that vary dynamically between 0.3 and 0.5 cm³/cm³ in response to atmospheric forcing. This is especially true during the summer months (week 12 – 22) and explained by a rapid succession of rainfall and drying events. The larger porosity values of the surface layer explain the relatively high soil moisture contents of the 5 cm measurement depth. The storage time series of the deeper soil layers at 20 and 50 cm depth exhibit a rather negligible temporal variation with soil moisture values that range between 0.3-0.4 cm³/cm³ and show a damped and lagged response to rainfall. Note that the soil water storage of the deepest layer increases steadily during the year. This implies a drainage flux from the top soil to the aquifer (and drainage channels).*"

*Page 11 Lines 34-36: "The soil moisture observation error is assumed to be normally distributed with mean equal to 0 and standard deviation equal to 0.02m³/m³, for both VIC-3L and CLM." Please discuss why you assume this uncertainty, especially since it is a mean of 41 values.*

**Reply: We admit that 0.02m³/m³ is larger than the uncertainty of the mean soil moisture content averaged over the 41 values. A larger observation error elevates potential problems with filter inbreeding. In addition, it adds flexibility in case of the presence of an observation bias or model structural error. See also our response to your most recent response in this respect. This was added in the revised manuscript (line 574-577):**

"*The measurement errors of the soil moisture observations are assumed to be zero-mean Gaussian with standard deviation, $\sigma = 0.02 \; m^3/m^3$. This results in $R = 4 \cdot 10^{-4} I_m$ in equations (4) and (17), respectively. We admit that 0.02m³/m³ is clearly larger than the uncertainty of the mean soil moisture content averaged over the 41 values. A larger observation error alleviates potential problems with filter inbreeding.*"

*Page 11 Lines 33-34: "Precipitation was perturbed were perturbed by multiplicative error N(1,0.1) to represent the uncertainty of measured precipitation at the site." Please give a reason for this error. What is the assumed error for evaporation?*

**Reply: In the Rollesbroich catchment, precipitation was measured by a tipping bucket. Therefore only a measurement error was assumed, which is typically around 10% of the measured value (*Hodgkinson et al.*, 2004). In this work the variables which govern evapotranspiration (incoming shortwave and longwave radiation, air temperature, relative humidity, wind speed), were not perturbed. This was added in the revised version of the manuscript (line 582-586):**

**"*The hyetograph of each ensemble member is derived by multiplying the measured hourly precipitation rates of the tipping bucket with multipliers drawn from a unit-mean normal distribution with standard deviation of 0.10. This is equivalent to a heteroscedastic error of 10% of the observed precipitation (Hodgkinson et al., 2004). Forcing variables which govern evapotranspiration (incoming shortwave and longwave radiation, air temperature, relative humidity, and wind speed) were not corrupted.*"**

*Page 2, Lines 13-15: "This approach allows for joint estimation of the states and parameters while taking into explicit consideration model structural error and forcing data errors (Liu and Gupta, 2007)." This is correct, but it is not to the point, since the authors later set the model error to zero (see Page 11, Lines 36-37) and hence do not consider model structural errors. Please discuss this.*

**Reply: Yes, model structural error is not considered in our work, but parameter uncertainties and forcing uncertainties are considered and we assume that these capture in this case the model uncertainty. However, we agree that it can be expected that we have other model structural errors, for example in relation to the representation of photosynthesis. We added the text in our manuscript (line 577-581):**

**"*Also, we account crudely for errors in LSM model formulation via parameter uncertainty and the use of a stochastic description of the precipitation record of the Rollesbroich site (discussed next). In other words, the $k \times 1$ process noise vector, $\mathbf{w}_t$, in equation (2) consists of zeros. However, we agree that it can be expected that we have other model structural errors, for example in relation to the representation of photosynthesis.*"**

*Page 6, Line 25 (Eq. 25): Please mention that the way R (and $y_t^i$) is described, you assume uncorrelated measurement errors.*

**Reply: This information was added in the revised manuscript as suggested by the reviewer (line 317-320):**

**"*We assume the soil moisture measurement errors at each depth to have a fixed and common variance $\sigma^2$, and to be uncorrelated in space and time. Thus we can write $R = \sigma^2 I_m$, where $I_m$ signifies the $m \times m$ identity matrix with zeros everywhere except on the main diagonal which stores values of $\sigma^2$.*"**

*Page 7, Line 25 (Eq. 33): Please describe the implications of employing this method. How does the performance of the filter depend on the choice of initial uncertainty?*

**Reply: Filter inbreeding is a problem associated with EnKF. The ensemble spread may narrow down in the course of parameter estimation so that most of the ensemble members would become very close to the ensemble mean value, which is called filter inbreeding and which might cause filter divergence [*Franssen and Kinzelbach*, 2008; *Han et al.*, 2014; *Whitaker and Hamill*, 2012]. The inflation method used in our work has been proven to be an efficient method to avoid filter inbreeding [*Han et al.*, 2014; *Whitaker and Hamill*, 2012]. In the revised manuscript, we used VIC model to test the effect of initial uncertainties on the performance of EnKF by**

**increasing the forcing error from 10% to 20%. The additional simulation results were explained in the new manuscript (line 759-763):**

"*The effect of initial uncertainties on the performance of EnKF with the ensemble inflation method is also tested with the VIC-3L model. Table 6 compares the RMSE values of EnKF-AUG and EnKF-DUAL for the calibration and evaluation period using heteroscedastic precipitation data errors equivalent to 10% (default) and 20% of their measured hourly rates plotted in Figure 2. We list separate RMSE values for each soil moisture measurement depth. In short, the results are equivalent for both EnKF implementations.*"

*Page 9, Line 13. How did you choose the tuning parameter s? How does it influence the performance? Please include the choice of s for the PF and initial uncertainties of the EnKF when comparing different methods.*

**Reply:** The optimal tuning parameter *s* is hardly known in applications [*Yan et al.*, 2015]. It was set to 0.01 in some applications [*DeChant and Moradkhani*, 2012; *Plaza et al.*, 2012]. In our work, to keep particle spread, *s* was set to 0.1. We tested other values for parameter *s*, like 0.01 and 0.5, to see how it influences the performance. The additional simulation results were explained in the manuscript (line 764-803):

"*Next, we evaluate the effect of the choice of the scaling factor $s$ in RRPF on VIC-3L output. This scalar plays a crucial role in the resampling of the parameters in the PF. If $s$ is taken too large, the resampling step will introduce parameter drift and corrupt the approximation of $p(X_{1:t}|\widetilde{Y}_{1:t})$ and $p(x_t|\widetilde{Y}_{1:t})$. On the contrary, if $s$ is too small, then the resampled parameters exhibit insufficient dispersion, and underestimate the actual parameter uncertainty. In the absence of theoretical convergence proofs and clear guidelines on the selection of $s$, the RRPF cannot estimate exactly the posterior state and parameter PDF (Vrugt et al., 2013; Yan et al., 2015). Previous applications of RRPF have suggested a value of $s = 0.01$ (DeChant and Moradkhani, 2012; Plaza et al., 2012), but thus far we have used $s = 0.1$ to avoid sample impoverishment. Table 7 lists RMSE values of VIC-3L for the 5, 20, and 50 cm measurement depth for the calibration and evaluation period using RRPF with $s = 0.01$, $s = 0.1$, and $s = 0.5$, respectively. These three runs are coined RRPF_0.01, RRPF_0.1 and RRPF_0.5, respectively. These results demonstrate that a value of $s = 0.5$ significantly enhances the performance of RRPF during the calibration period. The RMSE values are reduced from 0.025, 0.012, and 0.113 to 0.015, 0.007, and 0.037 for the 5, 20 and 50 cm measurement depths. RRPF_0.5 also shows substantial improvements over RRPF_0.01 during the evaluation period. This improvement is most apparent for the 20 and 50 cm soil depths with RMSE values that have decreased from 0.025 and 0.119 to 0.020 and 0.071, respectively. These results are on par with our default setting of $s = 0.1$ in RRPF. These findings provide evidence for our claim that the scaling factor $s$ plays a crucial role in RRPF. What is more, it provides support for our conclusion in Fig. 5 that RRPF underestimates the actual uncertainty of $\log_{10}k_s$ and β. Larger values of $s$ will increase the parameter spread, which in turn will enhance the uncertainty among the particles' forecasted states. This makes it easier for RRPF to track the observed soil moisture data during the calibration period.*

*Figure 8 displays traceplots of the sampled $N = 100$ trajectories of the saturated hydraulic conductivity ($\log_{10}k_S$ in m/s) at 50 cm depth (left column) and parameter β (right column) of VIC-3L during the 5-month assimilation period using (A-B) RRPF_0.01, (C-D) RRPF_0.1, and (E-F) RRPF_0.5. As expected, larger values of $s$ increase the spread of the sampled values of the VIC-3L parameters as evidenced by an increasingly larger particle coverage of the prior parameter distribution. This larger spread of the particles' parameter values also enhances the ability of RRPF to track properly the joint parameter and state PDF. Yet, larger values of $s$ have two important*

*drawbacks. Not only can it obstruct parameter convergence (as evidenced in Fig. 8e), but also many of the resampled parameter values might be deemed nonbehavioral, enhancing considerably the chances of particle degeneration. To demonstrate this more explicitly, Figure 9 shows traceplots of the VIC-3L predicted soil moisture contents of the $N = 100$ particles at 50 cm depth using (A) RRPF_0.01, (B) RRPF_0.1, and (C) RRPF_0.5. The RRPF is excessively optimistic for $s = 0.01$ with a negligible uncertainty in the predicted soil moisture values between weeks 2-14. Note that in weeks 2-4 the ensemble has collapsed to a deterministic simulation (appears as single line). A similar result is observed for RRPF_0.1 but with enhanced uncertainty in soil moisture values for the second part of the calibration period. The use of $s = 0.5$ enhances considerably the spread of the VIC-3L soil moisture predictions. Yet, the ensemble spread has become quite large from week 15 onwards. For these reasons, we are satisfied with our value of $s = 0.1$ in RRPF, although this decision is subjective and would require much testing via trial-and-error. This has stimulated Vrugt et al. (2013) to introduce a parameter resampling method which is properly rooted in statistical theory and uses laws of probability to rejuvenate the ensemble.”*

**Also in the discussion (line 936-938):**

*“Moreover, the use of a larger value of the scaling s in RRPF reduced considerably the RMSE values of VIC-3L in the calibration data period, particularly at the 50 cm measurement depth, whereas model performance was hardly improved during the evaluation period.”*

*Page 15 Line 32: You state: "It is not surprising that the EnKF is more efficient and effective than the PF." I would follow this statement in case of strictly Gaussian distributions and linear measurement operators. In those cases the EnKF is expected to outperform the PF. Nevertheless, when dealing with non-linear processes that challenge the Gaussian assumption, the better performance is not clear at all.*

Reply: **In the new manuscript we discussed this part with the results of CLM, because for VIC, EnKF and PF performed similarly. We tried to formulate more precisely in the new manuscript (line 939-962):**

*“For CLM, larger differences were observed in the performance of the different data assimilation methods. This larger disparity among the methods is explained by the considerably larger number of soil layers (ten) used by CLM. This increased significantly the dimensionality of the parameter estimation problem. The overall best results at the 5, 20 and 50 cm measurement depths were observed for EnKF-AUG and EnKF-DUAL with RMSE values that were somewhat smaller than their counterparts derived from PMCMC. This was true for both the calibration and evaluation periods. The RRPF exhibited the worst performance, in part determined by the use of a relatively small ensemble of N = 100 particles. The superiority of the EnKF-AUG and EnKF-DUAL methods for CLM is consistent with our expectations articulated previously in Section 3.1. The analysis step of the EnKF makes it much easier for EnKF-AUG and EnKF-DUAL to track the measured soil moisture dynamics, thereby promoting convergence in high-dimensional state-parameter spaces. PF-based methods, on the contrary, deteriorate in robustness and efficiency with larger dimensionality of the state-parameter space as they lack a state analysis step and approximate the transient state-parameter PDF via the particles' likelihoods. This likelihood is only a low-dimensional summary statistic of the distance between the forecasted and measured values of the states. Resampling with MCMC via the likelihood thus becomes increasingly more difficult in high-dimensional state-parameter spaces. For CLM, the PMCMC method still achieves comparable results to EnKF-AUG and EnKF-DUAL as the dimensionality of the state-parameter PDF of this model is only somewhat larger than its counterpart of VIC-3L. Of course, the use of a larger ensemble size makes it easier to characterize the transient state-parameter PDF, but at the expense of a significantly increased CPU-cost. For PMCMC, multiple different MCMC resampling steps can also enhance significantly the particle ensemble by allowing each particle trajectory to improve its likelihood. Yet, this deteriorates significantly the efficiency of implementation as each candidate*

*particle requires a separate model evaluation of VIC-3L or CLM to determine its likelihood. Thus, for LSMs with relatively few state variables and model parameters, we expect the EnKF and PF methods to achieve a comparable performance. For larger dimensional state-parameter spaces we would recommend EnKF-AUG and EnKF-DUAL, unless one can afford a very large number of particles.*"

*Page 25, Table 1: The EnKF assumes Gaussian distributions. What is the reason that you sample for all but one parameter from an initial uniform distribution? What are the implications for the chosen inflation method?*

**Reply: We want to compare EnKF and PF starting from the same prior distribution in order to make a more meaningful comparison. It is right that EnKF assumes a Gaussian distribution, but the PF not. We believe that assuming an initial uniform distribution is a neutral assumption good for comparing EnKF and PF. This was clarified in the revised version of the manuscript (line 599-601):**

**"*We want to compare EnKF and PF starting from the same prior distribution in order to make a more meaningful comparison. EnKF assumes a Gaussian distribution, but the PF not. We believe that assuming an initial uniform distribution is a neutral assumption good for comparing EnKF and PF.*"**

*Page 16, Lines 1-2: You state: "The value of the likelihood does generally not say anything about how close the forecasted variables are to their measured counterparts." I disagree, the likelihood yields information about the distance.*

**Reply: We deleted this statement in the new manuscript. But we still argue that the likelihood yields information about the relative, but not the absolute distance between forecasted variables and measurements. We can say that particles with higher weights are closer to the measurements but we cannot tell how close they are to the measurements.**

*Discussion of results: Since the results show a wealth of information. I would appreciate a detailed discussion. Especially consider the following points in detail and incorporate them into your conclusion:*

*Page 29, Figure 3: The large deviations in the Particle Filter might hint at filter inbreeding in the states. You observe the deviation but do not discuss it and actually exclude the possibility of inbreeding by investigating the parameters: "A too narrow spread of ensemble members would lead to filter divergence. For the state augmentation (AUG) and dual estimation (DUAL), the spread of the ensemble members is kept large enough during the whole assimilation period as the ensemble inflation method helped to keep adequate ensemble spread. RRPF and MCMCPF also have enough ensemble spread because of parameter perturbation and MCMCPF resampling." (Page 13 Lines 15-19). Please address the question of adequate ensemble spread in the states by actually showing the ensemble of states there.*

**Reply: We showed the evolution of the state ensemble of PF in our revised manuscript and addressed this question (line 792-798):**

**"*Figure 9 shows traceplots of the VIC-3L predicted soil moisture contents of the N=100 particles at 50 cm depth using (A) RRPF_0.01, (B) RRPF_0.1, and (C) RRPF_0.5. The RRPF is excessively optimistic for s=0.01 with a negligible uncertainty in the predicted soil moisture values between weeks 2-14. Note that in weeks 2-4 the ensemble has collapsed to a deterministic simulation (appears as single line). A similar result is observed for RRPF_0.1 but with enhanced uncertainty in soil moisture values for the second part of the calibration period. In PF_0.1 particle degeneration from March to June explains its bad performance from March to June in Fig. 3.*"**

*Page 30 Figure 4: During the calibration period the filter without parameter estimation performs better. Please discuss possible reasons.*

**Reply: When only states (soil moisture content) are assimilated, states are updated directly by observations. However, when states and parameters are updated jointly, the nonlinear relation between states and parameters is considered, which may introduce inconsistency. We further discuss these results in the paper (line 683-686):**

*"Perhaps surprisingly, but the best performance of VIC-3L is obtained for state estimation only (noParamUpdate) using model parameterizations drawn randomly from the prior parameter distribution. We posit that the nonlinear relationship between states and parameters may introduce inconsistencies in PMCMC, RRPF, EnKF-AUG and EnKF-DUAL which jointly estimate VIC-3L states and parameters."*

*Page 34, Figure 8: There is basically no difference for the prediction of the water content whether there are parameters estimated or not. Please discuss why this is the case.*

**Reply: Predictions of soil moisture content for layer 2 and layer 3 (in the verification period) improved significantly for the case of parameter estimation. Concerning the soil moisture content of layer 1, the RMSE value of the open loop run is 0.053m³/m³, which is already quite close to the observed values. In addition, the soil moisture content for the upper layer is strongly driven by single precipitation events. We extended the discussion of these results (line 740-746):**

*"In general, the RMSE values of the evaluation period are much higher than their counterparts of the assimilation period, and noParamUpdate produces RMSE values similar to that of an open loop simulation. VIC-3L parameter estimation is productive, as it substantially reduces the RMSE values of 20 and 50 cm measurement depths compared to a model run with state estimation only (noParamUpdate) and parameters drawn randomly from their prior distribution. More specifically, the PMCMC, RRPF, EnKF-AUG and EnKF-DUAL show a RMSE improvement of about 54% and 42% for the second and third measurement depth compared to OpenLoop and noParamUpdate."*

See also the comment below.

*Figures 7, 8, 12, 13: Especially water content of the top layer can almost not be represented by either model, although improved with state of the art data assimilation methods. Please discuss this including the representation of the physics in the models and implications of the perfect model assumption.*

**Reply: We provided additional discussion of the results and discuss reasons for the larger deviations in the fit (line 877-881):**

*"Despite this improvement in model performance over an open-loop simulation, VIC-3L and CLM do not adequately characterize soil moisture dynamics of the top layer (5 cm measurement depth) during the evaluation period (RMSE values of about 0.05 cm³/cm³). We posit that these two models do not characterize adequately processes such as water infiltration, soil evaporation, and/or root water uptake (transpiration), which govern rapid variations in soil moisture storage in the top soil."*

*The difference in the assimilation methods is small. Please discuss if this difference is significant. Do you expect the same results for other applications? What is the influence of specific filter settings on the performance?*

**Reply: The performance of the four data assimilation algorithms in which states and parameters are jointly estimated does not differ very much in our study. Nevertheless, the small difference in performance between EnKF and PF based algorithms indicates that PF is also an efficient data**

assimilation algorithm for problems of this size. It can be expected that larger ensemble numbers can improve the performance of EnKF and PF based algorithms. For MCMCPF, multiple MCMC resampling steps can also help improve performance. Given the CPU-intensity of the calculations a larger comparison is beyond the scope of this work. In our work, all algorithms are evaluated with real-world data, so we think our results are meaningful for other applications. We expect that for example with more unknowns (i.e., 2D and 3D-applications) EnKF-based algorithms will perform better than PF, as PF will become extremely CPU-intensive and needs many more particles. We evaluated the impact of filter settings on the results of this paper, like the impact of initial conditions on the inflation method for EnKF, the effect of tuning parameter *s* on PF and how the ending date of assimilation affects the verified results. However, it will be difficult to evaluate the significance of the difference between the DA-algorithms on the basis of this single study.

We revised the text in the discussion (line 931-962):

"*The different data assimilation methods (EnKF-AUG, EnKF-DUAL, RRPF and PMCMC) led to a rather similar performance of VIC-3L during the calibration and evaluation period. The only exception to this was the anomalous RMSE value of RRPF at the 50 cm measurement depth during the calibration period. This was explained by the slow convergence of the maximum baseflow velocity in RRPF. Our results for VIC-3L further demonstrated that the results of EnKF-AUG and EnKF-DUAL were equivalent for a 10% and 20% rainfall error. Moreover, the use of a larger value of the scaling s in RRPF reduced considerably the RMSE values of VIC-3L in the calibration data period, particularly at the 50 cm measurement depth, whereas model performance was hardly improved during the evaluation period.*

*For CLM, larger differences were observed in the performance of the different data assimilation methods. This larger disparity among the methods is explained by the considerably larger number of soil layers (ten) used by CLM. This increased significantly the dimensionality of the parameter estimation problem. The overall best results at the 5, 20 and 50 cm measurement depths were observed for EnKF-AUG and EnKF-DUAL with RMSE values that were somewhat smaller than their counterparts derived from PMCMC. This was true for both the calibration and evaluation periods. The RRPF exhibited the worst performance, in part determined by the use of a relatively small ensemble of N = 100 particles. The superiority of the EnKF-AUG and EnKF-DUAL methods for CLM is consistent with our expectations articulated previously in Section 3.1. The analysis step of the EnKF makes it much easier for EnKF-AUG and EnKF-DUAL to track the measured soil moisture dynamics, thereby promoting convergence in high-dimensional state-parameter spaces. PF-based methods, on the contrary, deteriorate in robustness and efficiency with larger dimensionality of the state-parameter space as they lack a state analysis step and approximate the transient state-parameter PDF via the particles' likelihoods. This likelihood is only a low-dimensional summary statistic of the distance between the forecasted and measured values of the states. Resampling with MCMC via the likelihood thus becomes increasingly more difficult in high-dimensional state-parameter spaces. For CLM, the PMCMC method still achieves comparable results to EnKF-AUG and EnKF-DUAL as the dimensionality of the state-parameter PDF of this model is only somewhat larger than its counterpart of VIC-3L. Of course, the use of a larger ensemble size makes it easier to characterize the transient state-parameter PDF, but at the expense of a significantly increased CPU-cost. For PMCMC, multiple different MCMC resampling steps can also enhance significantly the particle ensemble by allowing each particle trajectory to improve its likelihood. Yet, this deteriorates significantly the efficiency of implementation as each candidate particle requires a separate model evaluation of VIC-3L or CLM to determine its likelihood. Thus,*

*for LSMs with relatively few state variables and model parameters, we expect the EnKF and PF methods to achieve a comparable performance. For larger dimensional state-parameter spaces we would recommend EnKF-AUG and EnKF-DUAL, unless one can afford a very large number of particles.*"

*Please improve the explanations on the Particle Filter and the two LSMs. Page 7 Line 32 - Page 8 Line 11: I do understand Particle Filters, but the given explanation is not clear. Especially clarify your description of transition and proposal densities.*

**Reply: We revised this part and improved the formulation in our manuscript (line 387-475):**

"***The PF was first suggested in the research area of object recognition, robotics and target tracking (Gordon et al., 1993) and was introduced to hydrology by Moradkhani et al. (2005a). The PF differs from the EnKF in that it describes the evolving probability density function (PDF) of the LSM state variables by a set of $N$ random samples, also called particles. Each particle carries a non-zero weight which determines its underlying probability, and these weights are updated as soon as a new datum (observation) becomes available. Before we proceed with a brief theoretical description of the PF we must first explicate our notation. We denote with symbol $X_{1:t}$ the collection of simulated values of the LSM state variables between the first observation at $t = 1$ and the present datum, $t$, hence $X_{1:t} = [x_1, \dots, x_t]$ is a $k \times t$ matrix with the LSM states at each measurement time stored as a column vector. The corresponding observations are stored in the $m \times t$ matrix, $\widetilde{Y}_{1:t} = [\widetilde{y}_1, \dots, \widetilde{y}_t]$. Finally, we use braces, $\{\cdot\}$, to denote our Monte Carlo ensemble of $N$ particle trajectories, $\{X_{1:t}^{1:N}\}$, and thus $\{x_t^{1:N}\}$ is a $k \times N$ matrix with sampled values of the LSM state variables at time $t$. The subsequent description of the PF follows closely the description of Vrugt et al. (2013). Interested readers are referred to this publication for further details.***

*If we assume the parameters to be known, then we can write the evolving posterior distribution, $p_\alpha(X_{1:t}|\widetilde{Y}_{1:t})$, for the state-space formulation of equation (2) as follows:*

$$p_\alpha(X_{1:t}|\widetilde{Y}_{1:t}) = \overbrace{p_\alpha(X_{1:t-1}|\widetilde{Y}_{1:t-1})}^{prior} \frac{\overbrace{\mathcal{M}_\alpha(x_t|x_{t-1})}^{model} \overbrace{L_\alpha(\widetilde{y}_t|x_t)}^{likelihood\ function}}{\underbrace{p_\alpha(\widetilde{y}_t|\widetilde{Y}_{1:t-1})}_{normalization\ constant}}, \qquad (14)$$

*where $p_\alpha(X_{1:t-1}|\widetilde{Y}_{1:t-1})$ denotes the prior state distribution, $\mathcal{M}_\alpha(x_t|x_{t-1})$ signifies the transition probability density of the state variables (= equation (2)), $L_\alpha(\widetilde{y}_t|x_t)$ is the likelihood function, and $p_\alpha(\widetilde{y}_t|\widetilde{Y}_{1:t-1})$ represents a normalization constant which ensures that the posterior state distribution integrates to unity. Equation (14) follows directly from Bayes' law (see Appendix A of Vrugt et al. (2013)), and does not use at once the data up to time $t$ to estimate $p_\alpha(X_{1:t}|\widetilde{Y}_{1:t})$ but rather estimates the evolving system state recursively over time using some mathematical model and new incoming measurements. If we integrate out the state trajectory $X_{1:t-1}$ from equation (14) then we can derive an expression for the marginal PDF of the state variables, $p_\alpha(x_t|\widetilde{Y}_{1:t})$, at time $t$:*

$$p_\alpha(x_t|\widetilde{Y}_{1:t}) = \frac{L_\alpha(\widetilde{y}_t|x_t)p_\alpha(x_t|\widetilde{Y}_{1:t-1})}{p_\alpha(\widetilde{y}_t|\widetilde{Y}_{1:t-1})}, \qquad (15)$$

*which is also referred to as the update step of the optimal filter (conditional independence of measurements). The state prediction step is equivalent to the Chapman-Kolmogorov equation:*

$$p_\alpha(x_t|\widetilde{Y}_{1:t-1}) = \int_\Omega \mathcal{M}_\alpha(x_t|x_{t-1})\,p_\alpha(x_{t-1}|\widetilde{Y}_{1:t-1})dx_{t-1},\qquad(16)$$

*where $\Omega$ signifies the feasible state space.*

*We conveniently assume herein, a Gaussian likelihood function:*

$$L_\alpha(\widetilde{y}_t|x_t) = \frac{1}{(2\pi)^{m/2}|R|^{1/2}}exp\left(-\frac{1}{2}(\widetilde{y}_t - H_x x_t)^T R^{-1}(\widetilde{y}_t - H_x x_t)\right),\qquad(17)$$

*where $R$ is the $m \times m$ measurement error covariance matrix, $|\cdot|$ signifies the determinant operator, and $m$ denotes the length of the observation vector, $\widetilde{y}_t$, at time $t$.*

*The PF makes use of the following identity of equation (14) to approximate the evolving state PDF:*

$$p_\alpha(X_{1:t}|\widetilde{Y}_{1:t}) \propto p_\alpha(X_{1:t-1}|\widetilde{Y}_{1:t-1})\mathcal{M}_\alpha(x_t|x_{t-1})L_\alpha(\widetilde{y}_t|x_t).\qquad(18)$$

*This recursion implies that we can use reuse the particles (samples) at $t-1$ that define the prior distribution, $p_\alpha(X_{1:t-1}|\widetilde{Y}_{1:t-1})$, to approximate the posterior state PDF, $p_\alpha(X_{1:t}|\widetilde{Y}_{1:t})$, at the next observation time. Yet, such recycling poses a problem, that is, we cannot sample directly from $p_\alpha(X_{1:t}|\widetilde{Y}_{1:t})$ as we do not know its multivariate distribution. We therefore resort to an easy-to-sample-from importance density, $q_\alpha(\cdot|x_{t-1},\widetilde{y}_t)$, and draw $\{x_t^{1:N}\}$ taking into consideration the current observation, $\widetilde{y}_t$, and previous state samples, $\{x_{t-1}^{1:N}\}$. We then calculate the unnormalized importance weight of the ith particle, $W_t^i$, as follows*

$$W_t^i \propto \overline{W}_{t-1}^i w_t(\{X_{1:t}^i\}),\qquad(19)$$

*where $w_t(X_{1:t}^i)$ signifies the incremental importance weight:*

$$w_t(\{X_{1:t}^i\}) = \frac{\mathcal{M}_\alpha(\{x_t^i\}|\{x_{t-1}^i\})L_\alpha(\widetilde{y}_t|\{x_t^i\})}{q_\alpha(\{x_t^i\}|\{x_{t-1}^i\},\widetilde{y}_t)},\qquad(20)$$

*and $\overline{W}_t^i = W_t^i/\sum_{i=1}^N W_t^i$ denote the normalized importance weights, which vary between 0 and 1.*

*Before we can implement the PF in practice, we need to specify the importance density, $q_\alpha(\cdot|\{x_{t-1}^{1:N}\},\widetilde{y}_t)$, for $t = \{2,\dots,n\}$. We follow Gordon et al. (1993) and set $q_\alpha(x_t|x_{t-1},\widetilde{y}_t) = \mathcal{M}_\alpha(x_t|x_{t-1})$ which results in the following equation for the incremental particle weights:*

$$w_t(\{X_{1:t}^i\}) = \frac{\mathcal{M}_\alpha(\{x_t^i\}|\{x_{t-1}^i\})L_\alpha(\widetilde{y}_t|\{x_t^i\})}{\mathcal{M}_\alpha(\{x_t^i\}|\{x_{t-1}^i\})} = L_\alpha(\widetilde{y}_t|\{x_t^i\}).\qquad(21)$$

*This approach gives satisfactory results if the transition density or model operator, $\mathcal{M}_\alpha(x_t|x_{t-1})$, adequately describes the observed system dynamics, and/or the observations, $\widetilde{Y}_{1:t}$, are not too informative. Otherwise, the repeated application of equation (19) causes particle impoverishment in which the sampled particle trajectories drift away from the actual posterior state distribution, and receive a negligible importance weight. This ensemble degeneracy (e.g. Carpenter et al., 1999) deteriorates PF performance and results in a poor computational efficiency of the filter as much of*

*the CPU-time is devoted to carrying forward particle trajectories whose contribution to* $p_\alpha(X_{1:t}|\widetilde{Y}_{1:t})$ *for* $t > 1$ *is virtually zero.*

*To combat particle degeneracy we monitor the effective sample size (ESS) after assimilation of each new observation:*

$$ESS = 1/\sum_{i=1}^{N}(\overline{W}_t^i)^2. \tag{22}$$

*If the ESS is smaller than some default threshold, say* $N/2$, *then the particle ensemble is said to be degenerating. Several methods have been developed in the statistical literature to rejuvenate the particle ensemble. Gordon et al. (1993) introduced Sequential Importance Resampling (SIR), where* $N$ *particles are drawn from the ensemble using selection probabilities equal to their normalized importance weights. This step replaces samples with low importance weights with exact copies of the most promising particles, and produces a resampled set of* $N$ *particles with equal weights of* $1/N$. *In our application of the PF we implement Residual Resampling (RR) developed by Liu and Chen (1998). This method has an important advantage over SIR in that it produces a resampled set of particles with more diverse weights (Weerts and Serafy, 2006). First, we compute a selection probability,* $p_{\{x_t^i\}}$, *of each individual particle as follows:*

$$p_{\{x_t^i\}} = \frac{N\overline{W}_t^i - \lfloor N\overline{W}_t^i \rfloor}{N - M}, \tag{23}$$

*where the* $\lfloor \cdot \rfloor$ *operator rounds down to the nearest integer, and* $M = \sum_{j=1}^{N} \lfloor N\overline{W}_t^j \rfloor$. *Then, the* $M$ *particles with largest normalized importance weights are retained, and the remaining* $N - M$ *spots are filled by drawing from the* $M$ *retained particles using their selection probabilities from equation (23). The resulting filter is referred to as RRPF.*

*In the present application of the RRPF, we not only estimate the LSM states but also jointly infer the values of the model parameters. We use state augmentation and add the model parameters to the vector of LSM state variables. Yet, this approach requires definition of an importance density for the parameters to avoid parameter impoverishment after several successive assimilation steps. This has been demonstrated numerically by Plaza et al. (2012) using a series of data assimilation experiments. In principle, we could corrupt the posterior parameter distribution using the ensemble inflation method of Whitaker and Hamill (2012) detailed in equation (13). This approach was used by Qin et al. (2009) to avoid degeneracy of the parameter values. Instead, we use the approach described by Plaza et al. (2012) and perturb the parameter values of the resampled particles using draws from a zero-mean* $d$-*variate Gaussian distribution with diagonal covariance matrix. This* $d \times d$ *matrix has zero entries everywhere (uncorrelated dimensions) except on the main diagonal which stores values of* $s^2 Var[\{\alpha_{0,j}^{1:N}\}]$, *where* $s$ *is a scaling factor,* $Var[\{\alpha_{0,j}^{1:N}\}]$ *signifies the prior variance of the jth parameter (at* $t = 0$), *and* $j = \{1, ..., d\}$. *This is an adaptation of the method*

*introduced by Moradkhani et al. (2005b) and uses the prior variance of the parameters rather than their variance at the previous measurement time, $t-1$. Yet, in the absence of a formal guidelines on the choice of $s$, this perturbation approach suffers from a lack of adequate statistical underpinning [Vrugt et a., 2013; Yan et al., 2015]. In our present application, we set $s = 0.1$, and evaluate the RRPF performance for VIC-3L model using other values for this scaling factor as well."*

*Page 8 Lines 21-30: You describe SIR and RR. What is the reason to choose RR?*

**Reply: RR is developed from SIR by (*Liu and Chen*, 1998). RR is one of the most popular methods for PF to reduce particle degeneration. This method has an important advantage over SIR in that it produces a resampled set of particles with more diverse weights (*Weerts and Serafy*, 2006). This explanation was added in the revised version of the manuscript (line 451-453):**

**"*In our application of the PF we implement Residual Resampling (RR) developed by Liu and Chen (1998). This method has an important advantage over SIR in that it produces a resampled set of particles with more diverse weights (Weerts and Serafy, 2006).*"**

*For this study the understanding of the different LSM models is important, since the main difference in performance is attributed to different models. Because of that a good description is necessary. Although I am not an expert on LSMs, I noticed the following: Page 17, Line 15 (Eq. A2): The equation describes the soil water movement in the top two layers. To me it was not clear if this description is valid for both layers individually. If so, why is the precipitation P and evaporation E the same?*

**Reply: This was corrected in the revised version (line 1040-1053):**

**"*Now we have discussed the different fluxes from the soil domain simulated by VIC-3L we can now write differential equations of the moisture dynamics in the individual soil layers (see also Liang et al., 1996).*"**

$$\frac{\partial \theta_1}{\partial t} z_1 = P + Q_{1,2} - Q_d - R_1 - E$$
$$\frac{\partial \theta_2}{\partial t} z_2 = Q_{2,3} - Q_{1,2} - R_2 \qquad , \qquad (A7)$$
$$\frac{\partial \theta_3}{\partial t} z_3 = -Q_{2,3} - R_3 - Q_b$$

*where $Q_{i,i+1}$ [LT$^{-1}$] is the vertical flux of water between two adjacent soil layers $i$ and $i+1$, $R_i$ [LT$^{-1}$] signifies the root water uptake of the ith layer, and $i = \{1, 2, 3\}$. Downward fluxes are negative to be consistent with convention used in soil hydrology. The canopy transpiration flux is equal to the total water uptake by the plant roots, thus $T = R_1 + R_2 + R_3$. All three soil layers contain roots and thus contribute to transpiration in our application of VIC-3L to the Rollesbroich site. The vertical flux of water between two adjacent soil layers is assumed to be equivalent to the hydraulic conductivity of the upper layer. VIC-3L computes the hydraulic conductivity of each soil layer using the formulation of Brooks and Corey (1988):*

$$Q_{i,i+1} = -k_{s,i} \left( \frac{\theta_i - \theta_{r,i}}{\phi_i - \theta_{r,i}} \right)^{\beta_i} \quad (i = 1, 2), \qquad (A8)$$

*where $k_{s,i}$ [LT⁻¹] and $\theta_{r,i}$ [L³L⁻³] signify the saturated hydraulic conductivity and the residual volumetric moisture content of the ith soil layer, respectively. The minus sign at the right-hand-side of equation (A8) matches the direction of the flux. The dimensionless exponent $\beta_i$ should be larger than 3.0. "*

*Page 18, Line 1 (Eq. A7): The dimensions are inconsistent: You add [LT-1], [] and [L]. $i_m$ should be $I_m$ (or is not introduced). Please also explain the distinction of the cases $P+I < I_m$ and $P+I > I_m$.*

**Reply: $i_m$ is $I_m$. We revised this part in our manuscript (line 1025-1028):**

*"If the rainfall depth exceeds the available moisture capacity of the soil, $(I_{max} - I_0)$, then the excess precipitation is removed via surface runoff. Otherwise, if $P\Delta t \leq (I_{max} - I_0)$, then a large fraction of the rainfall will infiltrate depending on the soil's available storage and the spatial variability of the moisture capacity within the grid cell."*

*Page 19, Lines 18-22: the Richards equation is formulated for a continuum. Please explain the modifications and the implications of applying it to layers. Please give the extent of these layers.*

**Reply: in the revised manuscript, table 2 was added to show the extent of the 10 layers, and text about the Richards equation was revised (line 1092-1108):**

*"A modified Richards' equation is used to predict water storage and movement in the variably-saturated soils of the Rollesbroich site:*

$$\frac{\partial \theta_i}{\partial t} = \frac{\partial}{\partial z}\left[k_i\left(\frac{\partial(\psi_i + z_i - C_i)}{\partial z}\right)\right] - R_i = \frac{\partial}{\partial z}\left[k_i\left(\frac{\partial(\psi_i - \psi_{e,i})}{\partial z}\right)\right] - R_i, \tag{B6}$$

*where $\theta_i$ [L³L⁻³], $\psi_i$ [L], $k_i$ [LT⁻¹], $z_i$ [L], and $\psi_{e,i}$ [L] denote the volumetric water content, matric head, hydraulic conductivity, depth, and equilibrium matric head of the $i^{th}$ soil layer, $C_i = \psi_{e,i} + z_i$, and $R_i$ [T⁻¹] is the loss of water via root water uptake (canopy transpiration). Note that Equation (B6) omits conveniently the evaporation flux from the first (top) layer. The hydraulic conductivity, $k_i$, of each layer depends on its moisture content, saturated hydraulic conductivity, and exponent B, and these values of the adjacent soil layer immediately below, with the exception of the bottom layer (Oleson et al., 2013; Han et al., 2014). The use of the constant $C_i$ in Equation (B6) allows CLM to simulate matric head variations below the water table. This modification maintains a hydrostatic equilibrium soil moisture distribution, and fixes a critical deficiency of the θ-based formulation of Richards' equation (Zeng and Decker, 2009; Oleson et al., 2013).*

*The matrix head, $\psi_i$, and equilibrium matric head, $\psi_{e,i}$, of each soil layer are computed as follows:*

$$\psi_i = \psi_{s,i}\left(\frac{\theta_i}{\theta_{s,i}}\right)^{-B_i} \quad and \quad \psi_{e,i} = \psi_{s,i}\left(\frac{\theta_{e,i}}{\theta_{s,i}}\right)^{-B_i}, \tag{B7}$$

*with*

$$\theta_{e,i} = \theta_{s,i}\left(\frac{\psi_{s,i} + z_\nabla - z_i}{\psi_{s,i}}\right)^{(-1/B_i)}, \tag{B8}$$

*where $z_\nabla$ [L] is the depth of the water table. "*

**More explanation related to this question:**

**The unmodified Richards equation is:**

**The soil water flux *q* can be described by Darcy's law:**

$$q = -k\left(\frac{\partial(\psi+z)}{\partial z}\right) \tag{1}$$

**For one-dimensional vertical water flow in soils, the conservation of mass is stated as:**

$$\frac{\partial \theta}{\partial t} = -\frac{\partial q}{\partial z} - E = \frac{\partial}{\partial z}\left[k(\frac{\partial(\psi+z)}{\partial z})\right] - E \tag{2}$$

*E* **is the ET loss.** *Zeng and Decker* **(2009) note that this equation cannot maintain the hydrostatic equilibrium soil moisture distribution because of the truncation errors of the finite-difference numerical scheme. They show that this deficiency can be overcome by subtracting the equilibrium state as:**

$$\frac{\partial \theta}{\partial t} = \frac{\partial}{\partial z}\left[k(\frac{\partial(\psi+z-C)}{\partial z})\right] - E \tag{3}$$

**Where C is a constant hydraulic potential above the water table z$_\nabla$:**

$$C = \psi_e + z \tag{4}$$

**Substitution of equation (4) into equation (3) yields the modified Richards equation (B7):**

$$q = -k\left(\frac{\partial(\psi-\psi_e)}{\partial z}\right) \quad \text{and} \quad \frac{\partial \theta}{\partial t} = -\frac{\partial q}{\partial z} - E = \frac{\partial}{\partial z}\left[k(\frac{\partial(\psi-\psi_e)}{\partial z})\right] - E \tag{5}$$

**Implication of this modified method: Richards equation (2) used a θ-based solution which cannot account for the variation of ψ below water table because θ is constant (at saturated value) while ψ varies temporally and spatially, which leads to the failure to maintain the hydrostatic equilibrium soil moisture distribution. However, the modified Richards equation in which constant hydraulic potential C is explicitly subtracted at each time step can fix this deficiency. Details about the implementation of the modified method can be seen in [*Zeng and Decker*, 2009].**

**Below we explain the application of the modifications to layers. These details are not presented in the revised manuscript and we refer to the CLM-manual (Oleson et al., 2013) where more details can be found. Table 2 shows the soil layer definition in CLM for *N$_{levsoi}$* =10 layers.**

**Table 2. Division of a soil column in layers in CLM. Layer node depth(*z*), thickness(Δ*z*), and depth at layer interface(*z$_h$*) for 10 soil layers. Unit is meter.**

| Layer *i* | *z* | Δ*z* | *z$_h$* |
|---|---|---|---|
| 1 (top) | 0.0071 | 0.0175 | 0.0175 |
| 2 | 0.0279 | 0.0276 | 0.0451 |
| 3 | 0.0623 | 0.0455 | 0.0906 |
| 4 | 0.1189 | 0.0750 | 0.1655 |
| 5 | 0.2122 | 0.1236 | 0.2891 |
| 6 | 0.3661 | 0.2038 | 0.4929 |
| 7 | 0.6198 | 0.3360 | 0.8289 |
| 8 | 1.0380 | 0.5539 | 1.3828 |

| 9 | 1.7276 | 0.9133 | 2.2961 |
|---|---|---|---|
| 10 | 2.8646 | 1.5058 | 3.8019 |

**Numerical solution of equation (5) for soil layer $i$:**

$$\Delta z_i \frac{\Delta\theta_i}{\Delta t} = -q_{i-1}^{t+1} + q_i^{t+1} - e_i \qquad (6)$$

where $\Delta t$ is the time step, $\Delta z_i$ is soil layer thickness, $\Delta\theta_i = \theta_i^{t+1} - \theta_i^t$, $q_i$ is the outgoing flux of water from layer $i$ to layer $i+1$, $q_{i-1}$ is the incoming flux of water from layer $i-1$ to layer $i$ and $e_i$ is layer-averaged ET loss. The water fluxes ($q_{i-1}^{t+1}$ and $q_i^{t+1}$) in equation (6) are linearized about $\theta$ using Taylor series expansion which results in a general tridiagonal equation set of the form [*Oleson et al.*, 2013]:

$$r_i = a_i\Delta\theta_{i-1} + b_i\Delta\theta_i + c_i\Delta\theta_{i+1} \qquad (7)$$

$a_i$, $b_i$, $c_i$ and $r_i$ will be discussed under different conditions below. This tridiagonal equation set is solved over $i=1,\ldots, N_{levsoi}+1$ where $i = N_{levsoi}+1$ is a virtual layer representing the aquifer.

For $i=1$, the boundary condition is the infiltration rate, and

$$a_i = 0$$

$$b_i = \frac{\partial q_i}{\partial\theta_i} - \frac{\Delta z_i}{\Delta t}$$

$$c_i = \frac{\partial q_i}{\partial\theta_{i+1}}$$

$$r_i = q_{infl}^{t+1} - q_i^t + e_i$$

$q_{infl}^{t+1}$ is the infiltration into the soil which is partitioned between water input flux (sum of precipitation reaching the ground and melt water from snow), surface runoff, and surface water storage.

For $i=2, \ldots, N_{levsoi} - 1$,

$$a_i = -\frac{\partial q_{i-1}}{\partial\theta_{i-1}}$$

$$b_i = \frac{\partial q_i}{\partial\theta_i} - \frac{\partial q_{i-1}}{\partial\theta_i} - \frac{\Delta z_i}{\Delta t}$$

$$c_i = \frac{\partial q_i}{\partial\theta_{i+1}}$$

$$r_i = q_{i-1}^t - q_i^t + e_i$$

For the lowest soil layer ($i = N_{levsoi}$), the bottom boundary condition depends on the depth of the water table. If the water table is within the soil column, a zero flux boundary condition is applied ($q_i^t = 0$) and

$$a_i = -\frac{\partial q_{i-1}}{\partial\theta_{i-1}}$$

$$b_i = -\frac{\partial q_{i-1}}{\partial \theta_i} - \frac{\Delta z_i}{\Delta t}$$

$$c_i = 0$$

$$r_i = q_{i-1}^t + e_i$$

**And for the aquifer layer $i = N_{levsoi} + 1$:**

$$a_i = 0 \quad b_i = -\frac{\Delta z_i}{\Delta t} \quad c_i = 0 \quad r_i = 0$$

**If water table is below the soil column, for $i = N_{levsoi}$:**

$$a_i = -\frac{\partial q_{i-1}}{\partial \theta_{i-1}}$$

$$b_i = \frac{\partial q_i}{\partial \theta_i} - \frac{\partial q_{i-1}}{\partial \theta_i} - \frac{\Delta z_i}{\Delta t}$$

$$c_i = \frac{\partial q_i}{\partial \theta_{i+1}}$$

$$r_i = q_{i-1}^t - q_i^t + e_i$$

**And for the aquifer layer $i = N_{levsoi} + 1$:**

$$a_i = -\frac{\partial q_{i-1}}{\partial \theta_{i-1}}$$

$$b_i = -\frac{\partial q_{i-1}}{\partial \theta_i} - \frac{\Delta z_i}{\Delta t}$$

$$c_i = 0$$

$$r_i = q_{i-1}^t$$

**Upon solution of the tridiagonal equation set, soil water content is updated as:**

$$\theta_i^{t+1} = \theta_i^t + \Delta \theta_i \Delta z_i \tag{8}$$

*Specific comments: Page 5, Line 36: "Commonly used data assimilation algorithms are EnKF, PF and variants of them." 4D-Var is also commonly used.*

**Reply: We admit that 4D-Var is also commonly used. The sentence was adapted to account for this (line 292-297):**

**"*This includes the use of four-dimensional variational data assimilation (4D-Var), EnKF, PF, and related assimilation schemes. These methods have been applied successfully to a large number of different fields for model-data fusion in the atmospheric, oceanic, biogeochemical and hydrological sciences. We now briefly discuss the theory of four different data assimilation methods which are used herein with VIC-3L and CLM to characterize spatiotemporal soil moisture dynamics at our experimental site.*"**

*Page 6, Line 19: "H [...] is the identity matrix if y refers to in-situ ground measurements available at all grid cells." This is only the case if the same quantity as the state is observed. Otherwise the quantity has to be transferred. Additionally mention that H has to be linear for the EnKF.*

**Reply: We revised this part according to our application in the new manuscript (line 324-327):**

**"(…)** *and the $m \times k$ matrix H signifies the measurement operator which maps the model output to the measurement space. It is linear for EnKF. In our present application, we observe directly the soil moisture content of respective measurement depth, and thus the matrix H is made up of values of zero and unity.***"**

*Page 11 Lines 11-12: "The parameters of the other layers were updated with help of the calculated spatial covariance in case of EnKF." This statement implies that the parameters were not estimated with the PF? Please clarify.*

**Reply: Parameters are also estimated in PF. In PF, each particle includes a state vector and a parameter vector. But the weight vector is calculated only by the state vector. Both the state vector and parameter vector are resampled with help of the weight vector. We added the text to make this clearer (line 571-573):**

**"***A slightly different approach was followed in RRPF and PMCMC, in which the soil parameters of the unmeasured moisture layers in CLM were updated to their weighted-average values of the resampled particles using the vector of normalized importance weights***."**

*Page 12 Line 9-10: "A NSE value equal to 1 and RMSE equal to 0 imply a perfect prediction." This is wrong. A RMSE equal to the measurement uncertainty is perfect.*

**Reply: Thanks, we admit that for a prediction in the verification phase we cannot expect a better result than a RMSE equal to the measurement uncertainty but still an RMSE equal to 0 would be the best result. We modified the sentence to accommodate this result (line 645-646):**

**"***Larger values of the NSE and smaller values of the RMSE are preferred as they indicate a better LSM performance***."**

**Previous comments from reviewer I in the second review round**

*Major comments:*

*1. The expected impact of heterogeneity is now explained in line 376-380: "Qu et al. (2014) described the statistics of soil properties for soil samples taken in the Rollesbroich catchment. Soil texture showed a relatively limited variation. In our work only vertical heterogeneity is considered. In this case, heterogeneity does not seem to be very strong and we do not face a challenging upscaling case for the land surface model." However, Qu et al. (2014) seem to state that there is considerable heterogeneity, e.g.: "Spatial variability of the measured soil water content was higher at the 50-cm depth than the 5- and 20-cm depths, as indicated by the temporal dynamics of the standard deviation of soil water content presented in Fig. 2 (bottom panel). We attribute this to the pedological situation (shallow soil above consolidated bedrock) in which the highly variable stone content in the subsoil leads to considerable spatial variability of soil water content at the 50-cm depth."*

*Furthermore, the revised discussion also states (line 622-624): "The fact that we replace heterogeneous soil properties and soil moisture content for a given area by spatially homogeneous values, also introduces temporal variability in the effective parameters that are estimated in this study." To me this sounds like heterogeneity is a rather important challenge for upscaling. Please clarify. In this context (if heterogeneity is important) I didn't apprehend why the representation of photosynthesis was considered the most noteworthy model structural error (line 451-454): "The model error was set to zero assuming that uncertainty was captured by uncertain parameters and model forcings. However, we agree that it can be expected that we have other model structural errors, for example in relation to the representation of photosynthesis."*

**Reply:** **We thank the reviewer for pointing out the improvement of our manuscript. It is true that in spite of a limited spatial variability in soil texture characteristics the spatial variability in soil moisture content is not so small. This could be related to the influence of groundwater and the presence of a drainage system, but also to variations in soil hydraulic properties although texture is quite homogeneous. We decided therefore to reformulate the sentences cited to (line 542-552):**

*"In this work, we conveniently assume the soil-land-surface domain of the Rollesbroich site to be homogeneous and characterized by areal average values of soil moisture content at 5, 20 and 50 cm depth. In other words, we consider only vertical variations in soil water storage. Common LSM data assimilation experiments published in the literature usually involve application to much larger spatial scales, especially when remote sensing data are used. Hence, it is important to evaluate the LSM performance for a site where heterogeneities are neglected. Qu et al. (2014) investigated the geostatistical properties of the soils of the Rollesbroich test site. This work demonstrated a rather small spatial variability of the soil texture. This does not suggest, however that we can ignore spatial variations in the measured soil moisture values. Indeed, the standard deviations of soil moisture vary between 0.04 and 0.07 cm$^3$/cm$^3$ depending on the actual soil layer. This spatial heterogeneity of the soil moisture data documents variability in the soil hydraulic properties, and complicates the application and upscaling of LSMs."*

**We take the simple or biased representation of photosynthesis as an example of model structure error, but it doesn't mean that the representation of photosynthesis is the only most noteworthy model structural error. Models are assemblies of assumptions and simplifications and thus inevitably imperfect approximations to the complex reality. We would say all these simple parameterizations and mathematical implementations (e.g., spatial and temporal discretization) can lead to model structure error. We reformulated the sentences in the new manuscript (line 577-581):**

*"Also, we account crudely for errors in LSM model formulation via parameter uncertainty and the use of a stochastic description of the precipitation record of the Rollesbroich site (discussed next). In other words, the $k \times 1$ process noise vector, $\mathbf{w}_t$, in equation (2) consists of zeros. However, we*

*agree that it can be expected that we have other model structural errors, for example in relation to the representation of photosynthesis."*

*Minor comments:*

*3. Table 5: In original manuscript Figure 8, there was basically no difference for the prediction of the water content of the first layer, whether there are parameters estimated or not. The authors clarified this: "Predictions of soil moisture content for layer 2 and layer 3 (in the verification period) improved significantly for the case of parameter estimation. Concerning the soil moisture content of layer 1, the RMSE value of the open loop run is 0.053m3/m3, which is already quite close to the observed values. In addition, the soil moisture content for the upper layer is strongly driven by single precipitation events. We extended the discussion of these results (line 533-536): "In the verification period, the RMSE values of the scenario noParamUpdate are close to the RMSE values of the open loop run. If soil parameters were updated during the assimilation period, the RMSE values for soil moisture characterization were reduced. More specifically, the four methods show a RMSE improvement of about 54% and 42% for the second and third model layer (compared with the open loop run)."*

*The part of the answer ("Concerning the soil moisture content of layer 1, the RMSE value of the open loop run is $0.053 m^3/m^3$, which is already quite close to the observed values. In addition, the soil moisture content for the upper layer is strongly driven by single precipitation events.") was added to the results for CLM instead of VIC-3L. Please correct. Furthermore, to me this statement doesn't seem entirely consistent with the discussion (line 675-682): "In the verification period soil moisture of the top layer cannot be represented perfectly by the two LSM's, in spite of parameter updating with state of the art data assimilation methods. Table 5 and table 9 illustrate that the RMSE values of the four joint state and parameter assimilation methods are similar for both models, which means that both models have larger errors for the top layer. There is a number of reasons for the larger soil moisture mismatches for the upper layer: (i) the memory effect from initial conditions, very well identified at the beginning of the verification period, is smaller for the upper soil layer, as this layer is more affected by precipitation events and evaporation; (ii) these soil moisture changes make that it is also more affected by model structural errors, for example concerning evaporation processes."*

**Reply:** **We discussed this issue in the discussion (line 877-881):**

**"Despite this improvement in model performance over an open-loop simulation, VIC-3L and CLM do not adequately characterize soil moisture dynamics of the top layer (5 cm measurement depth) during the evaluation period (RMSE values of about 0.05 $cm^3/cm^3$). We posit that these two models do not characterize adequately processes such as water infiltration, soil evaporation, and/or root water uptake (transpiration), which govern rapid variations in soil moisture storage in the top soil."**

*7. Figure 11: I agree that the shown parameters are more meaningful than the estimated soil texture. I still think it is worth to mention (e.g. in the caption), that the shown parameters are not the directly estimated ones.*

**Reply:** **The text was added in the caption of figure 11 in the revised manuscript:**

**"Please note that $\log_{10}ks$ (in $\log_{10}(m/s)$) and parameter B are derived from the sand, clay, and organic matter fractions of each soil layer, which are estimated during the assimilation period."**

*9. Concerning the reply about RMSE: "Thanks, we admit that for a prediction in the verification phase we cannot expect a better result than a RMSE equal to the measurement uncertainty but still an RMSE equal to 0 would be the best result." A RMSE equal to 0 is not the best result. It would mean, that the model perfectly describes the measurement noise, but not the actual state.*

**Reply:** **Thanks, you are right. But in reality we don't know the actual state values, so we still think that a smaller RMSE value indicates a better prediction (line 645-646):**

**"Larger values of the NSE and smaller values of the RMSE are preferred as they indicate a better LSM performance."**

[revised manuscript text omitted]

---

## Author Response (AR4)

**Editor Decision: Publish subject to technical corrections (17 July 2017) by Kurt Roth**

*Comments to the Author*:

*This manuscript took an unusually long path but now is an overall good contribution.*

*I am still concerned about the issue of a significant parameter dependence on the choice of the random generator's seed (hess-2016-42-author_response-version3.pdf, pg 3). Such a dependence is an alarming signal in my understanding of these methods and certainly warrants further exploration. Just having the proposed sentence*

*"This initial parameter ensemble is the same for all the assimilation methods."*

*in the manuscript neither addresses the source of the problem nor gives any hint to the reader. I am not content with this approach! Still, as far as I can see it, this does not have an impact on the message of this paper. I thus*

*1. ask the authors to take the chance and set things straight by at least mentioning the issue in the manuscript*

*2. other than that accept the paper as is for publication.*

**Reviewer I**

*I thank the authors for answering the previously asked questions again.*

*I have one more comment about the changing results (compared to an earlier version of the manuscript). The authors state in their response, that this change happened due to a change in the random seed. I think it should be mentioned in the paper, that the results for that parameter change a lot with the seed.*

*I will leave the decision, if this is an issue to the editor.*

*Apart from this, I think the manuscript can be published.*

**Reply: Thanks for your work on the review. We are very happy to see the acceptance of this paper. Since these two questions are similar, we answer them together.**

**From our work, we repeated our experiments for several times with different random seeds generating initial parameter ensemble (i.e., initial parameter ensemble in this version differs from initial parameter ensemble in the earliest version), we found parameter evolution of EnKF-AGU, EnKF-DUAL and PF are slightly affected by the initial parameter ensemble (or random seeds) and parameter evolution of PMCMC is more affected by the initial parameter ensemble ( or random seeds) (comparing figures of parameter evolution in this version and the earliest version).**

**In PMCMC method, MCMC resampling step allows for relatively large moves in the parameter space which causes this method may have different results with different initial parameter ensemble. In the manuscript, we explained this issue with more details in the analysis of the parameter evolution in the assimilation period for VIC-3L model (line 695-702):**

*"PMCMC exhibits significant temporal dynamics. This is not surprising, and a consequence of the MCMC resampling step that is used to rejuvenate the parameter samples (e.g. Vrugt et al., 2013). In the first place, the DREAM-type proposal distribution that is used to create candidate particles allows for relatively large moves in the parameter space. Second, only a small LSM trajectory between two successive soil moisture observations is used to determine the acceptance probability of each candidate particle. With such a short (re)-simulation period, insensitive parameters are allowed to transition to very different values, as they do not affect the model output between the two observations, and thus likelihood of a candidate particle."*

**To address the effect of initial parameter ensemble on the parameter evolution, we added new sentences in the new manuscript (line 702-704):**

*"Altogether, this also contributes to a stronger dependency of PMCMC on the initial parameter ensemble. This collection of parameter vectors is drawn randomly from the prior parameter distribution and differs per trial depending on the random seed."*